

# Simulating liquid water distribution at the pore scale in snow: water retention curves and effective transport properties

Lisa Bouvet[1,2], Nicolas Allet[1,2], Neige Calonne[1], Frédéric Flin[1], and Christian Geindreau[2]

[1]Univ. Grenoble Alpes, Université de Toulouse, Météo-France, CNRS, CNRM, Centre d'Études de la Neige, Grenoble, France
[2]Université Grenoble Alpes, CNRS, Grenoble INP, 3SR, Grenoble, France

**Correspondence:** Lisa Bouvet (lisa.bouvet@meteo.fr) and Frédéric Flin (frederic.flin@meteo.fr)

**Abstract.**

Liquid water flows by gravity and capillarity in snow and modifies drastically its properties. Unlike dry snow, observing wet snow remains a challenge and data from 3D pore-scale imaging are scarce. This limitation hampers our understanding of the water, heat and vapor transport processes in wet snow, as well as their modeling. Here, we explore a simulation-based approach

in which 3D images of dry snow were digitally filled and drained with water through imbibition and drainage simulations driven by capillarity. Time series of wet snow images at various water contents were produced. The water retention curves, i.e. the capillary pressure as a function of the water content, are derived for different snow types. New parameters to model the water retention curves of snow are proposed based on the van Genuchten model and compared to existing ones from laboratory experiments. From the 3D images, the hydraulic conductivity, water permeability, effective thermal conductivity, and water

vapor diffusivity of wet snow were computed. We analyzed their evolution in relation to water content, density and snow type, compared them with existing regressions and presented new ones when needed. The proposed relationships are a step forward in improving the physical description of wet snow in snowpack scale models.

## 1 Introduction

Wet snow is characterized by the presence of liquid water in the snow microstructure. The liquid water, mostly introduced

by rain or melt events, often flows from the surface downwards through the snowpack and implies drastic changes, which can lead to wet snow avalanches (e.g., Schweizer et al., 2003) and the release of large amounts of water and flooding (e.g., Singh et al., 1997). Those changes are associated with morphological transformations of the snow microstructure called wet snow metamorphism (e.g., Wakahama, 1968; Colbeck, 1973; Raymond and Tusima, 1979; Marsh and Woo, 1985), which are characterized by the formation of large, rounded, often clustered grains, referred to as melt forms (MF) (Fierz et al., 2009).

The physics of liquid water transport in snow is complex and involves water flow by gravity and capillarity, which can present features such as hysteresis wetting-drying processes, preferential flow or capillary barriers. Water transport is coupled with heat transport, which includes heat conduction and convection, as well as latent heat from phase changes. To predict wet snow, detailed snow cover models rely on a modeling of the liquid water transport in snow based on simplified representations, such as the bucket-type approach in the Crocus model (Vionnet et al., 2012). Recently, more sophisticated models of water flow in





snow have been presented and describe processes such as water flow by capillarity or preferential flow along water channels (e.g., Daanen and Nieber, 2009; Hirashima et al., 2014; Leroux and Pomeroy, 2017; Moure et al., 2023; Jones et al., 2024). These models were little evaluated, and rely on effective wet snow properties, including specific hydraulic properties of snow for water flow, which are often estimated empirically.

For unsaturated porous media, the movement of water is classically described by the Richards equation, expressing the volu-

metric liquid water content evolution (Richards, 1931):

$$\frac{\partial \theta_w}{\partial t} = \frac{\partial}{\partial t}\left(\mathbb{K}_w^{\mathrm{u}}(\theta_w)\left(\frac{\partial h}{\partial t}+1\right)\right) - S \tag{1}$$

with $\theta_w$ (-) the volumetric liquid water content, $h$ (m) the liquid pressure head (linearly related to the capillary pressure), $\mathbb{K}_w^{\mathrm{u}}$ (m s$^{-1}$) the unsaturated hydraulic conductivity, and $S$ (s$^{-1}$) a source/sink term. In general porous media applications, the liquid pressure head $h$ is expressed as a function of the water content $\theta_w$ through the water retention curve (WRC). The WRC, de-

scribing the evolution of capillary pressure in the pores as a function of the water content, reflects the hydraulic behavior of any porous medium. This relationship can be provided for imbibition, which consists in wetting a dry porous medium by a liquid until it reaches saturation, or for drainage, which consists in letting the liquid contained in a fully saturated porous medium drain off. WRCs are classically modeled by the van Genuchten (VG) model (van Genuchten, 1980), initially developed and widely-used for soil. Such a model includes several hydraulic parameters that depend on the morphology of the material being

considered.

WRCs have been traditionally measured by imbibition or drainage experiments in the laboratory. For soils, drainage experiments were performed using hanging water columns (drainage by gravity) or by pressure plates (drainage by a pressure controlled pump). For snow, few WRC measurements have been conducted and are mostly limited to drainage (e.g., Colbeck, 1974; Yamaguchi et al., 2010, 2012; Katsushima et al., 2013). Overall, most authors report large dependencies on snow density

and grain size. Recently, magnetic resonance imaging methods were adapted by Adachi et al. (2020) to measure WRCs in snow for both imbibition and drainage. This method enables to capture the liquid water content in 3D at a mm-scale resolution. The authors have revealed a significant hysteresis between both processes, which shows that, as for most porous media, drying and wetting in snow are distinct processes that should be considered separately. Some studies related the VG model to experimental WRCs and obtained estimates of the VG parameters for snow. Regressions were presented for these parameters based on snow

microstructural properties. Daanen and Nieber (2009) and Yamaguchi et al. (2010) proposed regressions depending solely on the snow grain size, based on a few drainage experiments on similar snow samples composed of rather dense MF. Later, Yamaguchi et al. (2012) presented improved regressions based on additional drainage experiments, which depend on both the snow density and grain size. Again, this study focuses only on MF samples, limiting the observations to coarse-grained, high-density snow. No estimates of the VG model parameters were provided for imbibition in snow.

While the classical experiments of imbibition and drainage enable to capture the overall hydraulic behavior of a snow sample, necessary to resolve the Richard equations, they do not provide the exact distribution of liquid water at the pore scale inside the snow microstructure. Accessing 3D wet snow microstructures at varying saturation levels would be critical to estimate the effective properties specific to wet snow, in order to feed macro-scale models. For dry snow, accurate estimates of the





effective properties are obtained by computations based on pore scale 3D images of dry snow obtained by X-tomography (e.g.,
Kaempfer et al., 2005; Calonne et al., 2011; Zermatten et al., 2011; Calonne et al., 2012, 2014a; Wautier et al., 2015; Fourteau
    et al., 2021b; Bouvet et al., 2022). Parameterizations of effective properties, such as the effective thermal conductivity and
    the effective water vapor diffusivity, have been derived and are used to model heat and mass transport in dry snow in macro-
    scale models (e.g., Calonne et al., 2014b; Hansen and Foslien, 2015; Calonne et al., 2015; Brondex et al., 2023; Bouvet et al.,
    2024). However, performing X-ray tomography on wet snow samples is still a challenge, notably due to the lack of contrast
between the X-ray absorption of ice and liquid water. The litterature studies propose refrozen states of wet snow (e.g., Flin
    et al., 2011; Avanzi et al., 2017) which do not allow a precise imaging of the ice-water interface. Direct imaging of wet snow is
    presently only achieved by MRI (Adachi et al., 2020), with mm-scale images, whereas observing pore scale processes requires
    a μm-scale resolution. Because of these limitations, our knowledge of the wet snow properties is not as advanced as that of dry
    snow. Macro-scale models such as the one derived by Daanen and Nieber (2009), Leroux and Pomeroy (2017) and Moure et al.
(2023) rely on effective wet snow properties which are often estimated empirically or directly extrapolated from dry snow as
    done for the thermal conductivity in Crocus (Lafaysse et al., 2017).
    In this work, we circumvent this obstacle by exploring a new simulation-based approach in which liquid water is digitally filled
    into a 3D image of dry snow, and then drained out of the porous microstructure. For that purpose, the distribution of liquid wa-
    ter in the microstructure is predicted, following the capillarity laws applied to an ice and air porous medium. As the traditional
numerical method based on solving the Navier-Stokes equation in each phase is numerically too costly, we focus here on a
    quasi-static approach (e.g., Hu et al., 2018; Konangi et al., 2021). Of course, wet snow metamorphism effects, characterized
    by the morphological changes driven by the presence of liquid water such as the rapid rounding and coarsening of grains are
    not accounted here and the results provided in this study should be considered as valid only for the initial state of the snow
    evolution. We thus propose here an exploratory numerical study where a quasi-static model is applied to 34 experimental 3D
tomography images of dry snow presenting diverse snow types and in which water is numerically introduced by capillarity
    and removed according to an imbibition-drainage pattern. Based on these series of wet snow images, the WRC of snow and its
    variations with the snow microstructure are first investigated. The VG model is applied to the simulated WRCs, which allows to
    derive its parameters for a large range of microstructures. The obtained parameters are used to propose new regressions for both
    imbibition and drainage cases, which are compared to the regressions of Daanen and Nieber (2009), Yamaguchi et al. (2010)
and Yamaguchi et al. (2012). Next, numerical computations on the simulated images of wet snow are performed to estimate
    the effective transport properties of wet snow, as required for the water flow, heat and vapor transport modeling. The investi-
    gated properties are the relative permeability (i.e. the unsaturated hydraulic conductivity), the effective thermal conductivity,
    and the effective water vapor diffusivity. The numerical results are compared with commonly-used estimates of the literature,
    such as the VGM model (Mualem, 1976; van Genuchten, 1980). New regressions of effective thermal conductivity and vapor
diffusivity are presented to account for saturation.



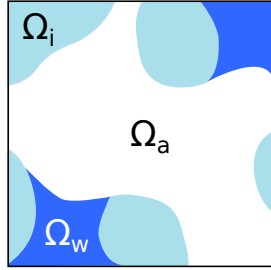

**Figure 1.** Illustration of the micro-scale air, ice and water phases in wet snow.

## 2 Methods

### 2.1 Definition of the micro-scale variables

Within the representative elementary volume (REV) of snow noted $\Omega$, we define $\Omega_i$, $\Omega_w$ and $\Omega_a$ the volumes occupied by the ice, the water and the air phase, respectively, as illustrated in Figure 1. The subscripts $(_i)$, $(_w)$ and $(_a)$, are related to quantities defined in $\Omega_i$, $\Omega_w$ and $\Omega_a$, respectively. The volume fractions $\theta$ of the different phases (ice, water, air) are written:

$$\theta_i = \frac{\Omega_i}{\Omega}, \quad \theta_w = \frac{\Omega_w}{\Omega}, \quad \theta_a = \frac{\Omega_a}{\Omega}, \tag{2}$$

with $\theta_i + \theta_w + \theta_a = 1$. The porosity $\phi$, the water saturation $S_w$ and air saturation $S_a$ are respectively defined as:

$$\phi = \frac{1 - \Omega_i}{\Omega} = \frac{\Omega_a + \Omega_w}{\Omega}, \quad S_w = \frac{\Omega_w}{\Omega_a + \Omega_w}, \quad S_a = \frac{\Omega_a}{\Omega_a + \Omega_w}, \tag{3}$$

which leads to:

$$\phi = 1 - \theta_i, \quad S_w + S_a = 1, \quad \theta_w = \phi S_w, \text{ and } \quad \theta_a = \phi S_a = \phi(1 - S_w). \tag{4}$$

### 2.2 3D images of snow and microstructural properties

The set of 3D images of snow used in this study for the simulations corresponds to the one from Calonne et al. (2012) (see the related supplement for details). This set consists of 34 images of different dry snow microstructures obtained by X-ray micro-tomography, covering a great diversity of snow properties in terms of density, grain size and snow type. The image volumes are cubic, with sides ranging in size from $\sim 2.5$ to 10 mm and resolution from $\sim 5$ to 10 μm. Of these 34 images, 5 images representative of the diversity of all the images were selected for detailed investigations. They include two images of melt forms at different densities and grain sizes, and one image of depth hoar, rounded grains and precipitation particles. Their main characteristics are shown in Table 1. To characterize these dry snow microstructures, we rely on the snow density $\rho$ in kg m$^{-3}$ (in the following the snow density will always refer to the dry density without liquid water) and the spherical equivalent radius $r_{es}$ in m, derived from the specific surface area (SSA) as $r_{es} = 3/(\text{SSA} \times \rho_i)$ with $\rho_i$ the ice density. Both snow density and SSA values were provided by Calonne et al. (2012) (see the related supplement for the detailed table) based on 3D image computations using simple voxel counting and a stereological method (Flin et al., 2011).





**Table 1.** Main characteristics of the 5 selected 3D images.

| Name | Snow type | Image size (voxel) | Resolution (μm) | $\rho$ (kg m$^{-3}$) | $r_{es}$ (mm) |
|------|-----------|--------------------|-----------------|----------------------|---------------|
| NH2 | MF | 651 | 8.6 | 503 | 0.53 |
| NH5 | MF | 1000 | 9.5 | 473 | 0.87 |
| grad3 | DH | 600 | 10 | 369 | 0.15 |
| 0A | RG | 700 | 8.4 | 315 | 0.12 |
| fr | PP | 1192 | 4.9 | 125 | 0.06 |

## 2.3 Numerical simulations of imbibition and drainage

To simulate the distribution of two distinct fluid phases in a porous material, we used the SatuDict module of the Geodict

software (Math2Market GmbH) based on a pore morphology method-based algorithm (Thoemen et al., 2008). The model assumes that water flow is in steady state and controlled solely by capillary pressure, and therefore by the pore morphology, using the Young-Laplace equation. Gravity and viscous effects are neglected compared to capillary pressure, which is a good approximation for small pore sizes and low viscosity fluids as here. Interface dynamics, as generated by phase changes, are not described. The model appears to work well for fluids with contact angles close to 0 (here 12° between ice and liquid water),

and for microstructures with rather spherical pores (Hu et al., 2018).

Based on the above approximations, we used SatuDict to generate series of wet snow at different water contents from the 34 tomographic images of dry snow. The model provides the 3D distribution of the air and liquid water phases inside the ice structure for different water contents. This can be seen as simulating the process of imbibition or drainage in snow as a sequence of equilibrium states. In more details, the following Jurin equation applied for snow is solved:

$$h = \frac{2\gamma}{r\rho_w g}cos(\psi) \tag{5}$$

where $h$ is the liquid pressure head in m, $\gamma = 0.0756$ N m$^{-1}$ is the surface tension at 0°C, $\psi = 12°$ is the contact angle between ice and liquid water, $r$ in m is the minimum radius of accessible pores, $g$ is the gravitational acceleration in m s$^{-2}$ and $\rho_w$ is the density of water in kg m$^{-3}$. The evolution of the air and liquid water in the ice structure can be seen as a series of erosion and dilation operations that depends on the pore distribution (Ahrenholz et al., 2008).

In this study, we simulated the following sequence: first the imbibition process is simulated on an initial dry snow image until the saturation is reached, then, the drainage process is simulated on the saturated image obtained after imbibition. This configuration can be seen as wetting a snow layer by capillarity from the bottom, with liquid rising inside the snow sample, and then draining the snow layer by gravity. This situation can be found when liquid water percolated through water channels and pounds over a less permeable layer so that the water flows horizontally and can reach a dry snow area where it rises by

capillarity (e.g., Coléou et al., 1999; Williams et al., 2010). For both imbibition and drainage, the boundary conditions consist of an air reservoir located at the top of the snow sample and a water reservoir located at the bottom. Periodic conditions are applied on the sides of the samples.





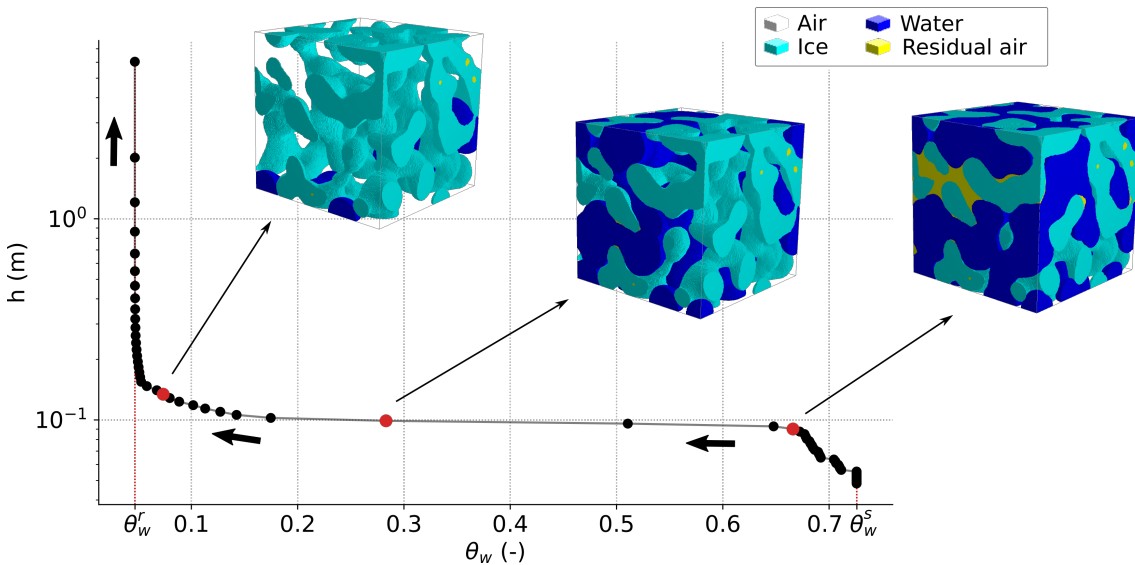

**Figure 2.** Example of a water retention curve estimated from a drainage simulation on a 3D tomographic snow sample of melt forms. The corresponding drying path is represented by the arrows. The simulated 3D water distribution in the pores is shown at 3 different stages.

For imbibition, the simulation starts with the 3D image of dry snow. A minimum pore radius is defined according to Eq. (5), which then increases step by step. At each time step, a pore is filled with liquid water if (i) the pore radius is below the minimum pore radius, and (ii) the liquid water phase is connected to the liquid water reservoir.

For drainage, the simulation starts with the 3D image of water saturated snow obtained at the end of the imbibition simulation. A maximum pore radius is defined according to Eq. (5), which then decreases step by step. At each time step, a pore is filled with air if (i) the pore radius is larger than the maximum pore radius, (ii) the entering air phase is connected to the air reservoir, and (iii) the initial liquid water phase of the pore is connected to the liquid water reservoir. Those conditions enable residual water or residual air to remain trapped in the porous media after the imbibition and drainage simulations, respectively. The output of the algorithm is a sequence of quasi-stationary three-phase distributions, which are used in SatuDict to compute the saturation-dependent water retention curve (WRC). Finally, we ensured that the simulations were performed on 3D images of sufficient size, of at least the size of the REV using increasing volumes (see Fig. 4.a).

## 2.4 Water retention curve analysis

Results from the SatuDict module are used to determine the WRC, by expressing the liquid pressure head $h$ as a function of the liquid water volumic fraction $\theta_w$, for both imbibition and drainage. An example of the WRC obtained from a drainage simulation on a small MF sample is shown in Fig. 2. The plot should be read from right to left, as water is gradually drained out of the porous microstructure. The liquid pressure head increases with decreasing water content, characterized by a sharp rise at the very beginning and very end of drainage and a near constant value for the intermediate water contents. The 3D images





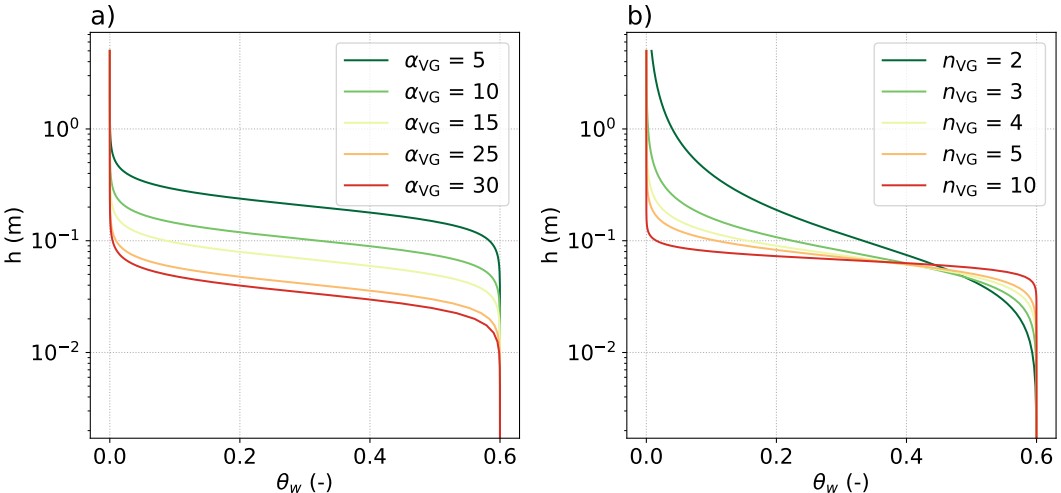

**Figure 3.** WRCs from the VG model: influence of a) the parameter $\alpha_{\mathrm{VG}}$ (with a fixed $n_{\mathrm{VG}}$ = 5) and of b) the parameter $n_{\mathrm{VG}}$ (with a fixed $\alpha_{\mathrm{VG}}$ = 15).

show the distribution of the air (transparent and yellow phases) and water (dark blue) in the pore space of the snow at different points on the WRC. WRCs also provide the residual water content $\theta_w^r$, being the remaining water content after drainage, and the saturated water content $\theta_w^s$, being the maximum amount of water the snow volume can store. Both parameters are illustrated in the figure as the end points of the WRC.

To further analyze the WRCs estimated from the different snow images, the curves are fitted to the van Genuchten (VG) model
(van Genuchten, 1980), which reads as follows:

$$\theta_w(h) = \theta_w^r + (\theta_w^s - \theta_w^r) \times (1 + (\alpha_{\mathrm{VG}} \, h)^{n_{\mathrm{VG}}})^{-m_{\mathrm{VG}}} \tag{6}$$

where $\alpha_{\mathrm{VG}}$, $n_{\mathrm{VG}}$ and $m_{\mathrm{VG}}$ are the unknown parameters that determine the shape of the WRC. These parameters depend on the snow microstructure and can be estimated by fitting the VG model to WRCs obtained experimentally (Daanen and Nieber, 2009; Yamaguchi et al., 2010, 2012). The parameter $m_{\mathrm{VG}}$ is usually approximated by $m_{\mathrm{VG}} = 1 - 1/n_{\mathrm{VG}}$ (e.g., Yamaguchi
et al., 2010). The influence of the parameters $\alpha_{\mathrm{VG}}$ and $n_{\mathrm{VG}}$ on the WRC is illustrated in Fig. 3. $n_{\mathrm{VG}}$ defines the steepness of the curve inflections and $\alpha_{\mathrm{VG}}$ defines the values of $h$ at which inflections occur. Roughly, the parameter $\alpha_{\mathrm{VG}}$ can be related to the average pore size and $n_{\mathrm{VG}}$ to the pore size distribution (e.g., Yamaguchi et al., 2010). In what follows, the WRCs of each imbibition and drainage simulation are considered and used to fit the VG model. For fitting, $\theta_w^s$ is set equal to the snow sample porosity and $\theta_w^r$ is set to 0 for imbibition and to the minimum value of the $\theta_w$ obtained from the drainage simulations. These
choices are justified by the fact that our simulations always start with imbibition, i.e. without any liquid water in the pores, and end with drainage, at the end of which the volume of the air residuals is negligible (around 0.03% in average) as compared to the overall porous volume (see a complementary discussion on this topic in subsection 3.1.3).

Finally, as mentioned above, simulations of imbibition and drainage were performed on REVs so that we obtain representative



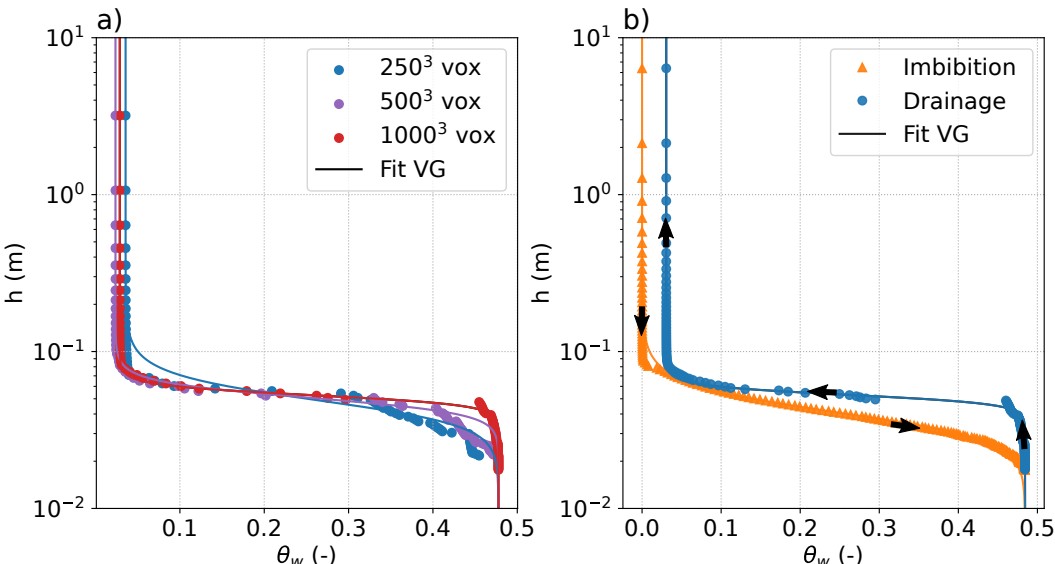

**Figure 4.** a) Influence of the volume size taken for drainage simulations on the WRC, for the sample NH5 of melt forms. Simulation data are represented by symbols and the fitted VG models by solid lines. b) Example of hysteresis of the WRC between imbibition and drainage, for the snow sample NH5. The corresponding path is represented by the arrows.

distributions of the fluids in the pores and representative WRCs. Figure 4.a illustrates the impact of the image size on the

simulated WRC. When volumes smaller than the REV are taken for simulation, the obtained WRCs are degraded and differ from the WRC obtained for a REV. Such tests were performed to assess the REV for WRCs of all our snow samples and ensure satisfactory results.

## 2.5 Computation of the effective transport properties of wet snow

Next, we study the effective transport properties of wet snow based on the 3D image series obtained from the drainage simu-

lations. We focus on the properties involved in the processes of heat, water vapor and liquid water transport, which are key for wet snow modeling. For the process of liquid water flow, the driving effective property is the water unsaturated hydraulic conductivity $\mathbb{K}_w^u$, expressed in Richards equation (Eq. 1). $\mathbb{K}_w^u$ can be expressed as a function of $\theta_w$ by $\mathbb{K}_w^u(\theta_w) = \mathbb{K}_w^{sat} \times K_w^r(\theta_w)$. $\mathbb{K}_w^{sat} = K\rho_w g/\mu_w$ is the saturated hydraulic conductivity of snow, which depends on the intrinsic permeability of snow $K$, $\mu_w$ (Pa s) being the water viscosity. $K$ depends solely on the microstructure of the dry snow and can be estimated from empirical

parameterizations, such as the one of Shimizu (1970) based on measurements or the one of Calonne et al. (2012) based on numerical computations on 3D tomographic images of snow. The term $K_w^r(\theta_w)$ corresponds to the relative water permeability at a given saturation, and is defined as $K_w^r(\theta_w) = K_w^u(\theta_w)/K$, $K_w^u(\theta_w)$ being the unsaturated water permeability. The unsaturated water permeability $K_w^u(\theta_w)$, or equivalently the relative water permeability $K_w^r(\theta_w)$, are classically modeled using the van Genuchten-Mualem (VGM) equation for unsaturated soils, which relates the permeability $K_w^u$ to the liquid water content $\theta_w$





(Mualem, 1976; van Genuchten, 1980). The processes of heat transport and water vapor transport are driven by the unsaturated effective thermal conductivity of snow $\mathbf{k}^{\mathrm{u}}(\theta_w)$ and the unsaturated effective water vapor diffusivity of snow $\mathbf{D}^{\mathrm{u}}(\theta_w)$. Both properties have been widely estimated and parameterized for the case of dry snow (e.g., Calonne et al., 2011, 2014b; Fourteau et al., 2021a, b) but very little for the case of wet snow.

To access $\mathbf{K}_w^{\mathrm{u}}(\theta_w)$, $\mathbf{k}^{\mathrm{u}}(\theta_w)$ and $\mathbf{D}^{\mathrm{u}}(\theta_w)$, the Geodict software was used to compute the 3D tensors of these three transport

properties of wet snow: the intrinsic water permeability $\mathbf{K}_w^{\mathrm{u}}$ in m$^2$, the effective thermal conductivity $\mathbf{k}^{\mathrm{u}}$ in W m$^{-1}$ K$^{-1}$ and the effective water vapor diffusivity $\mathbf{D}^{\mathrm{u}}$ in m$^2$ s$^{-1}$. Computations were performed on the series of 3D images of wet snow obtained from the imbibition and drainage simulations, so for different water contents, snow densities and microstructures. For each property, a specific boundary value problem, resulting from a homogenization technique, is solved on the REV applying periodic boundary conditions on the external boundaries of each volume. The equations to be solved are provided in the Sup-

plement and correspond to Eq. (S.1) - (S.4) for the intrinsic water permeability $\mathbf{K}_w^{\mathrm{u}}$, Eq. (S.5) - (S.15) for the effective thermal conductivity $\mathbf{k}^{\mathrm{u}}$, and Eq. (S.16) - (S.20) for the effective vapor diffusivity $\mathbf{D}^{\mathrm{u}}$. Computations of the thermal conductivity were carried out using the thermal properties of ice, air and liquid water at 0°C ($k_i = 2.14$ W m$^{-1}$ K$^{-1}$, $k_a = 0.024$ W m$^{-1}$ K$^{-1}$, $k_w = 0.556$ W m$^{-1}$ K$^{-1}$). As the non-diagonal terms of the tensors $\mathbf{K}_w^{\mathrm{u}}$, $\mathbf{k}^{\mathrm{u}}$ and $\mathbf{D}^{\mathrm{u}}$ are negligible, we consider only the diagonal terms, that are seen as the eigenvalues of the tensors. In the following, $K_w^{\mathrm{u}}$, $k^{\mathrm{u}}$ and $D^{\mathrm{u}}$ refer to the average of the

diagonal terms of $\mathbf{K}_w^{\mathrm{u}}$, $\mathbf{k}^{\mathrm{u}}$ and $\mathbf{D}^{\mathrm{u}}$.

## 3 Results and discussion

### 3.1 Water retention curves

#### 3.1.1 Hysteresis of the WRCs

A first result is that the WRCs of the snow samples present generally significant hysteresis, i.e. they differ between imbibition

and drainage. A steeper WRC is found for imbibition compared to drainage. This is illustrated in Fig. 4.b for the sample NH5 composed of large MF, showing significantly different WRCs between both processes. This result indicates that the imbibition and drainage are irreversible processes and should be considered separately, as already pointed out by previous works (e.g., Adachi et al., 2020). Both processes can be linked with the following path: (i) wetting of a dry snow sample until it reaches the saturated water content $\theta_w^s$, and (ii) drainage of the sample until it reaches the residual water content $\theta_w^r$ (following the arrows

in Fig. 4.b).

#### 3.1.2 WRCs of different snow microstructures and their related VG fits

Figures 5.a and 5.b present the WRCs of the imbibition and drainage simulations applied to the 5 selected snow samples presented in Table 1. The influence of the snow geometrical properties can be observed. Snow samples presenting small grains

(0.06 - 0.15 mm), such as the samples fr, 0A and grad3, show higher pore pressures at a given water content than the samples





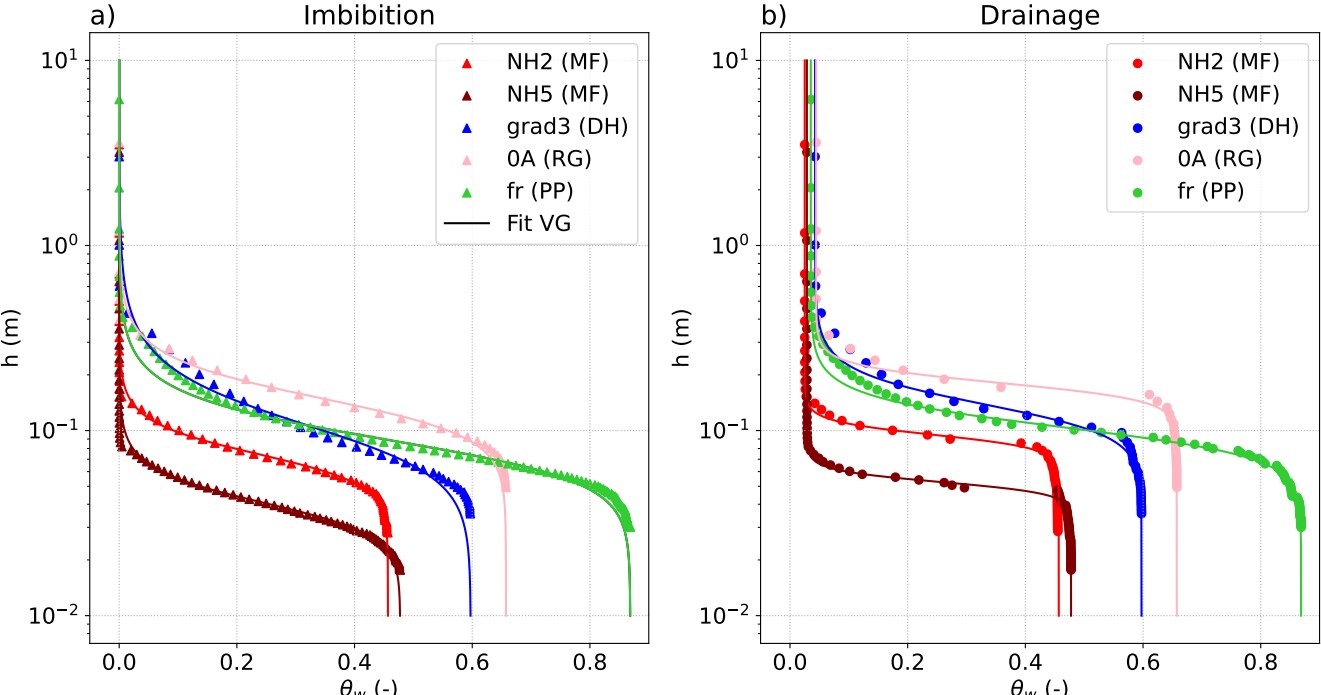

**Figure 5.** Numerical imbibition and drainage WRCs for different types of snow samples with the corresponding VG fits. The curve colors represent the different snow types.

NH2 and NH5 presenting coarse grains (0.53 - 0.87 mm). Snow density has also a direct influence on the WRCs, as it limits the maximum water content that can be reached. In a more subtle way, it also seems to influence the shape of the WRCs, which tend to show smoother inflections for lower densities, as seen when comparing the WRC of the lightest sample fr and the denser sample NH2, for example.

For each microstructure, we also present the related VG fit (Eq. 6) that best reproduce the simulated WRC. In other words, the VG parameters $\alpha_{\mathrm{VG}}$, $n_{\mathrm{VG}}$ and $\theta_w^r$ were optimized to best fit the simulated WRC of each snow sample for imbibition and for drainage. As a consequence, the fitted WRCs show good agreement with the simulated WRCs, with an $R^2$ greater than 0.98 for each sample, as illustrated for the five selected snow samples in Fig. 5a. and b. Overall, the fits show slightly better results for imbibition than for drainage, due to the sharper inflections of WRCs for the latter one.

### 3.1.3 Analysis of the VG parameters

Figure 6 presents the VG parameters $\alpha_{\mathrm{VG}}$, $n_{\mathrm{VG}}$ and $\theta_w^r$ obtained by fitting the VG model to our simulated WRCs for all our dataset. The parameters, obtained for imbibition and drainage, are expressed as a function of the term $\rho/d$ or $\rho$, following Yamaguchi et al. (2012), and compared to the regressions suggested by Yamaguchi et al. (2012) (black solid curves in Fig. 6). In addition, the measurement data of $\alpha_{\mathrm{VG}}$ and $n_{\mathrm{VG}}$ from Yamaguchi et al. (2012) and Adachi et al. (2020) are also shown with





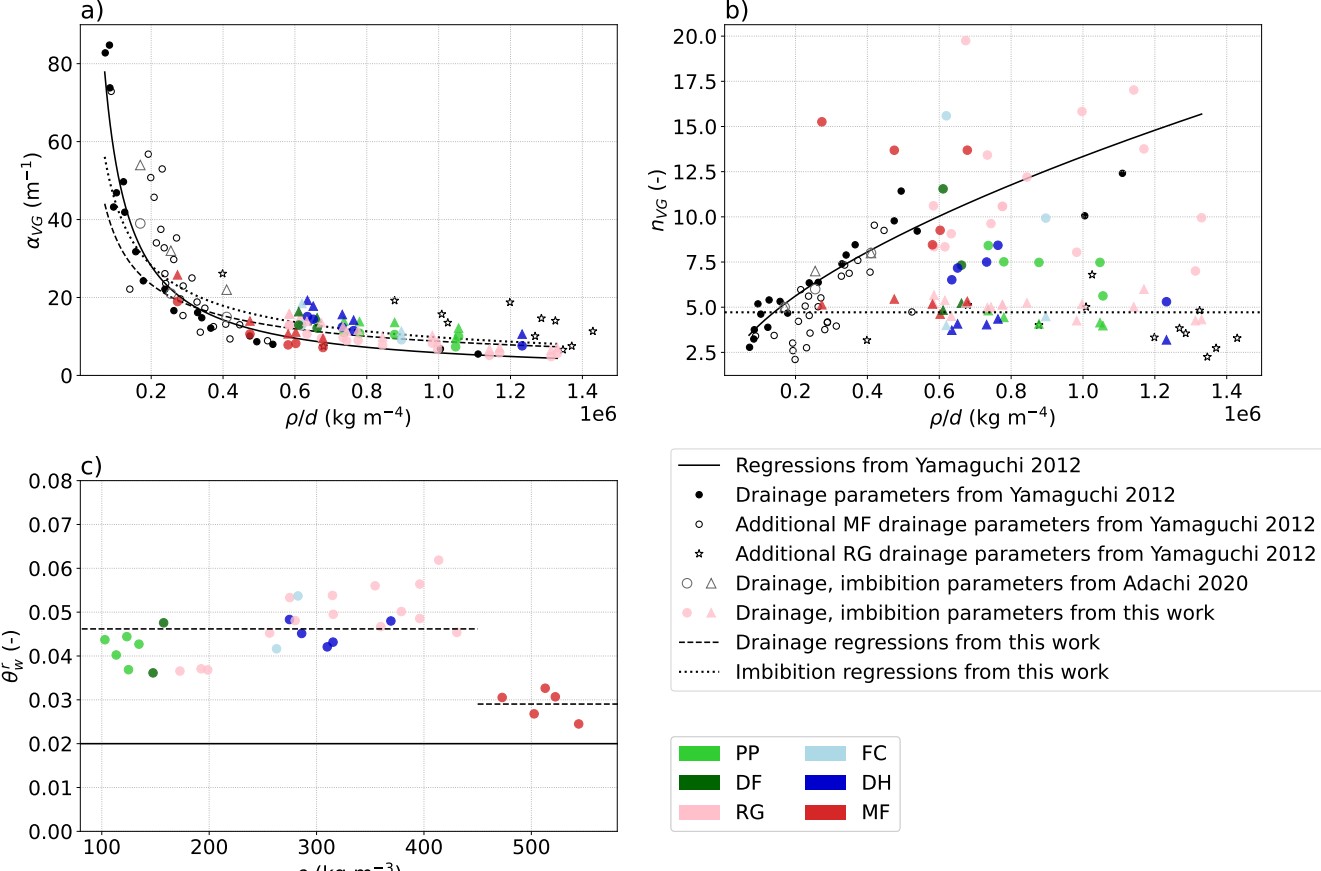

**Figure 6.** a) $\alpha_{VG}$, b) $n_{VG}$ and c) $\theta^r_w$ parameters of the VG model as a function of $\rho/d$ or $\rho$. The regressions of Yamaguchi et al. (2012) are shown by black lines, the parameter values from drainage measurements used to deduct those regressions are shown by black disks ("S-samples", composed of refrozen melt forms - see Yamaguchi et al. (2012)). The additional drainage measurements from Yamaguchi et al. (2012) are shown by circles (MF samples) and stars (RG samples). Imbibition and drainage measurements from Adachi et al. (2020) are shown by empty gray markers. From this work, parameters from imbibition and drainage simulations are shown by colored disks and triangles, respectively, the colors showing the snow types. The proposed regressions based on our simulated data are shown by dashed and dotted lines (see Table 2).

black and gray markers. They correspond to values derived from WRCs obtained from laboratory experiments of imbibition and drainage. In all our computations, $d$ was estimated based on the SSA, following the formula $d = 2 \times r_{es} = 6/(\text{SSA} \times \rho_i)$ with $\rho_i = 917$ kg m$^{-3}$ the ice density.

**$\alpha_{VG}$ parameter:**

Overall, the parameter $\alpha_{VG}$ decreases exponentially with increasing $\rho/d$, so when density increases and/or grain size de-
creases. Values obtained from the simulations of imbibition (triangle symbols) are systematically larger than the ones from



the simulations of drainage (circle symbols), in agreement with the observation of Adachi et al. (2020). The $\alpha_{\mathrm{VG}}$ values from our simulations and the regression of Yamaguchi et al. (2012) are overall in good agreement, a little less so for imbibition. Following the formulation of Yamaguchi et al. (2012), we proposed two new regressions of $\alpha_{\mathrm{VG}}$ that allow reproducing the different behaviors between imbibition and drainage, as shown in the figure. Their expressions are provided in Table 2. The

proposed regressions were derived from our snow samples, with $\rho/d$ values comprised between 0.25 and 1.3 and consequently do not perfectly capture the steep evolution of $\alpha_{\mathrm{VG}}$ for the lowest values of $\rho/d$ ($<0.25$) reported by the measurements of Yamaguchi et al. (2012) and Adachi et al. (2020).

**$n_{\mathrm{VG}}$ parameter:**

For the parameter $n_{\mathrm{VG}}$, the results obtained from the simulations are more surprising. Values from imbibition remain around

the value $n_{\mathrm{VG}} = 4.7$, showing little variations for the whole range of $\rho/d$. They do not agree with the regression of Yamaguchi et al. (2012), which predicts a gradual increase of $n_{\mathrm{VG}}$ with $\rho/d$. We thus suggest using a constant value of 4.7 to estimate $n_{\mathrm{VG}}$ for imbibition. For the drainage simulations, values are widely spread, show little correlation with $\rho/d$, and do not seem to follow the regression of Yamaguchi et al. (2012). The observed scatter could come, in part, from the performance of the fit of the VG model. As seen in Fig. 5, the fitted VG model follows less closely the simulated data from drainage compared to

imbibition, especially near the second inflection point. Quality fits at that location are more difficult because the inflection is actually not perfectly well described in the drainage simulations, with a lack of simulated data, compared to imbibition. This roughness in the data for drainage may affect the fit quality, and thus the derived VG parameters. It is however unlikely to fully explain the observed scatter. Another explanation for this scatter could be related to the relationship between $n_{\mathrm{VG}}$ and $\rho/d$. Indeed, while Yamaguchi et al. (2012) found a similar correlation between $n_{\mathrm{VG}}$ and $\rho/d$ for both of the MF sample categories

(the samples used for the regression (black disks) and the independent ones (circles)), they did not succeed for the RG samples (stars). Yamaguchi et al. (2012) attributed these different behaviors between snow types to differences in pore-size uniformity, which seems coherent with the definition of $n_{\mathrm{VG}}$. To test this hypothesis, we characterized pore size variations by using the distributions of mean curvature of the snow microstruture. The mean curvature was calculated at each point on the surface of the 3D images and represented as a statistical distribution (see Flin et al. (2004, 2005), Calonne et al. (2014a), Bouvet et al.

(2022) and the supplement for additional information). Figure 7 displays $n_{\mathrm{VG}}$ as a function of the standard deviation of the mean curvature $\sigma_{\mathrm{MC}}$ for each 3D snow image. A trend can be observed, in the form of an inverse function between $n_{\mathrm{VG}}$ and $\sigma_{\mathrm{MC}}$. Lower values of $\sigma_{\mathrm{MC}}$ tend to be correlated to larger values of $n_{\mathrm{VG}}$, which is consistent with the fact that, during drainage, water leaves the pores more or less all at once for snow with rather uniform pore shapes, such as melt forms (in red), as showed by very sharp WRCs (see Fig. 3.b). On the other extend, for snow showing large pore shape variabilities, such as fresh snow (in

light green), $n_{\mathrm{VG}}$ is low and the resulting WRC shows a smoother transition as a function of the water content, i.e. the drainage is more gradual. In conclusion, to estimate $n_{\mathrm{VG}}$ for drainage, our results do not support the use of the $\rho/d$ ratio but rather of more refined parameters, such as the mean curvature distribution (see Fig. 7 and Table 2 for the detailed regression proposed). This can be seen as a limitation for larger-scale modeling as this parameter can currently only be derived from 3D images.




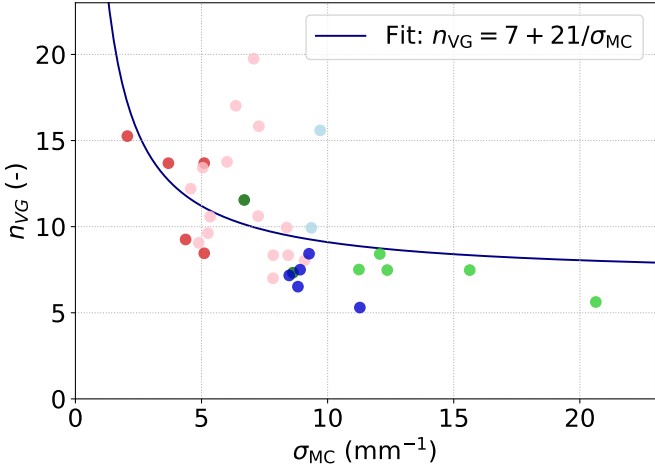

**Figure 7.** The $n_{VG}$ parameter for drainage as a function of the standard deviation $\sigma_{MC}$ obtained for the mean curvature distribution computed on each 3D dry snow image.

**$\theta_w^r$ parameter:**

The last parameter required for our use of the VG model is the residual water content $\theta_w^r$. As already mentioned, this parameter was set to 0 for imbibition as simulations were performed on fully dry snow images, and was only determined for drainage as the minimum value of water content reached during the drainage simulations. Two distinct groups are observed (Fig 6 c.). Samples with density below 450 kg m$^{-3}$ show values centered around 0.046, which slightly increase with density from about 0.036 to 0.061. Samples of melt forms with density above 450 kg m$^{-3}$ show smaller values around 0.029, including the NH2

and NH5 samples. This division is probably due to the fact that the denser snow samples, composed of MF grown under conditions of liquid water saturation, show large pores which can hold little water by capillarity. All the values of $\theta_w^r$ from our simulations are larger than the value of 0.02 proposed in Yamaguchi et al. (2012). Following our results, we propose to approximate $\theta_w^r$ by two constant values depending on the snow type: 0.029 for melt forms and 0.046 for the other snow types (Fig. 6 and Table 2).

**$\theta_w^s$ parameter:**

To complete the picture, it seems worth discussing the saturated water content $\theta_w^s$. This parameter is here approximated by the snow porosity $\phi$, as opposed to observations from the snow drainage and imbibition experiments (Yamaguchi et al., 2012; Adachi et al., 2020) which found $\theta_w^s \approx 0.9\phi$. In porous media such as soils and sands, imbibition and drainage experiments show similar results, with $\theta_w^s$ up to 30% of the porosity. Those values of $\theta_w^s$ could either be underestimated due to the exper-

imental limits linked to the challenge of filling complex geometries, or could reflect the real physical processes at stake (e.g., Clayton, 1999; Cho et al., 2022). Therefore, two approaches are commonly used in the literature, either the approximation $\theta_w^s = \phi$ is taken or $\theta_w^s$ is adjusted to experimental data. The experiments of Likos et al. (2014) and Farooq et al. (2024) showed that having $\theta_w^s$ smaller than $\phi$ generally implies greater $\alpha_{VG}$ values, but has no significant impact on the $n_{VG}$ values. For snow





|  | $\alpha_{\mathrm{VG}}$ | $n_{\mathrm{VG}}$ | $\theta_w^r$ |
|---|---|---|---|
| Daanen and Nieber (2009) | $30 \times d + 12$ | $0.8 \times d + 3$ | 0.05 |
| Yamaguchi et al. (2010) | $7.3 \times d + 1.9$ | $-3.3 \times d + 14.4$ | 0.02 |
| Yamaguchi et al. (2012) | $4.4 \times 10^6 \times (\rho/d)^{-0.98}$ | $1 + 2.7 \times 10^{-3} \times (\rho/d)^{0.61}$ | 0.02 |
| This work, imbibition | $8.9 \times 10^4 \times (\rho/d)^{-0.66}$ | 4.7 | 0 |
| This work, drainage | $4.0 \times 10^4 \times (\rho/d)^{-0.61}$ | $7 + 21/\sigma_{\mathrm{MC}}$ | 0.029 for MF samples |
|  |  |  | 0.046 for the other snow types |

**Table 2.** Regressions of the VG model parameters $\alpha_{\mathrm{VG}}$, $n_{\mathrm{VG}}$ and $\theta_w^r$ proposed by Daanen and Nieber (2009), Yamaguchi et al. (2010), Yamaguchi et al. (2012) and from this work for both imbibition and drainage.

purposes, the actual values of the saturated water content still remain unclear and should be further investigated to refine the
VG parameters.

**Conclusion on the VG parameters:**

The regressions of Yamaguchi et al. (2012) are in good agreement with our simulations for $\alpha_{\mathrm{VG}}$, especially for drainage, but
not for $n_{\mathrm{VG}}$ for both imbibition and drainage, and not for $\theta_w^r$. We recall that the regressions of Yamaguchi et al. (2012) are
based on drainage experiments only and realized on a limited number of snow types, mainly composed of dense MF, which
may explain some of the observed discrepancies. For $\alpha_{\mathrm{VG}}$, we proposed new regressions based on $\rho/d$ for both imbibition and
drainage, for a wide range of $\rho/d$ values. For $n_{\mathrm{VG}}$, a constant value was suggested for imbibition, but no estimates for drainage
could be proposed using $\rho/d$. A parameter that captures the pore size distribution of snow such as the standard deviation of the
mean curvature seems to be required. For $\theta_w^r$, two mean values were provided depending on the snow type for drainage. Finally,
we should also recall that comparing results from experiments and from our numerical simulations is not straightforward. As
already mentioned, in the experiments, the snow microstructure can evolve rapidly when in contact with liquid water, whereas
in the simulations, the ice skeleton is fixed and defined by the provided tomography image, always remaining in its initial stage.
This could also explain some differences in the WRCs.

### 3.1.4 Evaluation of the proposed VG model on two distinct snow samples

Here, we compare our results with the different VG models suggested for snow in the literature (see Table 2). All these studies
present estimates of the VG parameters $\alpha_{\mathrm{VG}}$, $n_{\mathrm{VG}}$, and $\theta_w^r$ based on microstructural properties, which allows using the VG
model for any snow sample providing the knowledge of such properties. In other terms, the regressions of Table 2 are of par-
ticular interest for the practical determination of snow WRCs.

Daanen and Nieber (2009) and Yamaguchi et al. (2010) present a regression of $\alpha_{\mathrm{VG}}$ and $n_{\mathrm{VG}}$ based on grain size, while Ya-
maguchi et al. (2012) include both grain size and snow density, using the variable $\rho/d$ with $d$ the mean grain diameter. In all
these studies, which are based on drainage measurements, a constant value of $\theta_w^r \neq 0$ is proposed.

In the present study, which is based on both imbibition and drainage simulations, we propose specific parameterizations de-





**Figure 8.** Numerical imbibition and drainage WRCs for a set of images whose VG parameters are the farthest from the regressions of Table 2 proposed in this work (E2B, for parameter $\alpha_{\mathrm{VG}}$ in imbibition, P11 for $\theta_w^r$ in imbibition, grad3 for $n_{\mathrm{VG}}$ in imbibition and I23 for $n_{\mathrm{VG}}$ in drainage - see Fig. 6, Fig. 7 and related supplement tables), with the corresponding VG models from Daanen and Nieber (2009), Yamaguchi et al. (2010) and Yamaguchi et al. (2012). The new VG model suggested in this paper is represented with solid lines.





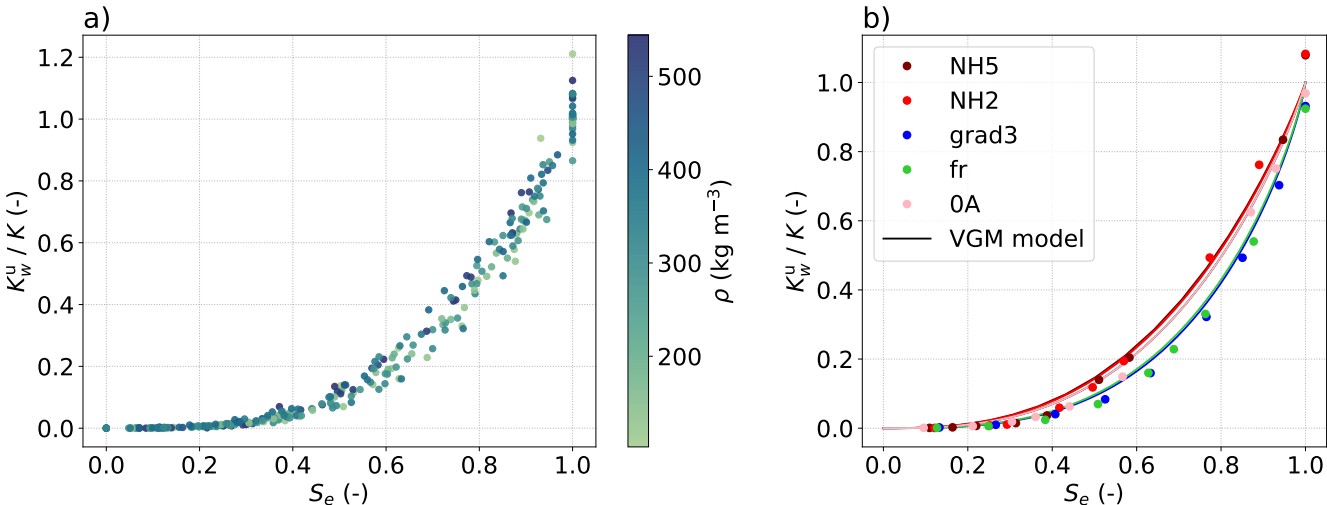

**Figure 9.** Effective relative water permeability as a function of the effective saturation for the whole set of snow samples (a) and for the 5 reference samples (b). The dry density of the snow samples is represented by the colorbar. The VGM models of relative permeability are shown by the solid lines.

pending on imbibition or drainage. Overall, the resulting WRCs provide accurate results on all our dataset. To illustrate this effect in more detail, we focused on four images whose VG parameters are the farthest from the provided regressions, namely E2B, for parameter $\alpha_{\mathrm{VG}}$ in imbibition; P11 for $\theta_w^r$ in imbibition; grad3 for $n_{\mathrm{VG}}$ in imbibition and I23 for $n_{\mathrm{VG}}$ in drainage
(see Fig. 6, Fig. 7 and related supplement tables) and plotted the simulated WRCs as well as their different VG models (Fig. 8). While the four images correspond to the worst cases for the regressions proposed in this work, our VG model still behaves very well as compared to the other models of the literature.

## 3.2 Effective wet snow transport properties

Unsaturated effective properties were computed on the simulated 3D images of wet snow obtained for different stages of the
325 drainage simulations. We present here the results for the effective water permeability $K_w^{\mathrm{u}}$, the effective thermal conductivity $k^{\mathrm{u}}$, and the effective water vapor diffusivity $D^{\mathrm{u}}$ (see Sec. 2.5). These properties are classically expressed as a function of the effective saturation (e.g., Mualem, 1976; d'Amboise et al., 2017), defined as:

$$S_e = \frac{\theta_w - \theta_w^r}{\theta_w^s - \theta_w^r}. \tag{7}$$



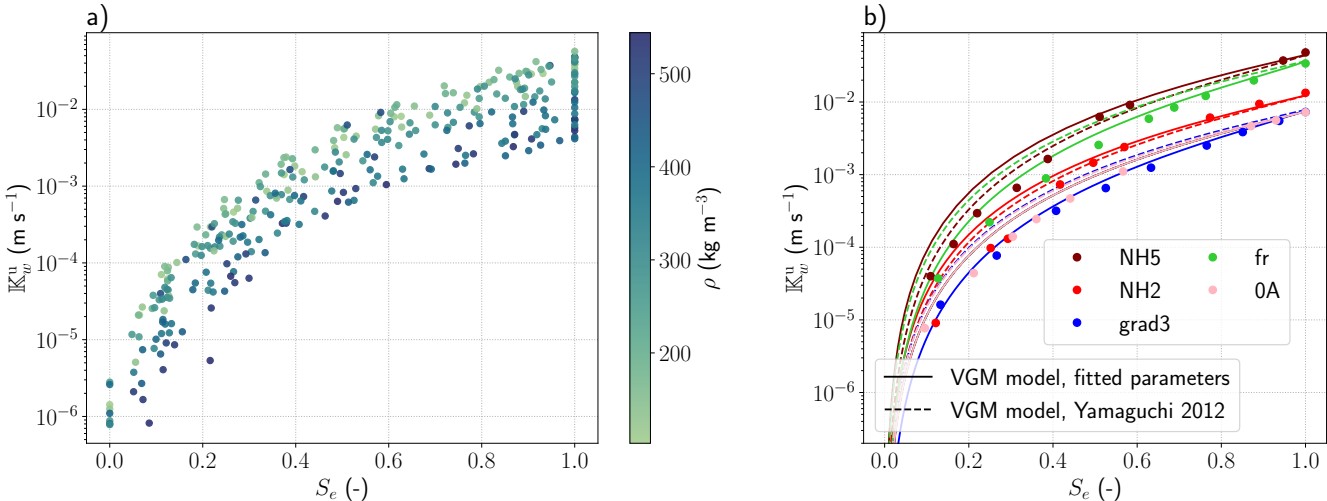

**Figure 10.** Unsaturated hydraulic conductivity $\mathbb{K}_w^u$ as a function of the effective saturation for the whole set of snow samples (a) and for the 5 selected ones (b). The dry density of the snow samples is represented by the colorbar. The VGM model, using the values of intrinsic permeability from Calonne et al. (2012) and the fitted VG parameters on each snow sample is shown with the solid lines. The VGM model, using the values of intrinsic permeability from Calonne et al. (2012) and the regression from Yamaguchi et al. (2012) is shown with the dashed lines.

### 3.2.1 Water permeability and hydraulic conductivity

First, we study the effective water permeability $K_w^u$. To compare all the samples together, we use the relative water permeability $K_w^r(\theta_w) = K_w^u(\theta_w)/K$, with $K$ the intrinsic permeability of the saturated media. In all our computations, the values of $K$ used are the numerical estimations from Calonne et al. (2012). The evolution of the relative permeability with the effective saturation is shown in Fig. 9.a and indicates an exponential increase. The evolution of all the snow samples seems to merge into a single curve, which indicates that the water permeability is primarily driven by the water content and the snow density and is little, or to a way lesser extent, influenced by the morphology of the snow microstructure. We compare our results with the van Genuchten-Mualem (VGM) model of relative water permeability, which writes as:

$$K_w^u(S_e) = K \times S_e^{1/2} \left(1 - (1 - S_e^{1/m_{VG}})^{m_{VG}}\right)^2 \tag{8}$$

with $S_e(h) = (1 + (\alpha_{VG} \, h)^{n_{VG}})^{-m_{VG}} \tag{9}$

where the VG parameters are the ones defined in Eq. 6 from the VG model (van Genuchten, 1980) and where Eq. (9) is from Mualem (1976). The VG parameters used here are the ones from the fits of the drainage WRCs (Fig. 5.b). For the 5 reference snow samples, good agreements are found between the VGM model and our numerical computations (Fig. 9.b).





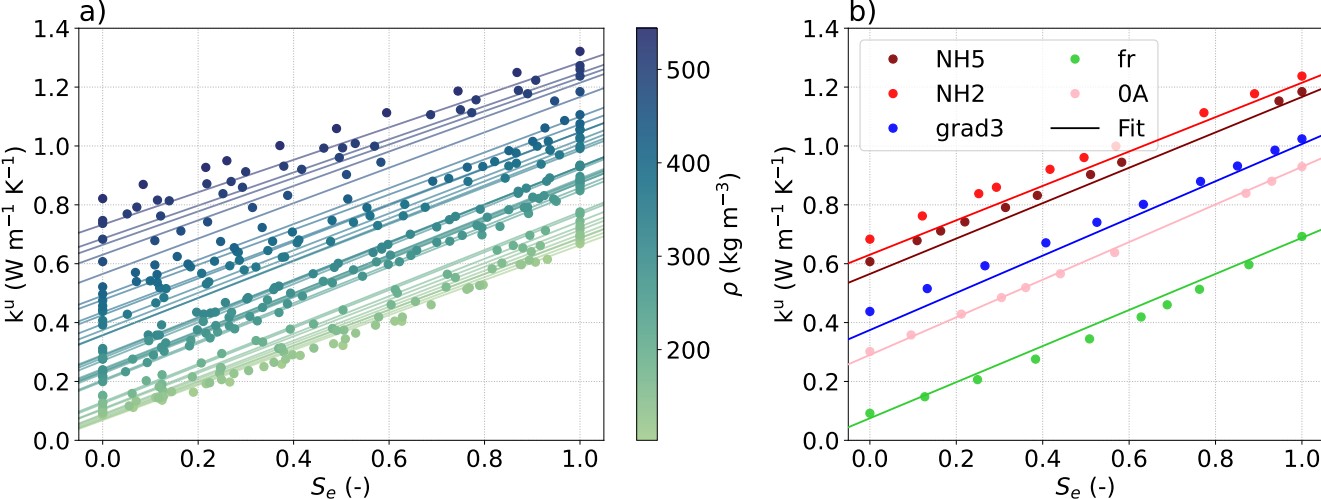

**Figure 11.** Unsaturated thermal conductivity $k^{\mathrm{u}}$ as a function of the effective saturation for the whole set of snow samples (a) and for the 5 selected ones (b). The dry density of the snow samples is represented by the colorbar. The suggested regression is shown by solid lines.

From the relative water permeability, the unsaturated hydraulic conductivity $\mathbb{K}_w^{\mathrm{u}}$ can be obtained as (see Sec. 2.5):

$$\mathbb{K}_w^{\mathrm{u}}(\theta_w) = \mathbb{K}_w^{\mathrm{sat}} \times K_w^{\mathrm{r}}(\theta_w) \tag{10}$$

with $\mathbb{K}_w^{\mathrm{sat}} = K \rho_w g / \mu_w$ (11)

$K$ depends on the snow microstructure and is estimated using the parameterization of Calonne et al. (2012) based on the density and the spherical equivalent radius of snow. Figure 10 presents the unsaturated hydraulic conductivity $\mathbb{K}_w^{\mathrm{u}}$ as a function of the effective saturation, showing an increase with increasing saturation. As $\mathbb{K}_w^{\mathrm{u}}$ contains a $K$ factor as compared to the relative permeability $K_w^{\mathrm{r}}$, an effect of the microstructure can now be observed: for a given water saturation, $\mathbb{K}_w^{\mathrm{u}}$ varies with density, so that a lighter snow shows a larger $\mathbb{K}_w^{\mathrm{u}}$, and inversely. Again, estimates of $\mathbb{K}_w^{\mathrm{u}}$ using the VGM model combined with the parameterization of Calonne et al. (2012) and the fitted VG parameters on each snow sample show overall good agreements with the computed data. For a sake of comparison, $\mathbb{K}_w^{\mathrm{u}}$ using the VGM model with $\alpha_{\mathrm{VG}}$, $n_{\mathrm{VG}}$ and $\theta_r$ from Yamaguchi et al. (2012) is shown with the dashed lines. Overall, the difference between the three estimates of the hydraulic conductivity is small, especially for high saturations.

### 3.2.2 Thermal conductivity

The unsaturated effective thermal conductivity of wet snow $k^{\mathrm{u}}$, which accounts for heat conduction in the ice, air and liquid water, is presented for different saturation levels in Fig. 11. As expected, thermal conductivity increases with increasing saturation, as liquid water conducts better than air. Values of thermal conductivity of fully saturated snow are increased of about



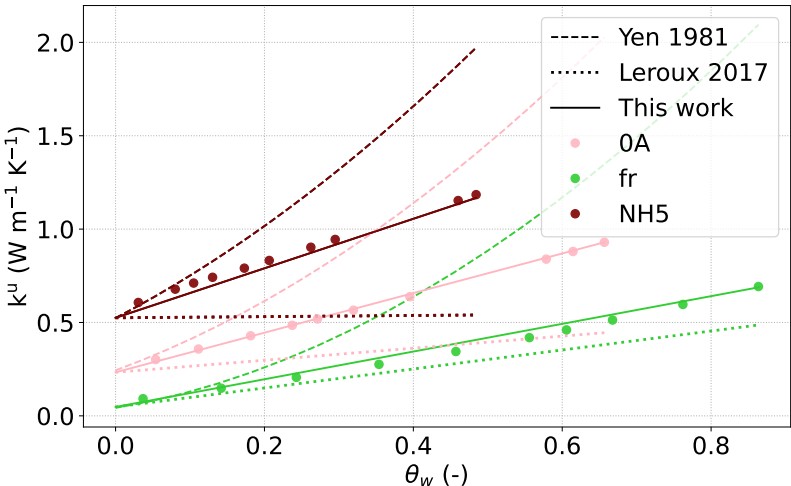

**Figure 12.** Comparison of the proposed regression of thermal conductivity of dry and wet snow with the regression from Yen (1981) extended for wet snow and the one proposed by Leroux and Pomeroy (2017), for three different snow samples.

0.5 to 0.6 W m$^{-1}$ K$^{-1}$ compared to the ones of dry snow, which means multiplying by 6 the thermal conductivity of a fresh snow sample or by 2 that of a melt form sample. The major impact of snow density is also shown, as already reported for dry snow (e.g., Sturm et al., 1997; Calonne et al., 2011). Density also influences the steepness of the linear conductivity-saturation relationship, such as dense snow shows less steep slopes than light snow. Indeed, the pore space available for a conductivity

gain due to an increase of water content is smaller for denser snow. To represent the evolution of thermal conductivity with both density and liquid water content, we propose the following regression based on our data:

$$k^{\mathrm{u}}(\rho, \theta_w) = k^{\mathrm{dry}}_{\mathrm{Calonne}}(\rho) + \theta_w \left( 1.68 \times 10^{-3} \rho + k_w - k_a \right) \tag{12}$$

$$k^{\mathrm{dry}}_{\mathrm{Calonne}}(\rho) = k_a + \rho \left( 2.5 \times 10^{-6} \rho - 1.23 \times 10^{-4} \right) \tag{13}$$

where $k^{\mathrm{dry}}_{\mathrm{Calonne}}(\rho)$ is the parameterization of thermal conductivity for dry snow of Calonne et al. (2011) and $\rho$ is the dry snow

density. The choice of the regression form was motivated by a concern for simplicity and to respect the two extreme cases: a volume fully made of liquid water ($\rho = 0$ kg m$^{-3}$ and $\theta_w = 1$) and a volume fully made of air ($\rho = 0$ kg m$^{-3}$ and $\theta_w = 0$). The case of a volume fully made of ice is not considered, as the regression of Calonne et al. (2011) is only valid for densities corresponding to snow ($\rho \leq 550$ kg m$^{-3}$). As illustrated in Fig. 11, the proposed regression reproduces well the computed data, yet slightly underestimating the values for the densest samples.

For several models of the literature, the thermal conductivity of snow as a function of the water content is needed, including snowpack models, such as the Crocus model (Vionnet et al., 2012), or detailed models of wet snow processes, such as the one of Leroux and Pomeroy (2017). They often rely on simple approximations to include the effect of water on the effective snow thermal conductivity. In the current version of Crocus, the thermal conductivity estimate from Yen (1981) $k_{\mathrm{Yen1981}}$, developed





for dry snow, is applied to wet snow by simply accounting for the volume fraction of both ice and liquid water, as:

$$k_{\mathrm{Crocus}}^{\mathrm{u}}(\rho,\theta_w) = k_{\mathrm{Yen1981}}^{\mathrm{dry}}(\rho + \theta_w \times \rho_w) = k_i \times \left(\frac{\rho + \theta_w \times \rho_w}{\rho_w}\right)^{1.88} \tag{14}$$

The model of Leroux and Pomeroy (2017) relies on a weighted average of the thermal conductivity of water and of dry snow, such as:

$$k_{\mathrm{Leroux2017}}^{\mathrm{u}}(\rho,\theta_w) = k_{\mathrm{Calonne}}^{\mathrm{dry}}(\rho) \times (1 - \theta_w) + k_w \times \theta_w. \tag{15}$$

To evaluate these two approaches, Fig. 12 presents a comparison with our data from computations and our proposed regression, for 3 snow samples. Major differences between the estimates are observed for each sample. The approximation used in Crocus largely overestimates the thermal conductivity of wet snow, up to a factor 3 for the case of light snow at full saturation. The formulation of Leroux and Pomeroy (2017) leads overall to large underestimations compared to our data. This comparison shows that the water distribution in the pore space plays an important role on the thermal conductivity of wet snow and that considering the bulk water content only is not sufficient. This motivates further studies to improve the modeling of wet snow conductivity and test the regression proposed here.

### 3.2.3 Water vapor diffusivity

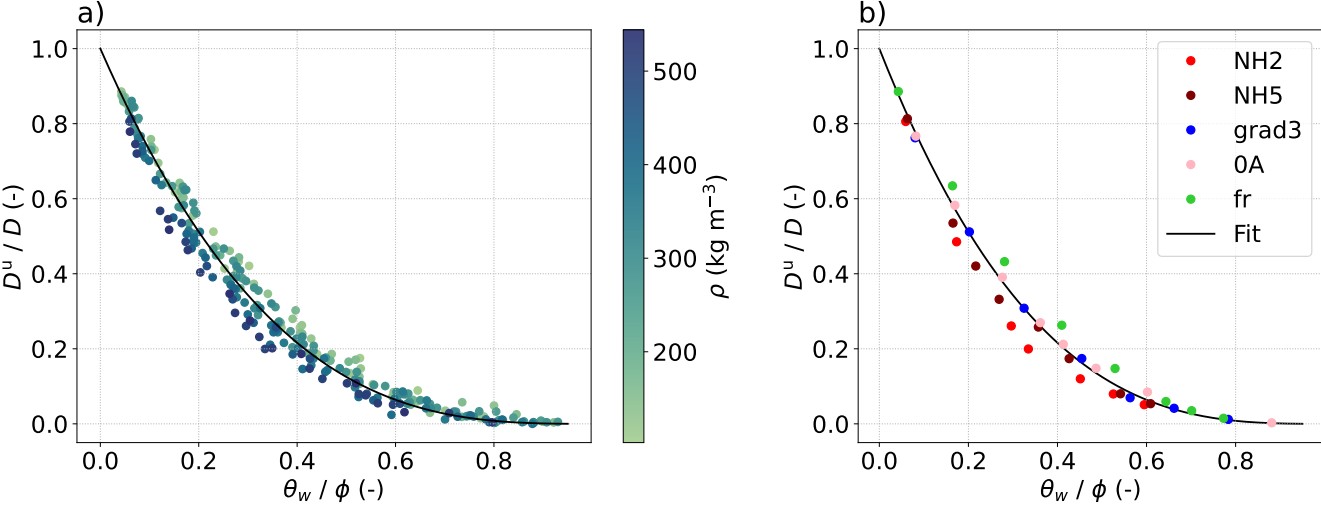

**Figure 13.** Relative water vapor diffusivity $D^{\mathrm{u}}/D$ as a function of $\theta_w/\phi$ for the whole set of snow samples (a) and for the 5 selected ones (b). The dry density of the snow samples is given by the colorbar. The proposed regression of unsaturated diffusivity is shown by a black solid line.

Extending the description of water vapor diffusion of Calonne et al. (2014b) and Bouvet et al. (2024) to the wet snow problem using a similar method as for the heat transport and the liquid water flow of Moure et al. (2023), would involve the effective



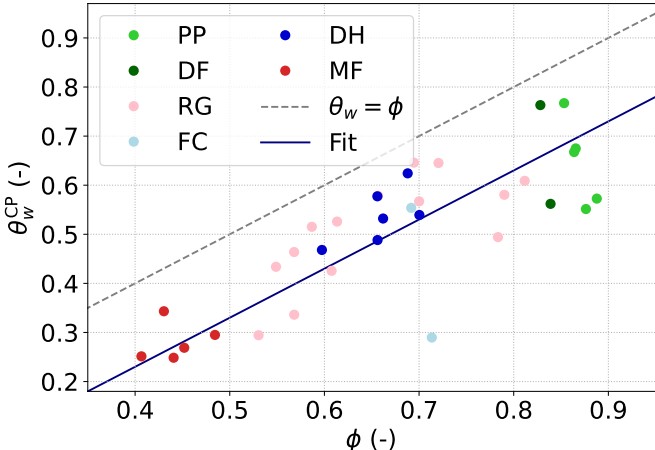

**Figure 14.** Values of the closed pore water content $\theta_w^{\mathrm{CP}}$ as a function of the snow porosity (circles). The colors represent the snow types. The saturation line $\theta_w = \phi$ is represented in gray and the proposed fit is shown in blue.

unsaturated water vapor diffusion $D^{\mathrm{u}}$. As done for the water permeability, the unsaturated diffusivity $D^{\mathrm{u}}$ is normalized, here

by the value of the dry effective vapor diffusivity $D$ using the numerical estimate from Calonne et al. (2014b).

    Figure 13 shows $D^{\mathrm{u}}/D$ as a function of $\theta_w/\phi$, with $\phi$ the porosity of dry snow. The term $\theta_w/\phi$ provides the proportion of the pore space occupied by liquid water. Very similar evolutions of the relative unsaturated water vapor diffusivity with the term $\theta_w/\phi$ are shown by all the samples, which highlights the strong correlation of the unsaturated diffusivity with both porosity and liquid water content. Diffusivity decreases exponentially as the proportion of liquid water increases, and for a given proportion

of liquid water, diffusivity is higher for low density snow, and inversely. Looking now at the maximum value of $\theta_w/\phi$ reached for each sample, we observe that this value varies, as suggested by Fig. 13.b. The domain of definition of $D^{\mathrm{u}}$ as a function of $\theta_w/\phi$ is determined by the value of water content at which the pore space is obstructed by liquid water, and so when $D^{\mathrm{u}}$ can no longer be estimated. This specific value of water content is referred here as the closed pore water content $\theta_w^{\mathrm{CP}}$. It is similar to the close-off density at which the air can no longer diffuse in the pore space, as used in Calonne et al. (2022) for dry snow and

firn samples. Values of $\theta_w^{\mathrm{CP}}$ for all our snow samples are presented as a function of the dry porosity in Fig. 14. The closed pore water content $\theta_w^{\mathrm{CP}}$ is significantly smaller than the dry porosity, in average 17% smaller. For comparison, the saturated water content $\theta_w^s$ is in average only 0.03% smaller than the porosity (see section 2.4). This means that vapor diffusion in wet snow is no longer possible long before saturation is reached. $\theta_w^{\mathrm{CP}}$ increases with increasing porosity (decreasing density), but does not seem specifically impacted by the type of snow considered. A fit is estimated here as $\theta_w^{\mathrm{CP}} = \phi - 0.17$ (in blue in Fig. 14).



Based on our data, a simple regression is proposed to estimate $D^{\mathrm{u}}$ from the liquid water content and the dry snow porosity:

$$D^{\mathrm{u}}(\phi,\theta_w) = \begin{cases} D_{\mathrm{SC}} \times \left(1 - \frac{\theta_w}{\phi}\right)^3 = D_{\mathrm{SC}} \times \left(1 - \frac{\theta_w}{1-\rho/\rho_i}\right)^3 & \theta_w \leq \theta_w^{\mathrm{CP}} \\ 0 & \theta_w > \theta_w^{\mathrm{CP}} \end{cases} \tag{16}$$

$$D_{\mathrm{SC}} = D_v(3\phi - 1)/2 \tag{17}$$

$$\theta_w^{\mathrm{CP}} = \phi - 0.17 \tag{18}$$

with $D_{\mathrm{SC}}$ the self consistent estimate of effective vapor diffusivity as used in Calonne et al. (2014b). Estimates with this proposed regression are shown in Fig. 13 by the black line. This regression is a first suggestion that could be improved by refining the effect of the density, as it predicts too high values for the denser snow samples compared to our simulations and too low values for the lighter snow samples (left panel of Fig. 13). The values of $\theta_w^{\mathrm{CP}}$ could also be refined by calculating $D_{\mathrm{u}}^{\mathrm{eff}}$ for more values of saturation.

### 3.2.4   Sum up on the effective properties

The analysis of the effective properties based on the images obtained from the drainage simulations showed interesting results. As already mentioned, it should be kept in mind that many of these images are likely far from natural snow microstructure as the ice structure remains unchanged in the presence of liquid water in the simulations. Yet, they offer new insights on how the liquid water distribution in snow could influence the effective transport properties, for a variety of microstructures. It also enables a comparison between the current parameterizations of the transport properties, and suggestions of new estimates. Especially, we showed that the classic VGM parameterization for the relative water permeability and the unsaturated hydraulic conductivity reproduces well our simulated data and seems thus a good choice for both properties. For the effective unsaturated vapor diffusivity and the effective unsaturated thermal conductivity, we proposed new regressions, which, for the conductivity, differs strongly from some of the current parameterizations used in snow modeling. Finally, as all the above effective properties were only computed on images from drainage simulations only, it could be interesting to compare those estimates to the imbibition case.

## 4   Conclusion

In this study, a quasi-static model was used to simulate the distribution of liquid water in the pore space of snow for various water contents. Liquid water is gradually introduced and then removed by capillarity during wetting (imbibition) and drying (drainage) simulations. Such simulations were applied on a set of 34 3D tomography images of dry snow presenting various microstructures. Unlike natural snow, the ice matrix is fixed and does not evolve with the liquid water in the simulations. This work constitutes an exploratory numerical work (i) to study the water retention curves and (ii) the effective transport properties of wet snow and how they are influenced by the water distribution at the pore scale. Both points are critical to better understand and model water flow, heat and vapor transport in wet snow.



From each initial image of dry snow, two series of 3D images were produced and correspond to different stages of the sim-
ulations of imbibition or drainage, which were used to derive the corresponding WRCs. We confirmed the hysteresis of both
processes in snow and highlighted the dependency of the WRC on the microstructural features of snow, such as the pore size
and the density. We then compared our results to the model of van Genuchten (1980), widely-used to model the WRCs of
porous materials. For that, the model was fitted to our simulated WRCs by adjusting the model parameters $\alpha_{\mathrm{VG}}$, $n_{\mathrm{VG}}$ and $\theta_w^r$,
$\theta_w^s$ being set to the porosity value. Comparing with previous estimates of those parameters, we found that the latest regressions
from Yamaguchi et al. (2012) provide the closest values to our fitted parameters, with overall fair agreement for $\alpha_{\mathrm{VG}}$ for both
imbibition and drainage, but poor agreement for $n_{\mathrm{VG}}$ and $\theta_w^r$. New regressions and estimates of the parameters of the model
of van Genuchten (1980) were proposed for both imbibition and drainage processes, fitting our numerical results. They should
contribute to a more generalized use of this model to predict the WRCs of snow, as needed for water flow modeling. Dedicated
investigations should however continue to refine those regressions, especially concerning the exact values of the saturated water
content $\theta_w^s$, which remains an open question.

The series of wet snow images simulating drainage were then used to estimate the effective transport properties of wet snow.
These properties were computed using the Geodict software, solving boundary problems resulting from the homogenization
process applied to the heat, vapor and water transport equations in wet snow (e.g., Leroux and Pomeroy, 2017; Moure et al.,
2023). Results allowed to describe the relationships of these effective properties with the water content as well as the dry snow
density. The computed properties were compared to commonly-used parameterizations. Especially, we showed that the relative
water permeability and the unsaturated hydraulic conductivity can be well reproduced by the VGM model (Mualem, 1976; van
Genuchten, 1980), combined with the estimates of intrinsic permeability for dry snow of Calonne et al. (2012). For the effec-
tive thermal conductivity of wet snow, large discrepancies were reported when comparing our computations to some current
estimates used in snow models, such as the Crocus model (Vionnet et al., 2012) or the model of Leroux and Pomeroy (2017).
A new regression depending on both snow density and water content was suggested. Finally, a regression was also proposed to
predict the water vapor diffusivity of wet snow, based on the dry snow diffusivity estimated with the self-consistent model, the
water content and the snow density. All the above results are a first step toward a better understanding of the influence of the
micro-scale distribution of liquid water on the hydraulic and physical properties of wet snow. Future studies should take into
account additional processes, neglected in this study, such as the transformation of the microstructure by phase changes and
the movement of water by gravity.



*Data availability.* The equations of the boundary problems that were solved to computed the effective wet snow properties are available in the Supplement. The Supplement also includes detailed presentations of the 34 images used in this study, with property tables, downward and upward views, and mean curvature histograms. Finally, the computed values of imbibition and drainage simulations on our snow samples, the resulting VG parameters, and the numerical estimations of conductivity, water vapor diffusivity and water permeability are also in the

Supplement.

*Author contributions.* FF, CG and NC proposed the study. FF and NC acquired and prepared the image dataset. NA, LB and CG conducted the imbibition and drainage numerical simulations. The analyses and interpretations were carried out by LB, NA, CG, NC and FF. LB and NA prepared the manuscript with contributions from all co-authors.

*Competing interests.* The contact author has declared that none of the authors have any competing interests.

*Acknowledgements.* The 3SR lab is part of the Labex Tec 21 (Investissements d'Avenir, grant ANR-11-LABX-0030). CNRM/CEN is part of Labex OSUG@2020 (Investissements d'Avenir, grant ANR-10-LABX-0056). This research has been supported by the Agence Nationale de la Recherche through the MiMESis-3D ANR project (ANR-19-CE01-0009). We thank the ESRF ID19 beamline and the tomographic service of the 3S-R laboratory, where the 3-D images were obtained. Lisa Bouvet's current position is funded by the European Research Council (ERC) under the European Union's Horizon 2020 research and innovation program (IVORI; grant no. 949516).



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
