# Peer review of "Simulating liquid water distribution at the pore scale in snow: water retention curves and effective transport properties"

_EGUsphere, 2025_

## Referee Comment (RC2)

General Comments

This paper introduces a novel approach for determining the water retention curve of snow with simulations of uCT tomography images. The paper also calculates the effect of liquid water on several effective transport properties. Overall, the scientific approach is novel, interesting, and addresses a significant knowledge gap. Specifically, there are few existing data on the hysteresis of the water retention curve of snow, how these curves (and hysteresis) depend on snow microstructure, and how the presence of liquid water affects other snow properties. I think the approach is very exciting and promising, but that the value of the work is undercut by the quality of the writing. The science unquestionably warrants publication, but the manuscript needs significant revision in the organization and writing before it is ready for publication. I suggest the manuscript be reviewed in detail by a native speaker for typographical/grammatical errors as well as continuity/flow.

Specific Comments

Introduction: The flow of topics is sometimes strange, which makes it difficult to follow the train of thought. Because of this, I feel the introduction fails to succinctly indicate what knowledge gap is being addressed, which undercuts the value of the work. The relevant information is there and mostly needs to be edited/reorganized. A few specific comments to the introduction and smaller typos throughout are noted in the Technical Comments below.

Introduction: I think you need to elaborate on Lines 70-71. You need to include more information on how the effective transport properties are currently estimated and why this is not sufficient and/or why new water retention curves will improve on the current methods. This is a knowledge gap you are filling, but it is not very clear how the new water retention curves will achieve this better compared to previous methods. This would also support your statement in Line 193, where you state that effective thermal conductivity of snow and unsaturated effective water vapor diffusivity have been estimated "very little for the case of wet snow".

Limitations: The main limitation and question surrounding this work is how much the results can be trusted, since there is no ground truth or independent measurements with which the simulations were verified. Because of this, there should be some discussion addressing the uncertainties, why we can consider the simulations accurate, and how the reader should interpret the results in order to help them to decide to use (or not use) the proposed parameterizations (instead of the parameterization by Yamaguchi2012 which is based on measurements). It is not immediately clear if the system used by Hu2018 (the citation used) is transferable to snow given the pore size distributions used in that work compared to some of the larger pores found with certain grain types. In general, it would be good to discuss the accuracy, errors, and biases of the model that should be considered when applying the model to snow from whatever system it was validated with.

Parameterizations: I recommend adding prediction intervals and standard deviations for the fits of n and alpha. These would be helpful for quantifying the uncertainty of the fits if they are used in the future.

Discussion: You need to discuss the fact that optically equivalent radius derived from the SSA is not the same as the radius used by Yamaguchi2012. I think it is a very important factor since the grain diameter can vary a lot between methods and definition (here, optically equivalent radius vs equivalent sphere radius).

Hysteresis: I would suggest adding the alpha ratio (alpha_wetting / alpha_drainage) as a quantification of hysteresis (e.g. Line 209) and compare this to the literature values (Adachi2020, Leroux2017). It would also be interesting to see if this ratio is dependent on grain type or another parameter/combination of parameters.

REV: There is some discussion/description of the effect of the REV. However, I was still confused by how the REV was selected (Line 175). Is it just the maximum size of the uCT scan by Calonne2012? Given the range of voxel side lengths from 2.5 to 10mm, compared with typical grain sizes of MF which can be up to 1-2mm, it would seem to me that there would be some relationship between the REV size and the snow microstructure/relevant length scale which would have an effect on the simulations. Could you provide some clarification on this?

R2: Is R2 the correct metric for such a nonlinear function? Or perhaps another metric like a mean absolute error which can provide an error in physical units? Or something else?

Mean curvature as proxy for pore size distribution: (Line 262) Could you cite work demonstrating that this is a good proxy? I understand that mean curvature and mean pore size are correlated but perhaps other microstructural parameters (e.g. thickness) would be more suitable? Could you elaborate on why this makes sense for draining but not for imbibition? Did you look at the relationship for imbibition?

Residual liquid water content: What was the rate of drainage in the drainage experiments and/or what pressure did you set to induce suction out of the snow? Is it possible that this had an effect on the amount of residual water that was trapped and therefore led to the higher residual liquid water content values compared to the literature?

n: (Line 252) I think you can remove the explanation of the fit error. As you state, this is not likely to account for the discrepancy and I think is overreaching what you can really say with the small differences in R2. I would also suggest adding the fact that the Yamaguchi2012 parameterization was based on their S-samples, which were sieved. As seen in Fig.4 of Yamaguchi2012, the parameterization of n also doesn't fit the N-sample data but does seem to fit your data and your value of 4.7.

Residual liquid water content: (Line 275) Can you really conclude that the separation is due to the grain type as opposed to density based on the presented data?

Saturated liquid water content: Why did you not use theta_s from the simulations? Or analyze theta_s as you did for theta_r in Figure 6?

Effective wet snow transport properties: (Line 355) Can you add some text addressing the "so what?" of these results like you do for the thermal conductivity? Currently, the results are just presented, and the conclusion is just that the model seems to match the data. The bigger picture or effect is missing.

Effective wet snow transport properties: Is it possible to include a comparison to current parameterizations/models for hydraulic conductivity and diffusivity as was done for thermal conductivity. The comparison for thermal conductivity really shows how different the results can be, and this large difference demonstrates the importance of the approach provided here.

Effective wet snow transport properties: (Line 388) As presented, your regression matches your simulations MUCH better than the other methods, which is partially due to the fact that you are fitting your own simulation data. Are you capturing any physical effects that lead to better results

that the other methods don't? And what limitations do we need to consider before simply adopting your regression?

Effective wet snow transport properties: (Line 415) What is causing this bias?

Effective wet snow transport properties: (Line 429): Can you provide some discussion about how different the results are if you use the imbibition simulations and why they are different?

Technical comments

Writing: Not having a space between paragraphs is confusing as some paragraph breaks are ambiguous since the document is full justified.

Line 2: It is somewhat unclear if the "its" refers to snow or liquid water. Suggest to rephrase.

Line 15: Grammar - I don't think "implies" can be used like this. Perhaps causes? Also, drastic changes of what? Microstructure?

Lines 17-24: This part of the Intro is confusing. Suggest rewording/reorganizing to make the topics in each sentence connect more clearly. E.g. Perhaps new paragraph at Line 20 and what does "to predict wet snow" mean?

Line 21: Please add citations for the flow types.

Line 24: Probably should cite SNOWPACK as well

Line 24-26: The sentence starts with "recently" but then you cite Daanen and Nieber from 2009. Perhaps just remove "recently".

Line 27: Grammar - "little" is not correct

Line 27: "effective wet snow properties" is not self-explanatory. Perhaps define or introduce the parameters you address in the paper.

Line 36-38: The way this definition of imbibition and drainage is written makes it seem like you always have to start or end at saturation, which I don't think is 100% correct or is, at a minimum, a bit confusing as written. Maybe consider rewording.

Line 41-43: Please add citations for the soil examples

Line 43-44: For completeness, I would add that there are a few studies that did measure imbibition: Coleou1999, Adachi2020, Lombardo2025 [Disclaimer – I am an author on this paper.]

Line 45: The brief discussion of Adachi2020's MRI method is a bit confusing since you seem to be transitioning to discuss hysteresis, but then go on to discuss how the VG parameters were fit in the other experiments. Since I think you should mention Adachi2020 in the comment above, perhaps you can just move that part and not discuss it here.

Lines 41-54: This paragraph generally needs rewriting. It is confusing and the message is unclear, likely because the topic changes several times and the transitions between the topics are not obvious/fluid.

Line 53: I think more precisely, Yamaguchit2025 provided a parameterization for sieved melt forms. There were also the "N-samples" which were natural snow samples, but they did not fit the parameterization.

Line 55: Grammar – "enable to capture" is incorrect and the commas are also incorrect.

Line 55: I think you need to discuss Hydraulic Conductivity before this paragraph since the "overall hydraulic behavior" is dependent on the not only the capillary forces described in the WRC but also the flow rate which is often done using the (saturated and relative) hydraulic conductivity, as you discuss later.

Line 65: Typo: literature

Line 67: I think Yamaguchi2025 is also a relevant citation here

Line 69: This sentence on models seems out of place with respect to the topic of this paragraph.

Line 78: Accounted "for" here

Line 76-79: I think you remove this sentence about wet snow metamorphism and just discuss this limitation later in the manuscript.

Line 94: Aren't the subscripts just the phase (ice, water, or air)? This sentence about the subscripts is confusing.

Line 102: Can you make a statement about whether the snow samples in Calonne2012 were natural, laboratory generated, sieved, etc.? Of course the information is in the original paper, but it would be easier for the reader if this was just briefly stated here.

Line 117-118: Add a citation for this

Line 119: Why did you choose 12 degrees? Cite?

Line 134: Typo: Ponds

Line 131-135: I don't think you need these sentences about the configuration. It is clear from the rest of the intro what imbibition and drainage are and the measurement of WRCs doesn't need to represent a situation that occurs in nature.

Line 161: Small formatting suggestion: in Eq 6, VG subscripts of alpha, n, and m look a bit weird to me. Perhaps increase their offset or decrease their size.

Line 167: I think this paper demonstrates nicely these relationships and how they are only roughly correct: van Lier and Pinheiro, 2018. Could be a nice citation to include.

Line 203: Not being particularly familiar with these tensors, I was wondering if the non-diagonal terms being negligible has a physical meaning and/or is standard. If it's important, perhaps provide a citation which contains this explanation so that a reader unfamiliar with this can inform themselves?

Line 213: It is unclear what this sentence is supposed to mean. It seems like you are just redescribing imbibition and drainage.

Line 222: I think you can't compare the shape when using liquid water content. If you want to compare the shape, you should plot saturation vs pressure as this normalizes the x-axis. I also don't really understand what is meant by "smoother inflections".

Line 228: It seems the point density for imbibition is higher than for drainage? i.e. there are more points on the imbibition curves in Fig 5. Is this true? That certainly would impact R2.

Line 241: Can you quantify this agreement?

Line 251: This makes sense since Yamaguchi2012 shows only drainage data – perhaps state this.

Line 283: I don't understand this sentence. As written, it seems like you are saying that the saturated liquid water content (LWC) can be up to 30% of the porosity. Do you mean that it can be 30% less than the porosity? i.e. Saturated LWC = 0.7xporosity?

Line 336: In order to make the statement in Line 336, I think you need to show the plots for other microstructural parameters such as ssa, curvature, etc.

Line 373: It is unclear if you mean Fig 11 a or b. Could you also postulate why there seems to be a density bias in your regression?

Line 396: This term is also the saturation, which might be better to use since saturation is used elsewhere in the paper already.

Line 400-401: For this statement, I think you need to include something about the close pore water content. As is, it doesn't make sense that the plot defines the domain of Du – as you state afterwards, it's theta_cp, so I would include that immediately in the sentence to make that clear.

Table 1: Can you rename the samples to something more intuitive? I understand these are probably the names in Calonne2012 but for the reader here that doesn't really matter (of course you can correlate them to the Calonne2012 paper). It would be easy to use the snow type as the name MF_l, MF_s, DH, etc. (or similar). This would allow the reader to immediately know what the snow properties are instead of having to memorize a random code since samples are grouped by grain type throughout the paper.

Table 2: Is there a unit issue in the fits for alpha? There are two orders of magnitude difference in the coefficient between Yamaguchi2012 and your data (10^4 vs 10^6).

Figure 5: It would be helpful to add the R2 values to the legends for each sample in Figure 5.

Figure 6: Something is weird with the caption formatting.

Figure 6: I think the (a)-plot as an overview of the data is nice, but could you also maybe replot only your data such that the reader can see the difference between imbibition and drainage for your data. Currently, the plot is too difficult to read to make out what is what.

Figure 7: Please add the grain type legend

Figure 8: This plot is very hard to read. Please improve the lines/marker differences to differentiate the datasets.

Figure 9a: Can you repeat this plot but with grain types and/or indicate grain types with the marker shape on the existing plot so that we can see how the grain type matches up?

Figure 13: It is really hard to tell that the solid line is blue. Just make it black since there are no other solid lines?

Figures 9b, 10b, 11b: Why are different saturation values (x coordinates) chosen for the different samples?

References

Adachi, S., Yamaguchi, S., Ozeki, T., and Kose, K.: Application of a magnetic resonance imaging method for nondestructive, three-dimensional, high-resolution measurement of the water content of wet snow samples, Frontiers in Earth Science, 8, https://doi.org/10.3389/feart.2020.00179, 2020.

Calonne, N., Geindreau, C., Flin, F., Morin, S., Lesaffre, B., Rolland du Roscoat, S., and Charrier, P.: 3-D image-based numerical computations of snow permeability: links to specific surface area, density, and microstructural anisotropy, The Cryosphere, 6, 939–951, https://doi.org/10.5194/tc-6-939-2012, 2012.

Coleou, C., Xu, K., Lesaffre, B., and Brzoska, J.-B.: Capillary rise in snow, Hydrological Processes, 13, 1721–1732, https://doi.org/10.1002/(SICI)1099-1085(199909)13:12/13<1721::AID-HYP852>3.0.CO;2-D, 1999.

Lombardo, M., Fees, A., Kaestner, A., van Herwijnen, A., Schweizer, J., and Lehmann, P.: Quantification of capillary rise dynamics in snow using neutron radiography, EGUsphere [preprint], https://doi.org/10.5194/egusphere-2025-304, 2025b.

van Lier, Q. d. J. and Pinheiro, E. A. R.: An Alert Regarding a Common Misinterpretation of the Van Genuchten $\alpha$ Parameter, Revista Brasileira de Ciencia do Solo, 42, e0170 343, https://doi.org/10.1590/18069657rbcs20170343, 2018.

Yamaguchi, S., Watanabe, K., Katsushima, T., Sato, A., and Kumakura, T.: Dependence of the water retention curve of snow on snow characteristics, Annals of Glaciology, 53, 6–12, https://doi.org/10.3189/2012AoG61A001, 2012.

Yamaguchi, S., Adachi, S., and Sunako, S.: A novel method to visualize liquid distribution in snow: superimposition of MRI and X-ray CT images, Annals of Glaciology, 65, e11, https://doi.org/10.1017/aog.2023.77, 2025.

---

## Author Comment (AC1)

**Authors' reply to referee comments RC1 of the paper egusphere-2025-2903 entitled "Simulating liquid water distribution at the pore scale in snow: water retention curves and effective transport properties" by Bouvet et al.**

We thank very much Reviewer 1 for the comments. Please find below our point-by-point replies in blue color.

The physical properties of wet snow are less well understood than those of dry snow, with fewer measurement data and experimental results available. In this study, the authors reported the relationship between the microstructure of snow and the physical properties of wet snow based on simulations. They also compare the simulation results with existing experimental results and propose new equations for estimating these physical parameters. Their simulation approach is unique, and there

10 is no doubt that their method is useful in situations where experimental results are lacking in the wet snow field. For these reasons, this paper makes a significant scientific contribution to the field of wet snow science and is worthy of publication in TC. The paper is well organized and easy for readers to understand.

On the other hand, the discussion in this paper is based on simulation results. Simulation results depend on conditions and parameters, so it is necessary to verify the accuracy of the simulation results based on experimental or observational results.

15 Until the accuracy of the simulation is guaranteed, it is not possible to evaluate the true scientific contribution of the research. Therefore, I believe that it is necessary to add a discussion on the accuracy of the simulation to the paper before it is accepted for publication. I will list a few points I have noticed, including these points.

**Major point**

20

As shown in Equation (5), the pressure head of a liquid depends on the radius of the pore, and this is the basic equation used in the simulations in this study. In order to use Equation (5), it is necessary to calculate the radius of the pores in the snow. However, since the pores in the snow are interconnected, it is necessary to separate them in some way in order to determine their radii. Therefore, the distribution of radii may vary depending on the method used to separate the pores, which in turn

25 may affect the simulation results. For this reason, the authors should add an explanation of the method used to separate the pores and discuss the impact of the pores separation method on the simulation results. Finally, based on comparisons of the measurement results of water retention curve, they need to discuss the accuracy of the simulation results, including the best method for separating each pore.

30 The model implemented in the software SatuDict is the Pore Morphology Method (PMM) (e.g., Hilpert and Miller, 2001; Silin and Patzek, 2006; Ahrenholz et al., 2008; Schulz et al., 2015; Berg et al., 2016; Liu et al., 2022; Arnold et al., 2023). The PMM does not require to separate the pores as in a Pore Network Model (PNM) (Vogel et al., 2005; Joekar-Niasar and Hassanizadeh, 2012; Xiong et al., 2016). The PMM uses a sphere with a radius $r$ as a probe to detect the pore space that is accessible by the non-wetting phase (NWP), here the air. This radius is computed from the Young–Laplace equation:

35 $r = 2\gamma\cos(\psi)/p_c$ where $p_c$ is the capillary pressure, $\gamma$ is the interfacial tension and $\psi$ is the contact angle. Morphological operations, namely, erosion and /or dilation are used in the PMM (Hilpert and Miller, 2001). The algorithm of the PMM can be decomposed into several steps as follows:

– In drainage condition, the porous medium is initially saturated with the wetting phase (WP) ($p_c = 0$). The invading NWP is connected to the inlet, which is the NWP reservoir, and the WP can escape through to the outlet, the WP reservoir.

40 (i) Then, $p_c$ is increased incrementally, i.e. $r$ is decreased incrementally. The solid phase is first dilated by a sphere with radius $r$ (ii) All the pores connected to the NWP reservoir are labelled as NWP. (iii) The NWP is then dilated with the same sphere with radius $r$. The remaining pores are filled with the WP. The saturation can be then calculated. (iv) All the pores filled by the WP disconnected from the WP reservoir are considered as WP residual, and are no more considered in the next steps. All these steps are repeated by increasing the value of the pressure, i.e. decreasing the value of $r$. In the

45 present case, the radius $r$ has been decreased gradually with a step of 2 pixel size.

– In imbibition condition, the porous medium is initially saturated with the NWP. The invading WP is connected to the inlet, which is the WP reservoir, and the NWP can escape through the outlet, the NWP reservoir. (i) Then, $p_c$ is decreased incrementally, i.e. $r$ is increased incrementally. The solid phase is first dilated by a sphere with radius $r$. (ii) The NWP

is then dilated with the same sphere with radius $r$. (iii) All the pores connected to the WP are now labelled as WP. The remaining pores are NWP. The saturation can then be calculated. (iv) All the pores filled by the NWP disconnected from the NWP reservoir are considered as NWP residual, and are no longer considered in the next steps. As for the drainage condition, all these steps are repeated for the next value of the pressure, i.e. the next value of $r$. In the present case, the radius $r$ has been increased gradually with a step of 2 pixel size.

The PMM has been used to simulate the quasi-static Water Retention Curves (WRC) of porous media (glass beads, sands,..) in drainage or imbibition conditions (Hilpert and Miller, 2001; Liu et al., 2022). In imbibition conditions, if step (iv) is ignored, the PMM also allows for the computation of Mercury Injection Capillary Pressure curves (MICP), which are a commonly used technique for measuring porosity and pore throat size distribution, for instance (Hilpert and Miller, 2001; Berg et al., 2016). As underlined in Hilpert and Miller (2001), the accuracy of the PMM may depend on the resolution and size of the 3D images and of the definition of the structural element. Finally, it is worth mentioning that the boundary conditions applied on the four sides of the 3D images that are not connected to the NWP or WP reservoirs may also play a role in the WRC curve simulations, even if this point has not been discussed in the literature, to the best of our knowledge. In Vogel et al. (2005), impervious boundary conditions are applied, whereas other conditions, such as symmetry, displaced fluid outlet, or invading fluid inlet, can also be applied (Berg et al., 2016). These boundary conditions may lead to different values of the NWP (air) residuals during the imbibition process, since these residuals can be trapped or not at the boundary. Depending on the boundary conditions, the maximum water saturation ($\theta_w^s$) can range from 45% to 90% of the porosity. Despite such large differences, the values of $\alpha_{\mathrm{vg}}$ and $n_{\mathrm{vg}}$ in the van Genuchten (VG) model (van Genuchten, 1980) remain almost constant, with an effect only visible for $\alpha_{\mathrm{vg}}$, with differences around 10% of the value compared to the case $\theta_w^s = \phi$. This was also reported in the experiments of Likos et al. (2014) and Farooq et al. (2024), which showed that having $\theta_w^s$ smaller than $\phi$ generally implies greater $\alpha_{\mathrm{vg}}$ values, but has no significant impact on the $n_{\mathrm{vg}}$ values. The impact of the boundary conditions is less pronounced in drainage conditions, since the water residuals are mainly located at the junction between grains

In the present study, the PMM implemented in the SatudDict software was used to compute the WRCs of the 34 snow samples. We computed (i) a primary imbibition curve assuming that there is no air (NWP) residuals as in MICP experiments, thus $\theta_w^s = \phi$ in van Genuchten (VG) model, and then (ii) a primary drainage curve until to reach the water (WP) residuals ($\theta_w^r$). In both cases, symmetric boundary conditions are applied on the four sides of the volumes.

Despite some limitations, the PMM appears as an appealing method to investigate two-phase flows in porous media in quasi-static conditions. Indeed, this method requires less computational time and computer memory compared to other two–phase flow simulation methods (lattice-Boltzmann methods, volume of fluids methods, phase fields models...) (Vogel et al., 2005; Ahrenholz et al., 2008; Bhatta et al., 2024) and provides similar results.

**In the revised version**, the description of the pore morphology method was substantially modified and includes all the above considerations (see introduction and Section 2.3)

**Specific points**

**L27:** I disagree because some studies evaluate results based on observations (e.g., runoff from snow cover) or comparisons with laboratory experiments. If they insist on their claim, they should point out the problems with past studies in more detail.

This sentence has been rephrased to specify the literature gap of estimating the hydraulic properties of snow for water flow, at the pore scale. The introduction of the revised version of the manuscript was substantially modified and provides now a more comprehensive state-of-the-art on studies and estimations of the effective properties of wet snow.

**L117:** If gravity is ignored, isn't it impossible to calculate the water retention curve? This is because gravity is included in equation (5).

The pore morphology method is based on the use of the Young-Laplace equation, not Jurin's law. We apologise for this error in the original version of the manuscript. We modified Equation (5) with the Young-Laplace equation. It represents the difference of pressure at a curved interface, and is a function of the mean curvature and the surface tension only. Here, gravity and viscous forces are neglected compared to capillary forces. At the pore scale, surface forces usually play a

much important role than gravity. For air and water, the Bond number, which measures the ratio between gravitational force and surface tension force, is defined as: $\text{Bo} = (\rho_w - \rho_a)gl^2/\gamma$ where $\rho_w$ and $\rho_a$ are the water and the air density respectively, $g$ is the gravity, $\gamma$ the surface tension, and $l$ a microscopic length as the pore size. This dimensionless number varies between $10^{-3}$ and $10^{-1}$ if the pore size $l$ varies between $10^{-4}$ and $10^{-3}$ m respectively, as in snow. These estimations show that gravitational forces are negligible at the pore scale in comparison to capillary forces. Similarly, it can be shown (Auriault, 1987; Auriault et al., 2009) that within a porous media, the viscous stress ($\sigma_v$) at the pore scale is negligible in comparison to the fluid pressure ($p$) which is the order of the capillary pressure ($p_c$): $p \approx p_c \approx (L/l)\sigma_v$ where $L$ is a macroscopic length, i.e. the size of the snowpack. If we assume that the pore size is $l = 10^{-3}$ m and $L = 0.1$ m, the capillary pressure is around 100 times larger than the viscous stress. This part has been included in the revised version of the manuscript (see section 2.3)

**L126:** Since cos12 is 0.97 and cos0 is 1, I don't think this will have a significant impact on the result. However, I think it would normally use 0 degrees, so what is the basis for using 12 degrees? Please provide the reference.

The contact angle between liquid water and ice close to the fusion temperature 0°C has been measured at 12° in Knight (1967). The reference has been added to the manuscript.

**L131:** As shown in L117, this model ignores the effects of gravity, so the simulation may not accurately reproduce the actual drainage process. How do they feel about this point?

As it has been shown previously, the viscous stress and the gravitational forces are negligible at the pore scale. The above estimations are valid for both drainage and imbibition conditions. So the WRC of snow can be obtained by considering only capillary forces at the pore scale. If the porous media present very large pores, typically the order of 1 cm, the Bond number will be the order of one, meaning that gravitational forces must be considered.

**L249:** Because nVG is said to depend on the distribution of pore sizes, the results of this study may be due to the method used to separate the pores. Therefore, this possibility needs to be discussed.

The pore morphology method (PMM) is used to compute the WRCs. The PMM method does not require to separate the pores, as in the case of a Pore Network Model. As mentioned above, a sphere of radius $r$ is used as probe. In drainage or imbibition condition, the radius is decreased or increased with a step of 2 pixel size (see the detailed description of the method provided above).

**L321:** This argument is meaningless. This is because the data used for comparison are simulation results, so it is only natural that the results of this study are the most appropriate results.

We agree that the phrasing l.321 can be misleading. We modified this part accordingly to separate the evaluation of our regressions on our dataset, and the comparison with the other regressions of the literature.

**L353** : As shown in Table 2, the parameter settings used in this study differ from those of Yamaguchi et al. (2012). For example, the $\alpha$VG value in Yamaguchi et al. (2012) is two orders of magnitude larger than the value used in this study.

On the other hand, as shown in Figure 10, the calculated unsaturated hydraulic conductivity using the parameter settings in this study shows relatively good agreement with the results of Yamaguchi et al. (2012).

This result may suggest that the parameterization of the VG model may not have a significant influence on the unsaturated hydraulic conductivity.

I propose adding further discussion on which parameterization of $\alpha$VG and nVG is more important in determining the unsaturated hydraulic conductivity. This discussion is considered useful for determining which parameterization requires higher accuracy.

There is two order of magnitude difference in the coefficient of the regressions for $\alpha_{\text{vg}}$ between Yamaguchi et al. (2012) and our regression because the exponent value is smaller in our case (-0.61 instead of -0.98, see Table 2). To avoid the fact that a large range of values for the two coefficients can describe almost the same WRC, we chose to simplify our new regression by keeping the multiplicator coefficient of Yamaguchi et al. (2012) ($4.4 \times 10^4$) and only fitting the exponent coefficient. This is motivated by the fact that the first coefficient has more impact on the small $\rho/d$ values, which correspond to most of the measured values of Yamaguchi et al. (2012), and the exponent has more impact on the

larger values of $\rho/d$, which correspond to most of our values from the numerical estimates. With this change, we obtain a new regression of $\alpha_{\mathrm{vg}}$ with an exponent of 0.95 ±0.003 that is not so different from the one of Yamaguchi et al. (2012) of 0.98. This has been modified in the revised version in Sec. 3.1.3.

[Figure]

**Figure 1.** a) $\alpha_{\mathrm{vg}}$, b) $n_{\mathrm{vg}}$ and c) $\theta_w^r$ parameters of the VG model as a function of $\rho/d$ or $\rho$ for imbibition and drainage. The regressions of Yamaguchi et al. (2012) are shown by black lines, the values used to deduct those regressions are shown by black disks ("S-samples", composed of refrozen MF - see Yamaguchi et al. (2012)). The additional drainage measurements from Yamaguchi et al. (2012) are shown by circles (MF samples) and stars (RG samples). Measurements from Adachi et al. (2020) are shown by empty gray markers. Measurements from Lombardo et al. (2025) and Katsushima et al. (2013) are shown by empty orange triangles and blue circles. From this work, parameters from imbibition and drainage simulations are shown by colored disks and triangles, respectively, the colors showing the snow types. The proposed regressions based on our simulated data are shown by dashed and dotted lines (see Table 2).

[Figure]

**Figure 2.** $n_{vg}$ parameter as a function of the interquartile range $IQR_{MC}$ obtained for the mean curvature distribution computed on each 3D dry snow image. Regressions for imbibition and drainage are shown with dotted and dashed lines.

Regarding the unsaturated hydraulic conductivity, as the log scale makes it somewhat hard to see in Fig. 10 (corresponding to Fig. 11 in the revised version), we now provide the mean absolute error (MAE) values to compare quantitatively the VGM model using the estimates of the shape parameters from this study and from Yamaguchi et al. (2012) against the estimated regressions based our numerical results (Table 2). For the melt forms samples, a slight improvement is found using our regressions, with MAE values around 10% for Yamaguchi et al. (2012) and around 9% for our model. This similarities can be related to the fact that, on one hand, the regression of Yamaguchi et al. (2012) provides slightly better estimates of $\alpha_{vg}$ for this snow type, compared to our regression (Fig. 1.a); on the other hand, $n_{vg}$ values are better estimates from our regression (Fig. 1.c and Fig. 2). For the other snow types, the VGM model using our shape parameter estimates provides better predictions of unsaturated hydraulic conductivity, showing MAE values around 17% compared to the VGM model using the estimates of Yamaguchi et al. (2012), showing MAEs around 22%. Indeed, for all the snow types excluding melt forms, both shape parameters $\alpha_{vg}$ and $n_{vg}$ are overall better estimated using our regression (as optimized to best match our numerical simulations). We recently improved the $n_{vg}$ parameterization by using the interquartile range (IQR) of the distribution, which is more suited to the shape of our distributions than the previously used standard deviation (see Fig. 2).

Hence, our results tend to show that both parameters $\alpha_{vg}$ and $n_{vg}$ have an impact on the quality of the prediction of the hydraulic conductivity. However, in agreement with the reviewer comment, we point out that the VGM model based on the shape parameter estimates of Yamaguchi et al. (2012), which only required the knowledge of density and grain size, allows overall for fair estimates of the unsaturated hydraulic conductivity, with MAEs ranging from 10 to 22%, when compared to the simulations on our five samples. All the above considerations were included in the revised manuscript (Sec. 3.2.1).

**References**

165    Adachi, S., Yamaguchi, S., Ozeki, T., and Kose, K.: Application of a magnetic resonance imaging method for nondestructive, three-dimensional, high-resolution measurement of the water content of wet snow samples, Frontiers in Earth Science, 8, https://doi.org/10.3389/feart.2020.00179, 2020.

Ahrenholz, B., Tölke, J., Lehmann, P., Peters, A., Kaestner, A., Krafczyk, M., and Durner, W.: Prediction of capillary hysteresis in a porous material using lattice-Boltzmann methods and comparison to experimental data and a morphological pore network model, Advances in
170    Water Resources, 31, 1151–1173, https://doi.org/10.1016/j.advwatres.2008.03.009, 2008.

Arnold, P., Dragovits, M., Linden, S., Hinz, C., and Ott, H.: Forced imbibition and uncertainty modeling using the morphological method, Advances in Water Resources, 172, 104 381, https://doi.org/https://doi.org/10.1016/j.advwatres.2023.104381, 2023.

Auriault, J.-L.: Non saturated deformable porous media: quasi-statics, Transport in porous media, 2, 45–64, https://doi.org/10.1007/BF00208536, 1987.

175    Auriault, J.-L., Boutin, C., and Geindreau., C.: Homogenization of coupled phenomena in heterogenous media, Wiley-ISTE, London, 2009.

Berg, S., Rücker, M., Ott, H., Georgiadis, A., van der Linde, H., Enzmann, F., Kersten, M., Armstrong, R., de With, S., Becker, J., and Wiegmann, A.: Connected pathway relative permeability from pore-scale imaging of imbibition, Advances in Water Resources, 90, 24–35, https://doi.org/https://doi.org/10.1016/j.advwatres.2016.01.010, 2016.

Bhatta, N., Gautam, S., Farhan, N. M., Tafreshi, H. V., and Pourdeyhimi, B.: Accuracy of the Pore Morphology Method
180    in Modeling Fluid Saturation in 3D Fibrous Domains, Industrial & Engineering Chemistry Research, 63, 18 147–18 159, https://doi.org/10.1021/acs.iecr.4c02939, 2024.

Farooq, U., Gorczewska-Langner, W., and Szymkiewicz, A.: Water retention curves of sandy soils obtained from direct measurements, particle size distribution, and infiltration experiments, Vadose Zone Journal, 23, e20 364, https://doi.org/10.1002/vzj2.20364, 2024.

Hilpert, M. and Miller, C. T.: Pore-morphology-based simulation of drainage in totally wetting porous media, Advances in Water Resources,
185    24, 243–255, https://doi.org/10.1016/S0309-1708(00)00056-7, pore Scale Modeling, 2001.

Joekar-Niasar, V. and Hassanizadeh, S. M.: Analysis of Fundamentals of Two-Phase Flow in Porous Media Using Dynamic Pore-Network Models: A Review, Critical Reviews in Environmental Science and Technology, 42, 1895–1976, https://doi.org/10.1080/10643389.2011.574101, 2012.

Katsushima, T., Yamaguchi, S., Kumakura, T., and Sato, A.: Experimental analysis of preferential flow in dry snowpack, Cold Regions
190    Science and Technology, 85, 206–216, https://doi.org/10.1016/j.coldregions.2012.09.012, 2013.

Knight, C. A.: The contact angle of water on ice, Journal of Colloid and Interface Science, 25, 280–284, https://doi.org/10.1016/0021-9797(67)90031-8, 1967.

Likos, W. J., Lu, N., and Godt, J. W.: Hysteresis and Uncertainty in Soil Water-Retention Curve Parameters, Journal of Geotechnical and Geoenvironmental Engineering, 140, 04013 050, https://doi.org/10.1061/(ASCE)GT.1943-5606.0001071, 2014.

195    Liu, X., Zhou, A., long Shen, S., and Li, J.: Modeling drainage in porous media considering locally variable contact angle based on pore morphology method, Journal of Hydrology, 612, 128 157, https://doi.org/https://doi.org/10.1016/j.jhydrol.2022.128157, 2022.

Lombardo, M., Fees, A., Kaestner, A., van Herwijnen, A., Schweizer, J., and Lehmann, P.: Quantification of capillary rise dynamics in snow using neutron radiography, EGUsphere, 2025, 1–36, https://doi.org/10.5194/egusphere-2025-304, 2025.

Schulz, V. P., Wargo, E. A., and Kumbur, E. C.: Pore-Morphology-Based Simulation of Drainage in Porous Media Featuring a Locally
200    Variable Contact Angle, Transport in Porous Media, 107, 13–25, https://doi.org/10.1007/s11242-014-0422-4, 2015.

Silin, D. and Patzek, T.: Pore space morphology analysis using maximal inscribed spheres, Physica A: Statistical Mechanics and its Applications, 371, 336–360, https://doi.org/https://doi.org/10.1016/j.physa.2006.04.048, 2006.

van Genuchten, M. T.: A closed-form equation for predicting the hydraulic conductivity of unsaturated soils, Soil Science Society of America Journal, 44, 892–898, https://doi.org/10.2136/sssaj1980.03615995004400050002x, 1980.

205    Vogel, H.-J., Tölke, J., Schulz, V. P., Krafczyk, M., and Roth, K.: Comparison of a Lattice-Boltzmann Model, a Full-Morphology Model, and a Pore Network Model for Determining Capillary Pressure-Saturation Relationships, Vadose Zone Journal, 4, 380–388, https://doi.org/https://doi.org/10.2136/vzj2004.0114, 2005.

Xiong, Q., Baychev, T. G., and Jivkov, A. P.: Review of pore network modelling of porous media: Experimental characterisations, network constructions and applications to reactive transport, Journal of Contaminant Hydrology, 192, 101–117,
210    https://doi.org/https://doi.org/10.1016/j.jconhyd.2016.07.002, 2016.

Yamaguchi, S., Watanabe, K., Katsushima, T., Sato, A., and Kumakura, T.: Dependence of the water retention curve of snow on snow characteristics, Annals of Glaciology, 53, 6–12, https://doi.org/10.3189/2012AoG61A001, 2012.

---

## Author Comment (AC2)

**Authors' reply to referee comments RC2 of the paper egusphere-2025-2903 entitled "Simulating liquid water distribution at the pore scale in snow: water retention curves and effective transport properties" by Bouvet et al.**

We thank very much the reviewer Michael Lombardo for the comments. Please find below our point-by-point replies in blue color.

**General Comments**

This paper introduces a novel approach for determining the water retention curve of snow with simulations of uCT tomography images. The paper also calculates the effect of liquid water on several effective transport properties. Overall, the scientific approach is novel, interesting, and addresses a significant knowledge gap. Specifically, there are few existing data on the hysteresis of the water retention curve of snow, how these curves (and hysteresis) depend on snow microstructure, and how the presence of liquid water affects other snow properties. I think the approach is very exciting and promising, but that the value of the work is undercut by the quality of the writing. The science unquestionably warrants publication, but the manuscript needs significant revision in the organization and writing before it is ready for publication. I suggest the manuscript be reviewed in detail by a native speaker for typographical/grammatical errors as well as continuity/flow.

**Specific Comments**

The flow of topics is sometimes strange, which makes it difficult to follow the train of thought. Because of this, I feel the introduction fails to succinctly indicate what knowledge gap is being addressed, which undercuts the value of the work. The relevant information is there and mostly needs to be edited/reorganized. A few specific comments to the introduction and smaller typos throughout are noted in the Technical Comments below.

We agree with the reviewer and edited the introduction substantially. The revised version provide a clearer overview of the state of the art in wet snow studies and property estimation. We enriched the description of the current limitations and stated more clearly our contribution.

I think you need to elaborate on Lines 70-71. You need to include more information on how the effective transport properties are currently estimated and why this is not sufficient and/or why new water retention curves will improve on the current methods. This is a knowledge gap you are filling, but it is not very clear how the new water retention curves will achieve this better compared to previous methods. This would also support your statement in Line 193, where you state that effective thermal conductivity of snow and unsaturated effective water vapor diffusivity have been estimated "very little for the case of wet snow".

The introduction of the revised manuscript was significantly modified to improve the state of the art on the estimation of the WRCs and effective properties of wet snow in current modeling. Especially, we highlight the following gaps: i) current regressions to estimate the shape parameters of the VG and MVG model for snow, used to predict the WRCs and the unsaturated hydraulic conductivity, were developed based on few drainage experiments using mostly large dense melt forms, so there is a gap in the knowledge of the WRCs and unsaturated hydraulic conductivity for the whole range of snow, ii) current regressions of the unsaturated thermal conductivity of snow are either extrapolations of thermal conductivity regressions for dry snow, or based on simple ponderation of the thermal conductivity of ice and of liquid water, neglecting the effect of the microstructure (which was shown to be important for the case of dry snow) or the fact that water conducts four times less than ice, and iii) effective diffusivity was simply not investigated for the wet snow case. To add more context, we added references for studies about water vapor diffusivity for unsaturated soils.

The main limitation and question surrounding this work is how much the results can be trusted, since there is no ground truth or independent measurements with which the simulations were verified. Because of this, there should be some discussion addressing the uncertainties, why we can consider the simulations accurate, and how the reader should interpret the results in order to help them to decide to use (or not use) the proposed parameterizations (instead of the parameterization by Yamaguchi2012 which is based on measurements). It is not immediately clear if the system used by Hu2018 (the citation used) is transferable

50    to snow given the pore size distributions used in that work compared to some of the larger pores found with certain grain types. In general, it would be good to discuss the accuracy, errors, and biases of the model that should be considered when applying the model to snow from whatever system it was validated with.

    We agree with the reviewer and we dedicated a new section in the revised manuscript ("4. Main limitations") to present the
55    main limitations of the study. The new section synthesizes the following points:

– In the present work, the Pore Morphology Method (PMM) has been used. This method is valid in quasi static regime, and when capillary forces dominate in comparison to gravity and viscous forces. Dynamic effects than can occur in practice can not be captured as in two phase flow simulations (Vogel et al., 2005; Ahrenholz et al., 2008; Bhatta et al., 2024; Prodanović and Bryant, 2006; Jettestuen et al., 2013) or using a Pore Network Model (PNM) (Vogel et al., 2005; Joekar-
60    Niasar and Hassanizadeh, 2012; Xiong et al., 2016). Despite such limitation, the PMM provides good estimations of the WRCs for porous media whose wetting phase shows generally spherical menisci, which is the case for snow (Hilpert and Miller, 2001; Vogel et al., 2005).

– Regarding the pore-scale simulations with Geodict, to get an idea of how well the simulation can be compared to reality, we would like to point out this video presented at the GeoDict User Meeting 2021, which shows the validation of the
65    pore morphology model in advances in two-phase and single-phase flow simulations : https://www.youtube.com/watch?v=WMSorqm3B_g(from 11:000 to 14:30). However, the boundary conditions applied on the four sides of the 3D images not linked to the WP or NWP reservoir may play an important role on the residuals after drainage or imbibition. While little influence was reported for the case of drainage, the amount of residual air (or entrapped air) during imbibition may vary significantly depending on the chosen boundary conditions. This point which concerns all the methods (PMM,
70    PNM, two-phases flow simulations) to describe two-phase flows has been little discussed in the literature, to the best of our knowledge, except in Galindo-Torres et al. (2016) and Zhang et al. (2025) in the case of lattice Boltzmann simulations. For our snow samples, preliminary tests showed that the maximum water saturation ($\theta_w^s$) ranges from 45% to 90% of the porosity, depending on the applied boundary conditions (symmetry, wall or displaced fluid outled). More generally, knowledge on the residual air in snow seems limited. Estimates based on measurements remain an
75    experimental challenge and show large differences (Yamaguchi et al., 2012; Katsushima et al., 2013; Adachi et al., 2020). Further studies would be required to validate the proposed approach through refined comparisons on experimental imbibition data coupling measurements of the microstructure by X-ray tomography and of the liquid water content by neutron radiography (see e.g., Tengattini et al., 2020). At this stage, given the uncertainty, the imbibition curves were simulated assuming that there is no air residuals, as in the Mercury Injection Capillary Pressure (MICP) experiments
80    (e.g. Hilpert and Miller, 2001; Berg et al., 2016), thus $\theta_w^s = \phi$. We evaluated the impact of this simplification on the shape parameters of the VG model. Even with $\theta_w^s$ as low as $0.6\phi$, the shape parameters remain almost constant, with an effect only visible for $\alpha_{vg}$, with differences around 10% of the value compared to the case $\theta_w^s = \phi$. This was also reported in the experiments of Likos et al. (2014) and Farooq et al. (2024), which showed that having $\theta_w^s$ smaller than $\phi$ generally implies greater $\alpha_{vg}$ values, but has no significant impact on the $n_{vg}$ values.

85    – Since the PMM is applied on 3D images, uncertainties can arise from the size and resolution of the images under consideration. The effects of both parameters on the results are available in Hilpert and Miller (2001) and Vogel et al. (2005), and are assumed to be transferable to snow. In the present study, we checked that our snow images correspond to representative elementary volumes. Incertainties remain regarding the size of the melt forms images, for which the maximum available size was taken, but still present a limited number of heterogeneities (see Fig. 1).

90    – Our simulated WRCs were compared to WRCs measured during experiments of drainage and or imbibition. Such a comparison is not straightforward, as, in the experiments, the snow microstructure can evolve rapidly when in contact with liquid water, whereas, in the simulations, the ice skeleton is fixed and defined by the provided tomography image, always remaining in its initial stage. The comparison simulation-experiment was mainly done through the comparison of the shape parameter of the VG model derived from the WRCs. Experimental estimates remain, however, limited,
95    often focusing either on imbibition or on drainage, or studying only a small range of snow types (Yamaguchi et al.,

[Figure]

**Figure 1.** REV evaluation on the van Genuchten parameters for 5 snow samples.

2010, 2012; Katsushima et al., 2013; Adachi et al., 2012; Lombardo et al., 2025). While estimates of $\alpha_{\mathrm{vg}}$ are rather consistent between all the measurements and our simulations for both imbibition and drainage (Fig. 2.a and b.), estimates of $n_{\mathrm{vg}}$ differ significantly (Fig. 2.c and d.). Hence, it is difficult to conclude on the evaluation of our simulations. Again, dedicated studies would be required to provide further experimental data.

100   – Finally, the uncertainties of the WRCs simulations are not necessarily transferred to the estimates of the effective transport properties of wet snow. The simulations provide the 3D skeleton of the air, ice, and liquid water, for which the distribution of each phase in space can contain errors, as discussed above. However, only the unsaturated hydraulic conductivity depends at the first order on both the volume fraction and the 3D distribution of the phases. Its predictions with the VGM model inherit from the uncertainties on the estimates of the VG parameters used to model the WRCs. In
105   contrast, the unsaturated thermal conductivity and water vapor diffusivity of snow depend, at first order, mainly on the volume fraction of the phases, and the contribution of the phase distribution is secondary.

I recommend adding prediction intervals and standard deviations for the fits of n and alpha. These would be helpful for quantifying the uncertainty of the fits if they are used in the future.

110  We agree with the reviewer and added the standard deviation of each coefficient of the proposed regressions in Table 2 of the revised version of the manuscript.

You need to discuss the fact that optically equivalent radius derived from the SSA is not the same as the radius used by Yamaguchi2012. I think it is a very important factor since the grain diameter can vary a lot between methods and definition
115  (here, optically equivalent radius vs equivalent sphere radius).

We agree with this comment. The grain diameter estimation of Yamaguchi et al. (2012) slightly differs from ours: our definition is based on 3D snow images while Yamaguchi et al. (2012) is based on its 2D counterpart obtained from outlines of disconnected ice grains. While the conceptual definition is quite similar for both of the methods, where an "equivalent diameter"
120  is estimated, some typical differences between 2D and 3D are well-documented (e.g Brzoska et al. (1999), Cooperdock et al. (2019)):

[Figure]

**Figure 2.** a) $\alpha_{\mathrm{vg}}$, b) $n_{\mathrm{vg}}$ and c) $\theta_w^r$ parameters of the VG model as a function of $\rho/d$ or $\rho$ for imbibition and drainage. The regressions of Yamaguchi et al. (2012) are shown by black lines, the values used to deduct those regressions are shown by black disks ("S-samples", composed of refrozen MF - see Yamaguchi et al. (2012)). The additional drainage measurements from Yamaguchi et al. (2012) are shown by circles (MF samples) and stars (RG samples). Measurements from Adachi et al. (2020) are shown by empty gray markers. Measurements from Lombardo et al. (2025) and Katsushima et al. (2013) are shown by empty orange triangles and blue circles. From this work, parameters from imbibition and drainage simulations are shown by colored disks and triangles, respectively, the colors showing the snow types. The proposed regressions based on our simulated data are shown by dashed and dotted lines (see Table 2).

– The 2D method tends to neglect all snow structures related to necks between large disconnected grains;

- Due to the combination of gravity and projective effects, grain diameters may appear different in 2D than they really are in 3D;

- Tiny air bubbles or holes inside ice structures are often overlooked in 2D outlines.

Depending on grain types and specific morphologies, these methodological differences may result in an overestimation or an underestimation in grain sizes so that an exact conversion between the two approaches cannot be reasonably achieved. As a consequence, comparing our results to those of Yamaguchi et al. (2012) in Fig. 2 should be realized by keeping in mind that estimation errors on the $\rho/d$ values might exist, potentially as systematic errors (stretching of regression curves to the right or to the left) or as an increased dispersion of the dataset. Based on existing literature comparisons (e.g Brzoska et al. (1999), Cooperdock et al. (2019)), and on the figures obtained here (number and diversity of investigated snow samples and morphologies) we are quite confident that such errors are moderate and that overall comparisons between regressions obtained from the different methodologies are valid. Very precise comparisons are however more difficult and the difference between the regression of Yamaguchi et al. (2012) and our regression for $\alpha_{vg}$ in drainage could be e.g. interpreted as a consequence of this methodology difference. With this regard, our regression can be seen as an updated version of Yamaguchi et al. (2012)'s regression, proposing estimations that are based on systematic 3D diameter estimations, which are perfectly adapted to tomographic measurements.
Such explanations where added in the "3.1.3 Analysis of the VG parameters" subsection.

I would suggest adding the alpha ratio (alpha_wetting / alpha_drainage) as a quantification of hysteresis (e.g. Line 209) and compare this to the literature values (Adachi et al., 2020; Leroux and Pomeroy, 2017). It would also be interesting to see if this ratio is dependent on grain type or another parameter/combination of parameters.

We agree with the Reviewer and included a paragraph on the quantification of the hysteresis based on the suggested ratio, in the revised paper Section 3.1.3. The paragraph reads: "We use the $\alpha_{vg}$ parameter to quantify the degree of hysteresis of the WRCs, based on the ratio $\alpha_{vg}$ from imbibition over $\alpha_{vg}$ from drainage. For our set of images, this ratio ranges from 1.19 to 1.42, with an average value of 1.28. It is consistent with the ratios measured by Adachi et al. (2012), for which values of 1.46, 1.52 et 1.38 are found for the S, M, and L samples, respectively. No correlation was found between this ratio and grain type, grain size, or density, and the values confirm a small range of hysteresis ratio around 1.5, which is lower than the classical value of 2 used for soils (Leroux and Pomeroy, 2017)."

There is some discussion/description of the effect of the REV. However, I was still confused by how the REV was selected (Line 175). Is it just the maximum size of the uCT scan by Calonne2012? Given the range of voxel side lengths from 2.5 to 10mm, compared with typical grain sizes of MF which can be up to 1-2mm, it would seem to me that there would be some relationship between the REV size and the snow microstructure/relevant length scale which would have an effect on the simulations. Could you provide some clarification on this?

The selected volumes are the maximum size of the uCT scans by Calonne et al. (2012), for which we checked that they correspond to REVs for the determination of the $n_{vg}$ and $\alpha_{vg}$ parameters. As shown in Figure 1, the REV is reached for all the images except for the $n_{vg}$ parameter for the two melt forms samples (NH5 and NH3) in the case of drainage, for which larger uncertainties could thus exist. This uncertainty is provided in the manuscript and recalled in the revised Section 4 on the study limitations.

Is R2 the correct metric for such a nonlinear function? Or perhaps another metric like a mean absolute error which can provide an error in physical units? Or something else?

We agree with the reviewer and chose to indicate the mean absolute errors (MAE) to characterize the nonlinear relationships, instead of the correlation coefficient R. The MAE values are for instead shown in the revised Figure 3 for each image.

[Figure]

**Figure 3.** Numerical imbibition and drainage WRCs for different types of snow samples with the corresponding VG fits. The curve colors represent the different snow types, the MAE on $\theta_w$ of the fits are expressed in percent.

(Line 262) Could you cite work demonstrating that this is a good proxy? I understand that mean curvature and mean pore size are correlated but perhaps other microstructural parameters (e.g. thickness) would be more suitable? Could you elaborate on why this makes sense for draining but not for imbibition? Did you look at the relationship for imbibition?

175  Mean curvature has been used for years as a grain size estimator (see e.g. Lesaffre et al. (1998) -section 4.2). It shows very good correlation to SSA (e.g. Flin et al. (2003) - Fig. 5) and its distribution is widely used to characterize snow microstructure and its time evolution (e.g. Calonne et al. (2014)). By just considering the complementary part of the ice grain image, the curvature distribution also gives pertinent information on the pore size distribution, as the curvature distribution inherently contains such type of information. We agree that thickness measurement might, potentially, be an interesting estimator, but we
180  presently have no access to any appropriate numerical tool to confidently characterize this quantity. Actually, we have already tried many other parameters in order to find correlations to the $n_{\rm vg}$ value, but only some very particular curvature distribution related estimators gave convincing results.
We recently improved the $n_{\rm vg}$ parameterization by using the interquartile range (IQR) of the distribution, which is more suited to the shape of our distributions than the previously used standard deviation (see Fig. 4). This estimator also applies with the
185  same success to the imbibition values, with much less amplitude on the $n_{\rm vg}$ variation.

What was the rate of drainage in the drainage experiments and/or what pressure did you set to induce suction out of the snow? Is it possible that this had an effect on the amount of residual water that was trapped and therefore led to the higher residual liquid water content values compared to the literature?
190

The model implemented in the software SatuDict is the Pore Morphology Method (PMM) (e.g., Hilpert and Miller, 2001; Silin and Patzek, 2006; Schulz et al., 2015; Berg et al., 2016; Liu et al., 2022; Arnold et al., 2023). The PMM does not require to solve any partial differential equations. Therefore, no physical boundary conditions, such as rate of drainage, pressure are applied. The PMM uses a sphere with a radius $r$ as a probe to detect the pore space that is accessible by the non-wetting phase
195  (NWP), here the air. This radius is computed from the Young–Laplace equation: $r = 2\gamma\cos(\psi)/p_c$ where $p_c$ is the capillary

[Figure]

**Figure 4.** $n_{vg}$ parameter as a function of the interquartile range $IQR_{MC}$ obtained for the mean curvature distribution computed on each 3D dry snow image. Regressions for imbibition and drainage are shown with dotted and dashed lines.

pressure, $\gamma$ is the interfacial tension and $\psi$ is the contact angle. Morphological operations, namely, erosion and /or dilation are used in the PMM (Hilpert and Miller, 2001). The algorithm of the PMM can be decomposed into several steps as follows:

– In drainage condition, the porous medium is initially saturated with the wetting phase (WP) ($p_c = 0$). The invading NWP is connected to the inlet, which is the NWP reservoir, and the WP can escape through to the outlet, the WP reservoir. (i) Then, $p_c$ is increased incrementally, i.e. $r$ is decreased incrementally. The solid phase is first dilated by a sphere with radius $r$ (ii) All the pores connected to the NWP reservoir are labelled as NWP. (iii) The NWP is then dilated with the same sphere with radius $r$. The remaining pores are filled with the WP. The saturation can be then calculated. (iv) All the pores filled by the WP disconnected from the WP reservoir are considered as WP residual, and are no more considered in the next steps. All these steps are repeated by increasing the value of the pressure, i.e. decreasing the value of $r$. In the present case, the radius $r$ has been decreased gradually with a step of 2 pixel size.

– In imbibition condition, the porous medium is initially saturated with the NWP. The invading WP is connected to the inlet, which is the WP reservoir, and the NWP can escape through the outlet, the NWP reservoir. (i) Then, $p_c$ is decreased incrementally, i.e. $r$ is increased incrementally. The solid phase is first dilated by a sphere with radius $r$. (ii) The NWP is then dilated with the same sphere with radius $r$. (iii) All the pores connected to the WP are now labelled as WP. The remaining pores are NWP. The saturation can then be calculated. (iv) All the pores filled by the NWP disconnected from the NWP reservoir are considered as NWP residual, and are no longer considered in the next steps. As for the drainage condition, all these steps are repeated for the next value of the pressure, i.e. the next value of $r$. In the present case, the radius $r$ has been increased gradually with a step of 2 pixel size.

The PMM has been used to simulate the quasi-static Water Retention Curves (WRC) of porous media (glass beads, sands,..) in drainage or imbibition conditions (Hilpert and Miller, 2001; Liu et al., 2022). In imbibition conditions, if step (iv) is ignored, the PMM also allows for the computation of Mercury Injection Capillary Pressure curves (MICP), which are a commonly used technique for measuring porosity and pore throat size distribution, for instance (Hilpert and Miller, 2001; Berg et al., 2016). As underlined in Hilpert and Miller (2001), the accuracy of the PMM may depend on the resolution and size of the 3D images and of the definition of the structural element. Finally, it is worth mentioning that the boundary conditions applied on the four sides of the 3D images that are not connected to the NWP or WP reservoirs may also play a role in the WRC curve simulations, even if this point has not been discussed in the literature, to the best of our knowledge. In Vogel et al. (2005), impervious boundary conditions are applied, whereas other conditions, such as symmetry, displaced fluid outlet, or invading fluid inlet, can also be applied (Berg et al., 2016). These boundary conditions may lead to different values of the NWP (air) residuals during the imbibition process, since these residuals can be trapped or not at the boundary. Depending on the boundary conditions, the

225    maximum water saturation ($\theta_w^s$) can range from 45% to 90% of the porosity. Despite such large differences, the values of $\alpha_{vg}$ and $n_{vg}$ in the van Genuchten (VG) model (van Genuchten, 1980) remain almost constant, with an effect only visible for $\alpha_{vg}$, with differences around 10% of the value compared to the case $\theta_w^s = \phi$. This was also reported in the experiments of Likos et al. (2014) and Farooq et al. (2024), which showed that having $\theta_w^s$ smaller than $\phi$ generally implies greater $\alpha_{vg}$ values, but has no significant impact on the $n_{vg}$ values. The impact of the boundary conditions is less pronounced in drainage conditions, 230    since the water residuals are mainly located at the junction between grains.

In the present study, the PMM implemented in the SatudDict software was used to compute the WRCs of the 34 snow samples. We computed (i) a primary imbibition curve assuming that there is no air (NWP) residuals as in MICP experiments, thus $\theta_w^s = \phi$ in van Genuchten (VG) model, and then (ii) a primary drainage curve until to reach the water (WP) residuals ($\theta_w^r$). In both cases, symmetric boundary conditions are applied on the four sides of the volumes.

235    **In the revised version**, the description of the pore morphology method was substantially modified and includes all the above considerations (see introduction and Section 2.3)

(Line 252) I think you can remove the explanation of the fit error. As you state, this is not likely to account for the discrepancy and I think is overreaching what you can really say with the small differences in R2. I would also suggest adding the fact that 240    the Yamaguchi2012 parameterization was based on their S-samples, which were sieved. As seen in Fig.4 of Yamaguchi2012, the parameterization of n also doesn't fit the N-sample data but does seem to fit your data and your value of 4.7.

We agree with the Reviewer and deleted the explanation about the fit error in the revised version of the paper. We included more explanations about the parameterizations of Yamaguchi et al. (2012), and added the data points of Katsushima et al. (2013) 245    and Lombardo et al. (2025). Also, as stated above, we recently improved the $n_{vg}$ parameterization by using the interquartile range (IQR) of the distribution for both drainage and imbibition, which is more suited to the shape of our distributions than the previously used standard deviation (see Fig. 4). The paragraph in the revised paper reads: "For drainage, our $n_{vg}$ values are spread and show little correlation with $\rho/d$. Looking at the experimental data, a large spread is also observed in Katsushima et al. (2013), Lombardo et al. (2025), and for the rounded grains samples of Yamaguchi et al. (2012). Our estimated $n_{vg}$ values 250    overall do not follow the regression of Yamaguchi et al. (2012), although the $n_{vg}$ values of the melt forms samples are closer to this regression compared to the other snow types. Let us recall that Yamaguchi et al. (2012) presented a regression based on drainage experiments on sieved melt forms only (black filled dots in Fig. 2.d), which was in good agreement with $n_{vg}$ values estimated on other samples of natural melt forms (black circles), but in poor agreement for samples of natural rounded grains (black stars), for which $n_{vg}$ appear to be independent of $\rho/d$."

255

(Line 275) Can you really conclude that the separation is due to the grain type as opposed to density based on the presented data?

Based on our results, it is difficult to conclude if the separation is due to grain type or density. Most of the time, melt forms 260    samples have high density, but snow types, such as rounded grains, can also feature high density. However, as seen in Fig. 2.e, the denser rounded grains sample (pink) has a density close to the less dense melt forms sample (red), while showing very different $\theta_w^r$ values, which suggest a stronger effect of snow type compared to density. We specify both effects in the revised text in the paragraph "$\theta_w^r$ **parameter**".

265    Why did you not use theta_s from the simulations? Or analyze theta_s as you did for theta_r in Figure 6?

As mentioned above, the boundary conditions on the four sides of the 3D images not linked to the WP or NWP reservoir may play a important role on the $\theta_w^s$. This is one of the limitation of the PMM method (see our previous reply to a main comment, concerning the limitations of our study). In the simulations, we first computed a primary imbibition curve assuming that there 270    is no air (NWP) residuals as in MICP experiments, thus $\theta_w^s = \phi$, and then a primary drainage curve until reaching the liquid water (WP) residuals $\theta_w^r$. In both cases, symmetric boundary conditions are applied on the four sides of the volumes, a liquid water reservoir is located at the bottom and an air reservoir is located at the top of the sample.

[Figure]

**Figure 5.** Unsaturated hydraulic conductivity $\mathbb{K}_w^u$ as a function of the effective saturation for (a) the whole set of snow samples, and (b) the 5 selected samples. Computations were performed on the snow images from the drainage simulations only. The dry density of the snow samples is represented by the colorbar. The VGM model, using the values of intrinsic permeability from Calonne et al. (2012) and the VG parameters from the regressions of this work presented in Table 2 on each snow sample is shown with the solid lines. The VGM model, using the values of intrinsic permeability from Calonne et al. (2012) and the regression from Yamaguchi et al. (2012) is shown with the dashed lines.

(Line 355) Can you add some text addressing the "so what?" of these results like you do for the thermal conductivity? Currently, the results are just presented, and the conclusion is just that the model seems to match the data. The bigger picture or effect is missing.

We agree with the reviewer and added more elements to our analysis in the text. For the unsaturated hydraulic conductivity, as the log scale makes it somewhat hard to see in Fig. 5, we now provide the MAE values to compare quantitatively our regressions and the regression of Yamaguchi et al. (2012) against the estimated regressions based our numerical results (Table 2). For the melt forms samples, a slight improvement is found using our regressions, with MAE values around 10% for Yamaguchi et al. (2012) and around 9% for our model. This similarities can be related to the fact that, on one hand, the regression of (Yamaguchi et al., 2012) provides slightly better estimates of $\alpha_{vg}$ for this snow type, compared to our regression (Fig. 2.b); on the other hand, $n_{vg}$ values are better estimates from our regression (Fig. 2.d and Fig. 4). For the other snow types, the VGM model using our shape parameter estimates provides better predictions of unsaturated hydraulic conductivity, showing MAE values around 17% compared to the VGM model using the estimates of Yamaguchi et al. (2012), showing MAEs around 22%. Indeed, for all the snow types excluding melt forms, both shape parameters $\alpha_{vg}$ and $n_{vg}$ are overall better estimated using our regression (as optimized to best match our numerical simulations). This difference highlights the advantage of considering a high diversity of snow in the regressions, compared to the regression of Yamaguchi et al. (2012) based only on melt forms samples.

Is it possible to include a comparison to current parameterizations/models for hydraulic conductivity and diffusivity as was done for thermal conductivity. The comparison for thermal conductivity really shows how different the results can be, and this large difference demonstrates the importance of the approach provided here.

The manuscript has been supplemented with more comparisons between our estimations of effective parameters and models of the literature. For the thermal conductivity, we added a self-consistent estimate that represents an arrangement of spheres with 3 connected phases (it is an implicit relationship where a polynomial equation needs to be solved) (Torquato, 2005). This

[Figure]

**Figure 6.** Unsaturated thermal conductivity $k^{\mathrm{u}}$ as a function of the effective saturation for (a) the whole set of snow samples, and (b) the 5 selected samples. Computations were performed on the snow images from the drainage simulations only. The dry density of the snow samples is represented by the colorbar. The suggested regression is shown by solid lines and the self-consistent estimate for 3 phases is shown by dashed lines.

[Figure]

**Figure 7.** Relative water vapor diffusivity $D^{\mathrm{u}}/D$ as a function of the water saturation $S_w$ for drainage simulations of (a) the whole set of snow samples and (b) the 5 selected samples. The dry density of the snow samples is given by the colorbar. The proposed regression of unsaturated diffusivity is shown by a black solid line, and the models of Millington and Quirk (1961) and Moldrup et al. (2000) are shown with gray dotted and dashed lines respectively.

estimate is shown in Figure 6 and is rather close to the proposed fit, which emphasizes the linear aspect of this property. For the water vapor diffusivity, we added the models for unsaturated structureless natural soils of Millington and Quirk (1961) and Moldrup et al. (2000) in Figure 7 (see Kristensen et al., 2010). They are derived from the same equation shape with a different exponent coefficient and show similar results. The relationship follow the general form of estimates: $D^{\mathrm{u}}/D_{\mathrm{dry}} = (1 - \theta_w/\phi)^a$ classically used for soils (Kristensen et al., 2010) with $a = 3$ in our study, $a = 10/3$ in Millington and Quirk (1961) and $a = 5/2$ in Moldrup et al. (2000). For the hydraulic conductivity, instead of comparing the result using the shape parameters of Yamaguchi et al. (2012) with our shape parameters fitted on each image, we compared it with our shape parameters from the estimated regressions shown in Table 2, which allows a fairer comparison.

(Line 388) As presented, your regression matches your simulations MUCH better than the other methods, which is partially due to the fact that you are fitting your own simulation data. Are you capturing any physical effects that lead to better results that the other methods don't? And what limitations do we need to consider before simply adopting your regression?

As clearly stated in the revised version of the introduction, in many models (Daanen and Nieber, 2009; Leroux and Pomeroy, 2017; Moure et al., 2023), the unsaturated thermal conductivity of snow $k^u$ is derived based on an arithmetic mean, such as $k^u = k_{dry}^{\mathrm{eff}}(1-\theta_w) + k_w\theta_w$, with $k_{dry}^{\mathrm{eff}}$ the effective thermal conductivity of dry snow and $k_w$ the intrinsic thermal conductivity of liquid water. By doing so, the impact of the microstructure and phase connectivity is not considered, although their importance was shown in dry snow (Calonne et al., 2011). In the operational Crocus model (Vionnet et al., 2012; Lafaysse et al., 2025), the parameterization of (Yen, 1981), only valid for dry snow, is extrapolated to wet snow, which leads to the assumption that ice and water have the same impact on the snow effective thermal conductivity, although ice conducts four times more than water. In contrast, the proposed regression is based on thermal conductivity simulations that account for the pore scale distribution of the different phases and consider the intrinsic thermal conductivity of each phase. This proposed regression is robust as it relies at the first order on the volume fractions of the different phases, which are easy to access, and does not rely on the shape parameters of the VG model (for which uncertainties could be large) as for the hydraulic conductivity for instance.

(Line 415) What is causing this bias?

We removed the sentence concerning those biases, as at this stage we do not have a physical explanation for it, and we wanted to keep a simple regression following the form of current regressions from the literature (Kristensen et al., 2010).

(Line 429): Can you provide some discussion about how different the results are if you use the imbibition simulations and why they are different?

The water vapor diffusion and thermal conduction depend primarily on the volume fraction and connectivity of each phase, which are well described in our simulations and should be similar between drainage and imbibition. We tested the difference between drainage and imbibition simulations for the thermal conductivity, and a difference of 1.8 % was found (without obvious negative or positive bias), which is small compared to the other sources of error. Only the unsaturated hydraulic conductivity depends clearly on both the volumetric fraction and the 3D distribution of the phases and could thus show some difference between imbibition and drainage. However, as seen with Fig. 8, the difference between our results in imbibition and drainage is smaller than the difference with the results of Yamaguchi et al. (2012), which suggests that small changes could be also expected between both processes for $\mathbb{K}_w^{\mathrm{u}}$. We added more discussion about this point on the manuscript Sec. 4 "Main limitations".

**Technical comments**

**Writing:** Not having a space between paragraphs is confusing as some paragraph breaks are ambiguous since the document is full justified.
The beginning of the paragraphs is indented in the revised manuscript.

[Figure]

**Figure 8.** Illustration of the VG shape parameters regressions on the WRCs of 2 representative samples as a function of the effective saturation. The presented VG models are: imbibition and drainage models from this work and the models of Daanen and Nieber (2009), Yamaguchi et al. (2010) and Yamaguchi et al. (2012) (see Table 2).

**Line 2:** It is somewhat unclear if the "its" refers to snow or liquid water. Suggest to rephrase.
  The sentence now reads: 'Liquid water flows by gravity and capillarity in snow and drastically modifies snow properties.'

**Line 15:** Grammar - I don't think "implies" can be used like this. Perhaps causes? Also, drastic changes of what? Microstructure?
  We agree with the reviewer and modified the sentence, so it reads: 'Liquid water, introduced by rain or melt, flows into the snowpack and causes drastic changes in the microstructures and properties of the snow'.

**Lines 17-24:** This part of the Intro is confusing. Suggest rewording/reorganizing to make the topics in each sentence connect more clearly. E.g. Perhaps new paragraph at Line 20 and what does "to predict wet snow" mean?
  The introduction was improved substantially, as described in the general comments.

**Line 21:** Please add citations for the flow types.
  We added the references of Gerdel (1954) and Colbeck (1976).

**Line 24:** Probably should cite SNOWPACK as well.
  Yes, we included SNOWPACK in the revised version.

**Line 24-26:** The sentence starts with "recently" but then you cite Daanen and Nieber from 2009. Perhaps just remove "recently".
  We agree and this sentence was modified in the revised introduction.

**Line 27:** Grammar - "little" is not correct.
  We agree and this sentence was modified in the revised introduction.

**Line 27:** "effective wet snow properties" is not self-explanatory. Perhaps define or introduce the parameters you address in the paper.
  Modified accordingly. The parameters are introduced in the revised introduction.

**Line 36-38:**  The way this definition of imbibition and drainage is written makes it seem like you always have to start or end at saturation, which I don't think is 100% correct or is, at a minimum, a bit confusing as written. Maybe consider rewording.
We changed the sentence accordingly : "This relationship can be provided for a first imbibition, which corresponds to the wetting a dry porous medium by a liquid until it reaches saturation, or for a first drainage, which corresponds to the drainage of a fully saturated porous medium, and also for a drainage or imbibition from an intermediate state of saturation."

**Line 41-43**  : Please add citations for the soil examples
The soil examples were removed from the revised verison.

**Line 43-44:**  For completeness, I would add that there are a few studies that did measure imbibition: Coléou et al. (1999), Adachi et al. (2020), Lombardo et al. (2025) [Disclaimer – I am an author on this paper.]
Thank you for pointing this out. We included these studies in the revised introduction.

**Line 45:**  The brief discussion of Adachi2020's MRI method is a bit confusing since you seem to be transitioning to discuss hysteresis, but then go on to discuss how the VG parameters were fit in the other experiments. Since I think you should mention Adachi2020 in the comment above, perhaps you can just move that part and not discuss it here.
We agree. This paragraph was modified in the revised introduction.

**Lines 41-54:**  This paragraph generally needs rewriting. It is confusing and the message is unclear, likely because the topic changes several times and the transitions between the topics are not obvious/fluid.
We agree. This paragraph was modified in the revised introduction.

**Line 53:**  I think more precisely, Yamaguchi et al. (2025) provided a parameterization for sieved melt forms. There were also the "N-samples" which were natural snow samples, but they did not fit the parameterization.
Thank you for the comment. We include this information in the revised paper.

**Line 55:**  Grammar – "enable to capture" is incorrect and the commas are also incorrect.
This sentence was modified.

**Line 55:**  I think you need to discuss Hydraulic Conductivity before this paragraph since the "overall hydraulic behavior" is dependent on the not only the capillary forces described in the WRC but also the flow rate which is often done using the (saturated and relative) hydraulic conductivity, as you discuss later.
We agree with the comment and modified the introduction accordingly. A paragraph on the unsaturated hydraulic conductivity of snow was added.

**Line 65:**  Typo: literature
The correction was done.

**Line 67:**  I think Yamaguchi et al. (2025) is also a relevant citation here
The citation was included.

**Line 69**  : This sentence on models seems out of place with respect to the topic of this paragraph.
We agree and, as the introduction was substantially modified, this sentence was deleted.

**Line 78:**  Accounted "for" here
The correction was done.

**Line 76-79:**  I think you remove this sentence about wet snow metamorphism and just discuss this limitation later in the manuscript.
Modified accordingly.

**Line 94:**  Aren't the subscripts just the phase (ice, water, or air)? This sentence about the subscripts is confusing.
This sentence was clarified in the revised paper

**Line 102** : Can you make a statement about whether the snow samples in Calonne2012 were natural, laboratory generated, sieved, etc.? Of course the information is in the original paper, but it would be easier for the reader if this was just briefly stated here.

The following sentence was included: 'The imaged snow samples are composed of natural snow collected in the field as well as snow obtained from evolution under controlled environmental conditions in cold laboratory, which replicates natural snow evolution.'

**Line 117-118:** Add a citation for this

A paragraph on the Bond number, measuring the importance of gravitational forces compared to surface tension forces, was included in the revised paper and the following citations were added: Auriault (1987); Auriault et al. (2009).

**Line 119:** Why did you choose 12 degrees? Cite?

The contact angle between liquid water and ice close to 0°C has been measured around 12° Knight (1967). The reference has been added to the manuscript.

**Line 134:** Typo: Ponds

The correction was done.

**Line 131-135:** I don't think you need these sentences about the configuration. It is clear from the rest of the intro what imbibition and drainage are and the measurement of WRCs doesn't need to represent a situation that occurs in nature.

We agree and removed these sentences.

**Line 161:** Small formatting suggestion: in Eq 6, VG subscripts of alpha, n, and m look a bit weird to me. Perhaps increase their offset or decrease their size.

Modified accordingly.

**Line 167:** I think this paper demonstrates nicely these relationships and how they are only roughly correct: van Lier and Pinheiro (2018). Could be a nice citation to include.

Thank you for the reference, which was included in the revised paper.

**Line 203:** Not being particularly familiar with these tensors, I was wondering if the non-diagonal terms being negligible has a physical meaning and/or is standard. If it's important, perhaps provide a citation which contains this explanation so that a reader unfamiliar with this can inform themselves?

Neglecting the diagonal components of the tensors is a classical operation that is for instance described in Calonne et al. (2011) or in Fourteau et al. (2021). Its means that the x, y ans z directions are the primary axes of the microstructure, which is the case if the snow has been sampled preserving its z axis. A reference with more details has been added to the sentence.

**Line 213:** It is unclear what this sentence is supposed to mean. It seems like you are just redescribing imbibition and drainage.

We removed the sentence which repeated elements already mentioned.

**Line 222:** I think you can't compare the shape when using liquid water content. If you want to compare the shape, you should plot saturation vs pressure as this normalizes the x-axis. I also don't really understand what is meant by "smoother inflections".

We want to keep the absolute scale and not introduce $S_e$ that soon in the paper as it allows a better appreciation of the different samples and their water contents. As seen in Fig. 3, the linear scale allows for a better observation of the WRCs of the different microstructures, thus from a qualitative point of view, comparisons can still be made. For instance, even with similar maximum saturation, NH2 (MF) and NH5 (MF) are very different from grad3 (DH) and show sharper curves. We changed the sentence to be more nuanced as it is hard to separate the effects of density and grain size on this effect: "More subtly, the WRCs tend to show sharper curves for high density and large grains samples. ".

**Line 228:** It seems the point density for imbibition is higher than for drainage? i.e. there are more points on the imbibition curves in Fig 5. Is this true? That certainly would impact R2.
There is same amount of points for the imbibition and drainage curves. In the flat area of the curve, more points are visible for the imbibition curve as they are less steep.

**Line 241** : Can you quantify this agreement?
The matching of the numerical estimates to the regression from Yamaguchi et al. (2012), and to the proposed regressions has been added to the text as MAEs.

**Line 251** : This makes sense since Yamaguchi2012 shows only drainage data – perhaps state this.
We included this information in the revised paper.

**Line 283:** I don't understand this sentence. As written, it seems like you are saying that the saturated liquid water content (LWC) can be up to 30% of the porosity. Do you mean that it can be 30% less than the porosity? i.e. Saturated LWC = 0.7xporosity?
Yes we meant 30% less than the porosity, this part has been revised for more clarity.

**Line 336:** In order to make the statement in Line 336, I think you need to show the plots for other microstructural parameters such as ssa, curvature, etc.
The sentence was reformulated to be clearer: "The relationship describes an exponential increase, which tends, for all samples, to merge into a single curve. This shows that the water permeability is at first order driven by the water content and the snow density, and that other dependencies with other microstructural parameters are, if any, of lesser strength."

**Line 373:** It is unclear if you mean Fig 11 a or b. Could you also postulate why there seems to be a density bias in your regression?
Both Fig. 11.a and Fig. 11.b illustrate the regressions. This was more clearly explained in the revised version of the manuscript. We think the density bias on the high-density and low-density can be a second order effect, and, as we did not want to go into this level of details, we removed the end of the sentence that can be misleading.

**Line 396:** This term is also the saturation, which might be better to use since saturation is used elsewhere in the paper already.
The term $\theta_w/\phi$ is indeed the liquid water saturation $S_w$, we changed the Figure and the text accordingly for more clarity.

**Line 400-401:** For this statement, I think you need to include something about the close pore water content. As is, it doesn't make sense that the plot defines the domain of Du - as you state afterwards, it's theta_cp, so I would include that immediately in the sentence to make that clear.
We agree and define the close off earlier in the revised paper.

**Table 1:** Can you rename the samples to something more intuitive? I understand these are probably the names in Calonne et al. (2012) but for the reader here that doesn't really matter (of course you can correlate them to the Calonne et al. (2012) paper). It would be easy to use the snow type as the name MF_l, MF_s, DH, etc. (or similar). This would allow the reader to immediately know what the snow properties are instead of having to memorize a random code since samples are grouped by grain type throughout the paper.
We keep the names as they refer to several previous papers (e.g. Flin et al., 2003; Calonne et al., 2014, 2012). However, we checked that the snow type is systematically provided in the paper together with the name sample.

**Table 2:** Is there a unit issue in the fits for alpha? There are two orders of magnitude difference in the coefficient between Yamaguchi et al. (2012) and your data ($10^4$ vs $10^6$).
There is two order of magnitude difference in the coefficient because the exponent value is smaller. To avoid the fact that a large range of values for the two coefficients can describe almost the same curve, we chose to simplify our new regression by keeping the multiplicator coefficient of Yamaguchi et al. (2012) (4.4 $10^4$) and only fitting the exponent coefficient. This is motivated by the fact that the first coefficient has more impact on the small $\rho/d$ values, where most of the measured values of Yamaguchi et al. (2012) are, and the exponent has more impact on the larger values of $\rho/d$, where most of our numerical estimates are.

[Figure]

**Figure 9.** Effective relative water permeability as a function of the effective saturation for the whole set of snow samples. The colors represent the snow types (see Fig. 2 for colors legend).

**Figure 5:** It would be helpful to add the R2 values to the legends for each sample in Figure 5.
The mean absolute errors (MAE) have been added to each sample the Figure (see Fig. 3).

**Figure 6:** Something is weird with the caption formatting.
The correction was done.

**Figure 6:** I think the (a)-plot as an overview of the data is nice, but could you also maybe replot only your data such that the reader can see the difference between imbibition and drainage for your data. Currently, the plot is too difficult to read to make out what is what.
Based on this comment, we modified the figure, and added subplots for the $n_{\mathrm{vg}}$ and $\alpha_{\mathrm{vg}}$ values separated for imbibition and for drainage, so that it enables a better vizualisation of the data points and the different regressions (see Fig. 2).

**Figure 7:** Please add the grain type legend
Changed accordingly.

**Figure 8:** This plot is very hard to read. Please improve the lines/marker differences to differentiate the datasets.
We simplified the Figure by changing the log scale of the figure to a linear scale. Besides, the comparison is now based on 2 'imaginary' snow samples, instead on 4 of our snow samples, for a more independent approach (see Fig. 8). The properties of the two samples have been chosen to be representative of melt forms (sample 1: d = 1.5 mm, $\rho$ = 450 kg m$^{-3}$, IQR$_{\mathrm{MC}}$ = 5 mm$^{-1}$) and precipitation particles (sample 2: d = 0.1 mm, $\rho$ = 130 kg m$^{-3}$, IQR$_{\mathrm{MC}}$ = 15 mm$^{-1}$).

**Figure 9a:** Can you repeat this plot but with grain types and/or indicate grain types with the marker shape on the existing plot so that we can see how the grain type matches up?
Figure 9 shows Figure 9.a of the paper, where the density colorbar has been replaced by the snow types colors. To our perspective, this display does not bring more than Fig. 9.b, considering that the apparent clusters of the snow types colors are also clusters of density (PP and DF are mostly low-density samples, and RG and MF are more high-density samples - see Fig. 2.e to see the distribution of density of our different snow types).

**Figure 13:** It is really hard to tell that the solid line is blue. Just make it black since there are no other solid lines?
Changed accordingly.

**Figures 9b, 10b, 11b:** Why are different saturation values (x coordinates) chosen for the different samples?

515        The steps of the simulations are computed in terms of pore size and not in terms of saturation. Thus, depending on the microstructure, the x-coordinates are different for each sample.

**References**

Adachi, S., Yamaguchi, S., Ozeki, T., and Kose, K.: Hysteresis in the water retention curve of snow measured using an MRI system, in: Proceedings to the 2012 International Snow Science Workshop, Anchorage, Alaska, vol. 485, https://arc.lib.montana.edu/snow-science/objects/issw-2012-918-922.pdf, 2012.

Adachi, S., Yamaguchi, S., Ozeki, T., and Kose, K.: Application of a magnetic resonance imaging method for nondestructive, three-dimensional, high-resolution measurement of the water content of wet snow samples, Frontiers in Earth Science, 8, https://doi.org/10.3389/feart.2020.00179, 2020.

Ahrenholz, B., Tölke, J., Lehmann, P., Peters, A., Kaestner, A., Krafczyk, M., and Durner, W.: Prediction of capillary hysteresis in a porous material using lattice-Boltzmann methods and comparison to experimental data and a morphological pore network model, Advances in Water Resources, 31, 1151–1173, https://doi.org/10.1016/j.advwatres.2008.03.009, 2008.

Arnold, P., Dragovits, M., Linden, S., Hinz, C., and Ott, H.: Forced imbibition and uncertainty modeling using the morphological method, Advances in Water Resources, 172, 104 381, https://doi.org/https://doi.org/10.1016/j.advwatres.2023.104381, 2023.

Auriault, J.-L.: Non saturated deformable porous media: quasi-statics, Transport in porous media, 2, 45–64, https://doi.org/10.1007/BF00208536, 1987.

Auriault, J.-L., Boutin, C., and Geindreau., C.: Homogenization of coupled phenomena in heterogenous media, Wiley-ISTE, London, 2009.

Berg, S., Rücker, M., Ott, H., Georgiadis, A., van der Linde, H., Enzmann, F., Kersten, M., Armstrong, R., de With, S., Becker, J., and Wiegmann, A.: Connected pathway relative permeability from pore-scale imaging of imbibition, Advances in Water Resources, 90, 24–35, https://doi.org/https://doi.org/10.1016/j.advwatres.2016.01.010, 2016.

Bhatta, N., Gautam, S., Farhan, N. M., Tafreshi, H. V., and Pourdeyhimi, B.: Accuracy of the Pore Morphology Method in Modeling Fluid Saturation in 3D Fibrous Domains, Industrial & Engineering Chemistry Research, 63, 18 147–18 159, https://doi.org/10.1021/acs.iecr.4c02939, 2024.

Brzoska, J. B., Lesaffre, B., Coléou, C., Xu, K., and Pieritz, R. A.: Computation of 3D curvatures on a wet snow sample, Eur. Phys. J. AP, 7, 45–57, https://doi.org/10.1051/epjap:1999198, 1999.

Calonne, N., Flin, F., Morin, S., Lesaffre, B., Rolland du Roscoat, S., and Geindreau, C.: Numerical and experimental investigations of the effective thermal conductivity of snow, Geophys. Res. Lett., 38, L23 501, https://doi.org/10.1029/2011GL049234, 2011.

Calonne, N., Geindreau, C., Flin, F., Morin, S., Lesaffre, B., Rolland du Roscoat, S., and Charrier, P.: 3-D image-based numerical computations of snow permeability: links to specific surface area, density, and microstructural anisotropy, The Cryosphere, 6, 939–951, https://doi.org/10.5194/tc-6-939-2012, 2012.

Calonne, N., Flin, F., Geindreau, C., Lesaffre, B., and Rolland du Roscoat, S.: Study of a temperature gradient metamorphism of snow from 3-D images: time evolution of microstructures, physical properties and their associated anisotropy, The Cryosphere, 8, 2255–2274, https://doi.org/10.5194/tc-8-2255-2014, 2014.

Colbeck, S. C.: An analysis of water flow in dry snow, Water Resources Research, 12, 523–527, 1976.

Coléou, C., Xu, K., Lesaffre, B., and Brzoska, J.-B.: Capillary rise in snow, Hydrol. Process., 13, 1721–1732, https://doi.org/10.1002/(SICI)1099-1085(199909)13:12/13<1721::AID-HYP852>3.0.CO;2-D, 1999.

Cooperdock, E. H. G., Ketcham, R. A., and Stockli, D. F.: Resolving the effects of 2-D versus 3-D grain measurements on apatite (U–Th) / He age data and reproducibility, Geochronology, 1, 17–41, https://doi.org/10.5194/gchron-1-17-2019, 2019.

Daanen, R. P. and Nieber, J. L.: Model for Coupled Liquid Water Flow and Heat Transport with Phase Change in a Snowpack, Journal of Cold Regions Engineering, 23, 43–68, https://doi.org/10.1061/(ASCE)0887-381X(2009)23:2(43), 2009.

Farooq, U., Gorczewska-Langner, W., and Szymkiewicz, A.: Water retention curves of sandy soils obtained from direct measurements, particle size distribution, and infiltration experiments, Vadose Zone Journal, 23, e20 364, https://doi.org/10.1002/vzj2.20364, 2024.

Flin, F., Brzoska, J.-B., Lesaffre, B., Coléou, C., and Pieritz, R. A.: Full three-dimensional modelling of curvature-dependent snow metamorphism: first results and comparison with experimental tomographic data, Journal of Physics D: Applied Physics, 36, A49–A54, https://doi.org/10.1088/0022-3727/36/10A/310, 2003.

Fourteau, K., Domine, F., and Hagenmuller, P.: Impact of water vapor diffusion and latent heat on the effective thermal conductivity of snow, The Cryosphere, 15, 2739–2755, https://doi.org/10.5194/tc-15-2739-2021, 2021.

Galindo-Torres, S., Scheuermann, A., and Li, L.: Boundary effects on the Soil Water Characteristic Curves obtained from lattice Boltzmann simulations, Computers and Geotechnics, 71, 136–146, https://doi.org/https://doi.org/10.1016/j.compgeo.2015.09.008, 2016.

Gerdel, R. W.: The transmission of water through snow, Eos, Transactions American Geophysical Union, 35, 475–485, https://doi.org/10.1029/TR035i003p00475, 1954.

Hilpert, M. and Miller, C. T.: Pore-morphology-based simulation of drainage in totally wetting porous media, Advances in Water Resources, 24, 243–255, https://doi.org/10.1016/S0309-1708(00)00056-7, pore Scale Modeling, 2001.

Jettestuen, E., Helland, J. O., and Prodanović, M.: A level set method for simulating capillary-controlled displacements at the pore scale with nonzero contact angles, Water Resources Research, 49, 4645–4661, https://doi.org/https://doi.org/10.1002/wrcr.20334, 2013.

570  Joekar-Niasar, V. and Hassanizadeh, S. M.: Analysis of Fundamentals of Two-Phase Flow in Porous Media Using Dynamic Pore-Network Models: A Review, Critical Reviews in Environmental Science and Technology, 42, 1895–1976, https://doi.org/10.1080/10643389.2011.574101, 2012.

Katsushima, T., Yamaguchi, S., Kumakura, T., and Sato, A.: Experimental analysis of preferential flow in dry snowpack, Cold Regions Science and Technology, 85, 206–216, https://doi.org/10.1016/j.coldregions.2012.09.012, 2013.

575  Knight, C. A.: The contact angle of water on ice, Journal of Colloid and Interface Science, 25, 280–284, https://doi.org/10.1016/0021-9797(67)90031-8, 1967.

Kristensen, A. H., Thorbjørn, A., Jensen, M. P., Pedersen, M., and Moldrup, P.: Gas-phase diffusivity and tortuosity of structured soils, Journal of Contaminant Hydrology, 115, 26–33, https://doi.org/10.1016/j.jconhyd.2010.03.003, 2010.

Lafaysse, M., Dumont, M., De Fleurian, B., Fructus, M., Nheili, R., Viallon-Galinier, L., Baron, M., Boone, A., Bouchet, A.,
580  Brondex, J., Carmagnola, C., Cluzet, B., Fourteau, K., Haddjeri, A., Hagenmuller, P., Mazzotti, G., Minvielle, M., Morin, S., Quéno, L., Roussel, L., Spandre, P., Tuzet, F., and Vionnet, V.: Version 3.0 of the Crocus snowpack model, EGUsphere, 2025, 1–75, https://doi.org/10.5194/egusphere-2025-4540, 2025.

Leroux, N. R. and Pomeroy, J. W.: Modelling capillary hysteresis effects on preferential flow through melting and cold layered snowpacks, Advances in Water Resources, 107, 250–264, https://doi.org/10.1016/j.advwatres.2017.06.024, 2017.

585  Lesaffre, B., Pougatch, E., and Martin, E.: Objective determination of snow-grain characteristics from images, Annals of Glaciology, 26, 112–118, https://doi.org/10.3189/1998AoG26-1-112-118, 1998.

Likos, W. J., Lu, N., and Godt, J. W.: Hysteresis and Uncertainty in Soil Water-Retention Curve Parameters, Journal of Geotechnical and Geoenvironmental Engineering, 140, 04013 050, https://doi.org/10.1061/(ASCE)GT.1943-5606.0001071, 2014.

Liu, X., Zhou, A., long Shen, S., and Li, J.: Modeling drainage in porous media considering locally variable contact angle based on pore
590  morphology method, Journal of Hydrology, 612, 128 157, https://doi.org/https://doi.org/10.1016/j.jhydrol.2022.128157, 2022.

Lombardo, M., Fees, A., Kaestner, A., van Herwijnen, A., Schweizer, J., and Lehmann, P.: Quantification of capillary rise dynamics in snow using neutron radiography, EGUsphere, 2025, 1–36, https://doi.org/10.5194/egusphere-2025-304, 2025.

Millington, R. and Quirk, J.: Permeability of porous solids, Transactions of the Faraday Society, 57, 1200–1207, https://doi.org/10.1039/TF9615701200, 1961.

595  Moldrup, P., Olesen, T., Schjønning, P., Yamaguchi, T., and Rolston, D. E.: Predicting the Gas Diffusion Coefficient in Undisturbed Soil from Soil Water Characteristics, Soil Science Society of America Journal, 64, 94–100, https://doi.org/10.2136/sssaj2000.64194x, 2000.

Moure, A., Jones, N., Pawlak, J., Meyer, C., and Fu, X.: A thermodynamic nonequilibrium model for preferential infiltration and refreezing of melt in snow, Water Resources Research, 59, e2022WR034 035, https://doi.org/10.1029/2022WR034035, 2023.

Prodanović, M. and Bryant, S. L.: A level set method for determining critical curvatures for drainage and imbibition, Journal of Colloid and
600  Interface Science, 304, 442–458, https://doi.org/https://doi.org/10.1016/j.jcis.2006.08.048, 2006.

Schulz, V. P., Wargo, E. A., and Kumbur, E. C.: Pore-Morphology-Based Simulation of Drainage in Porous Media Featuring a Locally Variable Contact Angle, Transport in Porous Media, 107, 13–25, https://doi.org/10.1007/s11242-014-0422-4, 2015.

Silin, D. and Patzek, T.: Pore space morphology analysis using maximal inscribed spheres, Physica A: Statistical Mechanics and its Applications, 371, 336–360, https://doi.org/https://doi.org/10.1016/j.physa.2006.04.048, 2006.

605  Tengattini, A., Lenoir, N., Andà$^2$, E., Giroud, B., Atkins, D., Beaucour, J., and Viggiani, G.: NeXT-Grenoble, the Neutron and X-ray tomograph in Grenoble, Nuclear Instruments and Methods in Physics Research Section A: Accelerators, Spectrometers, Detectors and Associated Equipment, 968, 163 939, https://doi.org/https://doi.org/10.1016/j.nima.2020.163939, 2020.

Torquato, S.: Random Heterogeneous Materials: Microstructure and Macroscopic Properties, Interdisciplinary Applied Mathematics, Springer New York, ISBN 9780387951676, https://books.google.fr/books?id=PhG_X4-8DPAC, 2005.

610  van Genuchten, M. T.: A closed-form equation for predicting the hydraulic conductivity of unsaturated soils, Soil Science Society of America Journal, 44, 892–898, https://doi.org/10.2136/sssaj1980.03615995004400050002x, 1980.

van Lier, Q. d. J. and Pinheiro, E. A. R.: An Alert Regarding a Common Misinterpretation of the Van Genuchten $\alpha$ Parameter, Revista Brasileira de Ciência do Solo, 42, e0170 343, https://doi.org/10.1590/18069657rbcs20170343, 2018.

Vionnet, V., Brun, E., Morin, S., Boone, A., Faroux, S., Le Moigne, P., Martin, E., and Willemet, J.-M.: The detailed snowpack scheme Crocus
615  and its implementation in SURFEX v7.2, Geoscientific Model Development, 5, 773–791, https://doi.org/10.5194/gmd-5-773-2012, 2012.

Vogel, H.-J., Tölke, J., Schulz, V. P., Krafczyk, M., and Roth, K.: Comparison of a Lattice-Boltzmann Model, a Full-Morphology Model, and a Pore Network Model for Determining Capillary Pressure-Saturation Relationships, Vadose Zone Journal, 4, 380–388, https://doi.org/https://doi.org/10.2136/vzj2004.0114, 2005.

Xiong, Q., Baychev, T. G., and Jivkov, A. P.: Review of pore network modelling of porous media: Experimental characterisations, network constructions and applications to reactive transport, Journal of Contaminant Hydrology, 192, 101–117, https://doi.org/https://doi.org/10.1016/j.jconhyd.2016.07.002, 2016.

Yamaguchi, S., Katsushima, T., Sato, A., and Kumakura, T.: Water retention curve of snow with different grain sizes, Cold Reg. Sci. Technol., 64, 87–93, https://doi.org/10.1016/j.coldregions.2010.05.008, 2010.

Yamaguchi, S., Watanabe, K., Katsushima, T., Sato, A., and Kumakura, T.: Dependence of the water retention curve of snow on snow characteristics, Annals of Glaciology, 53, 6–12, https://doi.org/10.3189/2012AoG61A001, 2012.

Yamaguchi, S., Adachi, S., and Sunako, S.: A novel method to visualize liquid distribution in snow: superimposition of MRI and X-ray CT images, Annals of Glaciology, 65, e11, https://doi.org/10.1017/aog.2023.77, 2025.

Yen, Y.-C.: Review of thermal properties of snow, ice, and sea ice, vol. 81, US Army, Corps of Engineers, Cold Regions Research and Engineering Laboratory, https://apps.dtic.mil/sti/pdfs/ADA103734.pdf, 1981.

Zhang, Q., Liang, M., Zhang, Y., Wang, D., Yang, J., Chen, Y., Tang, L., Pei, X., and Zhou, B.: Numerical Study of Side Boundary Effects in Pore-Scale Digital Rock Flow Simulations, Fluids, 10, https://doi.org/10.3390/fluids10120305, 2025.

---

## Author Comment (AC3)

**Authors' reply to referee comments RC3 of the paper egusphere-2025-2903 entitled "Simulating liquid water distribution at the pore scale in snow: water retention curves and effective transport properties" by Bouvet et al.**

We thank very much Reviewer 3 for the comments. Please find below our point-by-point replies in blue color.

The study addresses the spatial distribution of liquid water in different snow types and estimates its effect on various transport properties. The authors analyzed in detail 34 scanned dry snow samples and presented the results on water distribution not only in the main text but also in a well-organized and complete Supplementary Information folder (SI). This was well-appreciated and made it very easy (and interesting) to scrutinize the results of the study. Considering that experimental data on liquid water distribution at the pore scale of snow are rare, the authors based their analysis on numerical experiments. This strategy relies on evidence that the computed properties are reasonable and in agreement with experimental findings. More specifically, to trust the outcome of the numerical experiments, I would expect that (i) the capacity of the model is shown for a very few experimental cases (see a proposal below) and that the (ii) general trends are in good agreement with experimental studies. This proof is missing here, and I have some doubts regarding the results of the numerical experiments as I explain below.

We modified the manuscript substantially according to the reviewer comments. Concerning point (i), improvements have been made on the evaluation of our simulations of WRCs. Additional comparisons to experimental studies from the literature have been included, so that, to the best of our knowledge, all the data available in the literature are now included in the paper. Concerning point (ii), the discussion about the agreement of our simulations against experimental data has been revised and includes new elements. The limitations of our study are now stated in a dedicated section in the revised paper. The description of our improvement, especially regarding points (i) and (ii), is provided below in the corresponding reviewer comment.

**Questions on the shape of the wetting curve**

Typically, the shapes of the wetting and drainage curves are similar with a shift of the drainage curve to higher (absolute) capillary pressure values. This would be manifested in similar values of the parameter "n" for wetting and drying and in higher "alpha" values for the wetting curves (with "alpha" and "n" as shape parameters of the van Genuchten model to fit the relationship between water content and capillary pressure). While the study shows the expected correlation between the "alpha" values (with alpha_wetting 1.3 alpha_drainage according to the data in the SI), the values of "n" are very different from the expectations, with more or less constant value "n" for all wetting curves (3 to 6) that differs from the drainage curve with values ranging from 5 to 20 (average 10). Such a clear mismatch of the shapes of the wetting and drainage curves is unusual and also contradicts the experimental outcome presented in Table 1 of Adachi et al. (2020) that reported very similar values of "n" for wetting and drying curves.
In short, I don't trust the shape of the wetting curve with such smooth shape (different from drainage curve) and converging to the porosity. I would expect a large amount of entrapped air with water saturation values considerably smaller than the porosity.

Two elements of the simulated WRCs are pointed out in the above comment: the shape of the WRCs for imbibition compared to drainage, and the fact that the WRCs converge to the porosity.

Regarding the shape of our simulated WRCs in imbibition and drainage, we obtain smoother inflexions of the WRCs in imbibition than in drainage (sharper inflexions), which translates to lower $n$ values for imbibition than for drainage; in addition, $n$ values vary little for imbibition (range from 3 to 6) while they are more scattered for drainage (range from 5 to 20). In both cases, $n$ values show no correlation with $\rho/d$. These findings are not consistent with the measurements of Adachi et al. (2020), as described by the reviewer. To further compare our results, we included additional comparisons in the revised version of the manuscript, shown in Figure 1: the studies of Katsushima et al. (2020) and Lombardo et al. (2025) provide $n$ values for snow from drainage experiments and imbibition experiments, respectively. In contrast with Adachi et al. (2020), which showed $n$ values similar for drainage and imbibition and in good agreement with the regression of Yamaguchi et al. (2012) (i.e. well correlated with $\rho/d$), these studies provide scattered values, ranging from 3 to 18 for the wetting process in Lombardo et al. (2025) and from 9 to 14 for the drainage in Katsushima et al. (2020). As in our data, these values are not correlated with $\rho/d$.

[Figure]

**Figure 1.** a) $\alpha_{vg}$, b) $n_{vg}$ and c) $\theta_w^r$ parameters of the VG model as a function of $\rho/d$ or $\rho$ for imbibition and drainage. The regressions of Yamaguchi et al. (2012) are shown by black lines, the values used to deduct those regressions are shown by black disks ("S-samples", composed of refrozen MF - see Yamaguchi et al. (2012)). The additional drainage measurements from Yamaguchi et al. (2012) are shown by circles (MF samples) and stars (RG samples). Measurements from Adachi et al. (2020) are shown by empty gray markers. Measurements from Lombardo et al. (2025) and Katsushima et al. (2013) are shown by empty orange triangles and blue circles. From this work, parameters from imbibition and drainage simulations are shown by colored disks and triangles, respectively, the colors showing the snow types. The proposed regressions based on our simulated data are shown by dashed and dotted lines (see Table 2).

50    This seems to indicate a more complex picture of the behavior of $n$ than in the data of Adachi et al. (2012), and we can not invalidate the $n$ values from our simulated WRCs. Moreover, different $n$ values for imbibition and drainage was also reported

by Ahrenholz et al. (2008) for sand (Tableau D1, Fig. 3 and 4 in their paper): performing a cycle of drainage - wetting - re-drainage, they obtained $n$ values of 11.4, 14.4, and 8.0 for the experiment referred as 'column 1' and 10.9, 4.9, and 8.7 for the experiment referred as 'column 2'.

55 **In the revised version**, the above considerations have been included. A discussion concerning the shape of our simulated WRCs and resulting $n$ values is provided, including the comparison to the literature values. These modifications are found in Section 3.1.3 "Analysis of the VG parameters" (paragraph "$n$ parameter") and in Figure 7 of the revised manuscript.

Regarding the second point of the reviewer's comment, the saturated water content $\theta_w^s$ equals the porosity in all our
60 simulations. Indeed, we computed a primary imbibition curve assuming that there is no air (the non-wetting phase NWP) residuals, as in the Mercury Injection Capillary Pressure (MICP) experiments. This is a simplification of the process that occurs in nature, as in reality, residual air likely exists. We chose to do this simplification because the amount of air residual in the structure given by the Pore Morphology Method may depend significantly on the chosen boundary conditions on the four sides of the volume that are not connected to the wetting phase (WP) and NWP reservoirs. In Vogel et al. (2005), impervious
65 boundary conditions are applied, whereas other conditions, such as symmetry, displaced fluid outlet, or invading fluid inlet, can also be applied (Berg et al., 2016). These boundary conditions may lead to different values of the NWP (air) residuals during the imbibition process, since these residuals can be trapped or not at the boundary. Testing that, we indeed obtained large discrepancies in the residual air content depending on the applied boundary conditions. The maximum water saturation ($\theta_w^s$) may range from 45% to 90% of the porosity. These effects of the boundary conditions on $\theta_w^s$ also concern other methods
70 to describe two-phase flows in 3D images and has been little discussed in the literature, to the best of our knowledge, except in (Galindo-Torres et al., 2016; Zhang et al., 2025) in the case of lattice Boltzmann simulations. This discrepancy was not, or to a much lesser extent, found in the residual of liquid water. Next, we looked at the literature on the residual air content in snow, we found $\theta_w^s$ values from measurements ranging from $0.6\phi$ to $0.9\phi$ (Yamaguchi et al., 2012; Katsushima et al., 2013; Adachi et al., 2020), with potentially large uncertainties linked to the experimental challenge of filling a snow column to its maximum,
75 as already highlighted for other porous media (e.g., Clayton, 1999; Cho et al., 2022). Yamaguchi et al. (2012) suggested using a unique, fixed value of $0.9\,\phi$ for snow, when using the VG model, which seems rather arbitrary. Finally, we tested the impact of the residual air content on the shape parameters of the VG model: even with $\theta_w^s$ as low as $0.6\phi$, the difference in terms of VG parameters is mainly visible for $\alpha_{\mathrm{vg}}$, with differences around 10% of the value with $\theta_w^s = \phi$. This was also reported in the experiments of Likos et al. (2014) and Farooq et al. (2024), which showed that having $\theta_w^s$ smaller than $\phi$ generally implies
80 greater $\alpha_{\mathrm{VG}}$ values, but has no significant impact on the $n_{\mathrm{VG}}$ values. To conclude, for all the above reasons, we decided to simulate complete wetting so that $\theta_w^s = \phi$, while addressing the associated sources of error, and we encourage future studies to examine this question.

**In the revised version**, we improved the description of the simulation to clarify that no residual air was simulated during wetting (Section 2.3). We added a description on the impact of the $\theta_w^s = \phi$ simplification on our findings, in Section 3.1.3
85 'Analysis of the VG parameters' and in the new Section 'Main limitation' in the revised version.

**Application of morphological pore network model (PNM)**

I had good experience with the application of morphological pore network models (MPN) for the simulation of drainage
90 processes in porous media, because it is controlled by the size distribution of the pore throats controlling the air invasion. The simulation of the wetting process, however, using spherical structural elements in the MPN, can be more critical: see Figures 5 and 6 in Ahrenholz et al. (2008) - you cited that study - and the sensitivity of the wetting curve with respect to the structural element. See the (i) large fractions of entrapped air in these figures and (ii) the similar shapes of wetting and drainage curves; these results are quite different compared to your findings shown for example in your Figure 4b (and in the data shown in
95 the SI). I am puzzled by the text on lines 138-140: "For imbibition, the simulation starts with the 3D image of dry snow. A minimum pore radius is defined according to Eq. (5), which then increases step by step. At each time step, a pore is filled with liquid water if (i) the pore radius is below the minimum pore radius, and (ii) the liquid water phase is connected to the liquid water reservoir." Such a description is notcomplete -> the air phase must be continuous as well; it can occur that the invading water encloses air bubbles and then the air must stay entrapped. Was this ignored in this model?

100

[Figure]

**Figure 2.** Example of a water retention curve estimated from a drainage simulation on a 3D tomographic snow sample of melt forms. The simulated 3D water distribution in the pores is shown at 3 different stages.

We agree with the reviewer. Air bubbles can be entrapped by the water during the imbibition process. As described in the previous comment, a complete imbibition process was simulated, and no residual air was computed. In the revised version, the description of the pore morphology method was substantially modified and includes all the above considerations (in the introduction and Section 2.3)

I'm also confused by the insets in Figure 2 that show the fluid distribution during the drainage process. The shape of the interface between water and air does not show the expected curvature 'with an air sphere invading the water filled pore space'.

We agree with the reviewer: the water menisci in Figure 2 did have a curvature in the wrong direction. This error was only made when plotting this particular example and not for the simulations. We changed Figure 2 accordingly, using the correct properties for a subsample of the image NH5. The corrected figure is presented here in Figure 2.

**Thermal conductivity**

The simulated thermal conductivity is a linear function of the water saturation. This linear relationship is unusual compared to findings for example in soils. In analogy to soils, I would assume that there is a fast increase in thermal conductivity with "the addition of a small amount of water" connecting the ice particles at the dry end. How can this strong linear relationship be explained? And what is with the case of theta=0 (after removal of the residual water)? Is there still a linear relationship between thermal conductivity and water content?

The thermal conductivity of water is four times less than the thermal conductivity of ice. Heat is thus conducted mostly through the ice skeleton. The dry snow thermal conductivity is described by a nonlinear (exponential) relationship with respect to the volume fraction of ice (Calonne et al., 2011), so that a small amount of ice connecting ice grains has a strong impact on conductivity. Due to its lower thermal conductivity, this is not true for water, and a change of water content in the snow microstructure only causes a linear change in snow thermal conductivity. In the revised version, we included an analytical model of the unsaturated thermal conductivity (the self-consistent estimate of thermal conductivity for a 3-phase composite aggregate of spherical inclusions), which reproduces well the linear relationship between water content and snow thermal conductivity (see Fig. 3).

**RECOMMENDATION**

[Figure]

**Figure 3.** Unsaturated thermal conductivity $k^u$ as a function of the effective saturation for (a) the whole set of snow samples, and (b) the 5 selected samples. Computations were performed on the snow images from the drainage simulations only. The dry density of the snow samples is represented by the colorbar. The suggested regression is shown by solid lines and the self-consistent estimate for 3 phases is shown by dashed lines.

[Figure]

**Figure 4.** Comparison between drainage and imbibition experimental WRCs of M and S samples from Adachi et al. (2020) (solid curves) and the numerical estimates of NH2 and NH5 samples (dots and dotted lines).

I propose that the authors compare their model with experimental data as follows: Adachi et al. (2020) plotted in Figure 6 measured drainage and wetting curves of three snow samples ("S", "M", and "L"). According to Table 1 in Adachi et al. (2020), the dry snow densities were around 500 kg/m$^3$ and the snow particle diameter was 1.2 and 1.9 mm for the two finer snow samples ("S" and "M"). Considering that the samples of Adachi et al. (2020) are melted forms (but refrozen) and that two of your 'melted form samples' ("NH2" and "NH5") have similar densities and snow particle sizes as in Adachi et al., it would be meaningful to compare samples "S" and "M" with "NH2" and "NH5", respectively. With such comparison, you may show that the prediction of the drainage curve is close to the experimental values (I doubt a bit that the prediction of the wetting is successful too).

As recommended by the reviewer, Figure 4 presents a comparison between the WRCs from the M and S samples of Adachi et al. (2020) and our numerical estimates of WRCs for the melt forms samples NH5 and NH2. Having both experimental and simulation results on the same plot is interesting but not easy to compare, as they do not come from the same snow samples, although we picked up the closest samples in terms of snow types. We can see some similarities, such as the drainage curve aspects, even if a large vertical shift is found for NH2. Considerable differences can also be observed, such as the slope of the imbibition curves. It is, however, difficult to impute those differences on either the differences of snow sample (i.e. snow microstructure), on the numerical estimations, or on the experimental conditions. Such direct comparison would be more meaningful by comparing WRCs simulated and measured on the exact same sample. In the paper, we compare simulations versus experiments in terms of the shape parameters of the VG model and comment on the overall trends. For further assessment of our simulations, additional comparisons have been included, with Lombardo et al. (2025) and Katsushima et al. (2013), in the revised manuscript.

After such experimental proof, the authors could relate the simulated properties to various metrics of the pore space. In the present version, the authors consider the mean curvature but there are other properties of the pore space that should be considered (maximum and width of pore size distribution, and connectivity of the pore space as for example quantified by the Euler-Poincarè-characteristics). After such more detailed quantification of the imaged pore space, new relations between pore space characteristics and transport properties could be deduced with more confidence. You could also consider to fit the 'tortuosity parameter' in the Mualem van Genuchten expression of hydraulic conductivity (now set to ½) to see if it changes with snow structure.

We agree that other parameters related to pore space might, potentially, give interesting correlations. However, we tried many of them without any successful results. Concerning some particular metrics recommended, we presently have no access to any efficient numerical tool to confidently characterize these quantities. The mean curvature distribution is an effective way to access size distributions (e.g. Lesaffre et al. (1998), Calonne et al. (2014)), which give good results on our dataset. In addition, we think the mean curvature distribution is a parameter of choice for drainage and imbibition phenomena as it is directly related to the capillary pressure through the Laplace equation (see also the response to RC2 on this topic). In the revised manuscript, we improved our characterization of pore size uniformity by taking the interquartile range (IQR) of the mean curvature distribution instead of the standard deviation. It provides better results, as the IQR is more suited to the shape of our distributions (not always Gaussian) than the standard deviation.

Concerning the tortuosity parameter that can be included in the Mualem van Genuchten expression of hydraulic conductivity, at this stage, we use the common value 1/2. Its evolution with the snow microstructure will be studied in further works. This point is mention in the revised version of the manuscript

I propose to show the figures on the relationship between water content and capillary pressure using a linear scale of the pressure head, focusing on small absolute capillary pressure values. This allows a better assessment of the shape of the saturation-pressure relationship.

We agree with the reviewer and modified our figures accordingly, using linear pressure scales.

180  I suggest that the authors also show some cross-sections of the modeled phase distribution (similar to the mean curvature now shown in SI). This would facilitate to assess if the phase distribution is reasonable.

  We agree with the reviewer. A new figure was made showing cross-sections of the 5 main images at 3 stages of the drainage and imbibition processes (Fig. 5). We added this figure to the manuscript in the subsection "WRCs of different
185 snow microstructures and their related VG fits" with the corresponding description: "Figure 2 presents vertical cross-sections of the drainage and imbibition simulations at three stages of water saturation for the five selected snow samples described in Table 1. The pore-scale distribution of liquid water in the microstructures can be observed. For imbibition, the small pores are first filled, and the larger pores are filled last. For drainage, it is the other way around, with water escaping the large pore first. Residual liquid water at the end of the drainage simulation is shown in pink. Air bubbles entrapped in the ice skeleton are shown
190 in yellow (for instance, in the bottom right of the 0A sample). This figure also highlights the fact that, for a given saturation, the liquid water distribution is different depending on the snow types, as well as between the imbibition and drainage processes."

  The authors should also provide more information on the morphological pore network model.

195  The description of the pore morphology model (PMM) was significantly improved in the revised version in Section 2.3. It includes now: "The PMM uses a sphere with a radius $r$ as a probe to detect the pore space that is accessible by the non-wetting phase (NWP, here the air). This radius is computed from the Young–Laplace equation: $r = 2\gamma\cos(\psi)/p_c$ where $p_c$ is the capillary pressure, $\gamma$ is the surface tension and $\psi$ is the contact angle between ice and liquid water. At 0°C, $\gamma = 0.0756$ N m$^{-1}$ and $\psi = 12°$ (Knight, 1967). Morphological operations, namely, erosion and /or dilation are used in the PMM (Hilpert
200 and Miller, 2001). The algorithm of the PMM can be decomposed into several steps as follows:

- In drainage condition, the porous medium is initially saturated with the wetting phase (WP, here the water) ($p_c = 0$). The invading NWP is connected to the inlet, which is the NWP reservoir, and the WP can escape through the outlet, the WP reservoir. (i) Then, $p_c$ is increased incrementally, i.e. $r$ is decreased incrementally. The solid phase is first dilated by a sphere with radius $r$. (ii) All the pores connected to the NWP reservoir are labeled as NWP. (iii) The NWP is then dilated
205  with the same sphere with radius $r$. The remaining pores are filled with the WP. The saturation can then be calculated. (iv) All the pores filled by the WP disconnected from the WP reservoir are considered as WP residual, and are no longer considered in the next steps. All these steps (i to iv) are repeated by increasing the value of the pressure, i.e. by decreasing the value of $r$. In the present case, the radius $r$ was decreased gradually with a step of 2 pixel size.

- In imbibition condition, the porous medium is initially saturated with the NWP. The invading WP is connected to the
210  inlet, which is the WP reservoir, and the NWP can escape through the outlet, the NWP reservoir. (i) Then, $p_c$ is decreased incrementally, i.e. $r$ is increased incrementally. The solid phase is first dilated by a sphere with radius $r$. (ii) The NWP is then dilated with the same sphere with radius $r$. (iii) All the pores connected to the WP are now labelled as WP. The remaining pores are NWP. The saturation can then be calculated. (iv) All the pores filled by the NWP disconnected from the NWP reservoir are considered as NWP residual, and are no longer considered in the next steps. As for the drainage
215  condition, all these steps (i to iv) are repeated for the next value of the pressure, i.e. the next value of $r$. In the present case, the radius $r$ was increased gradually with a step of 2 pixel size.

  In the present study, the PMM implemented in the SatudDict software was used to compute the WRCs of the 34 snow samples. We computed (i) a primary imbibition curve assuming that there is no air (NWP) residuals as in MICP experiments, thus $\theta_w^s = \phi$ in Eq. 9, and then (ii) a primary drainage curve until to reach the water (WP) residuals ($\theta_w^r$ in Eq. 9). In both cases,
220 symmetric boundary conditions are applied on the four sides of the volumes. The series of 3D snow images at different stages of drainage were then used to compute the relative permeability (i.e. the unsaturated hydraulic conductivity), the effective thermal conductivity, and the effective water vapor diffusivity.

**Specific comments**
225

[Figure]

**Figure 5.** Vertical cross-sections of the 5 images with drainage and imbibition processes at 3 stages of effective saturation. The cross-sections are taken in the center of the samples. The reservoir of the wetting phase is located on the bottom boundary and the reservoir of the non-wetting phase is located on the top boundary.

**Line 6:** The term "time series" does not fit entirely, because temporal aspects are not considered.
The sentence was rephrased as: 'A series of wet snow images at different stages of drainage and imbibition, with different water contents, was produced.'

**Line 39:** I would write "shape parameters", not "hydraulic parameters".
The manuscript was modified accordingly.

**Lines 42:** There are other lab methods (evaporation method and dew point methods for dry soils.
The manuscript was modified accordingly.

**Line 53:** I think that in Yamaguchi et al. (2012) there are 12 samples that are not melted forms ("rounded grains" in Table 1).
We agree and corrected the description of the samples of Yamaguchi et al. (2012) in the introduction. It now reads 'The measurements of Yamaguchi et al. (2012) are the most extensive, based on 60 snow samples, yet restricted to sieved or natural melt forms and natural rounded grains, with high density values from 360 to 630 kg m$^{-3}$, and grain size from 0.3 to 5.8 mm.'

**Line 56:** Richards equation, not Richard equations.
The manuscript was modified accordingly.

**Line 115:** I found no specific information on the morphological pore network model in that paper.
The reference was removed (wrongly used).

**Line 124:** I propose to refer to the Young-Laplace equation because gravity is not included in the model (but the applied pressure)
The equation was changed to the Young-Laplace equation.

**Lines 128-129:** The text "The evolution of the air and liquid water in the ice structure can be seen as a series of erosion and dilation operations that depends on the pore distribution (Ahrenholz et al., 2008)" does not fit here -> erosion and dilation can be used to determine the pore size in complex structure but the evolution of the fluid phases is controlled by the connectivity as well.
We agree with the reviewer. As described in the above reply to the recommendation, the description of the pore morphology model used was significantly improved in section 2.3 of the revised version of the manuscript and this sentence was deleted

**Line 131:** How is it possible to reach saturation when air will be entrapped (and there must be air entrapment)?
As described in a dedicated comment above, residual air was not simulated and the saturation water content was set to equal the porosity.

**Figure 2** : Write in the captions the name of the sample. Is there a melt form sample with such a high porosity?
Modified accordingly. The name of the snow sample was included in the caption.

**Line 144:** Why the "initial" liquid water? The connectivity of the liquid phase must be tested for each drainage step.
We agree with reviewer. The description of the pore morphology model used was significantly improved in section 2.3 of the revised version of the manuscript and takes into account this point.

**Line 148:** Have you tested the REV for the porosity? How is the porosity changing with image size?
The REV of the porosity for different types of snow has been investigated in Coléou et al. (2001), showing that most of the times the minimum REV is cubes with side of size 2 mm, which confirms that all our snow samples are larger than the REV for porosity.

**Figure 3:** this figure is "basic textbook" and is not needed here; please provide always the units of alpha.
The figure has been modified accordingly and the unit of $\alpha_{vg}$ was included. We think the figure is useful to make these concepts accessible to the snow community.

**Line 166:** it is the inverse of alpha that is related to the inflection point.
Modified accordingly.

**Line 167:** it is the "width" of the pore size distribution that controls the value of parameter "n".
Modified accordingly.

**Lines 168/169:** this is not correct considering the presence of entrapped air.
We did not simulate entrapped air. The lack of entrapped air in our simulations is addressed in a specific point above.

**Line 169:** state that the porosity can be deduced from dry bulk density and the density of ice.
Modified accordingly. The link between snow density and snow porosity is now presented in Section 2.2 of the manuscript.

**Line 229:** I am not sure that the smooth shape of the wetting curve is realistic.
The comment on the shape of our simulated WRCs is addressed in a specific point above.

**Figure 6b:** I cannot see the drainage regression line.
Given the observed scatter, no regression of the $n$ parameter was provided as a function of $\rho/d$. However, regressions based on the mean curvature are provided, and are shown here in Figure 6.

**Line 239:** The exponential trend is not obvious because you have not enough values on the left side of the x-axis compared to Yamaguchi et al. (2012); in your data, there is a single sample with a x-value around 250000 kg/m$^4$; and a linear fit is almost as good as the exponential one for x-values $\geq$250000 kg/m$^4$.
We agree and deleted the term 'exponential' to describe the relationship of our simulated $\alpha$ parameter with $\rho/d$. We still used an exponential regression for $\alpha$ to be consistent with the regression form presented by Yamaguchi et al. (2012).

**Line 259** : the differences in the findings of Yamaguchi et al. (2012) as function of snow type is important; but in your study you have also a few melted forms; have you checked if you find a trend in your melted-form-subset similar to Yamaguchi?
The $n_{\mathrm{vg}}$ values estimated for our five MF samples are further away from the regression of Yamaguchi et al. (2012) than the $n_{\mathrm{vg}}$ of Yamaguchi et al. (2012) measured on their MF samples (S samples and MF types from the N samples). The $n_{\mathrm{vg}}$ values estimated for our other snow types (RG, DF, PP and DH) are even more further away from the regression of Yamaguchi et al. (2012), which is consistent with the fact that the regression of Yamaguchi et al. (2012) was developed based on melt forms samples mostly. A sentence was included in the revised manuscript to highlight this effect: "Our estimated $n_{\mathrm{vg}}$ values for drainage overall do not follow the regression of Yamaguchi et al. (2012), even if the $n_{\mathrm{vg}}$ values of the melt form samples are closer to the regression compared to the other snow types.".

**Lines 262** : To characterize the pore size variation, why don't you use the pore size distribution of the scanned sample? You should have access to the imaged pore size distribution (it is used as basis of the pore network model) and you could easily compute its width. This would be a more direct link to pore sizes compared to the curvature.
We tried several parameters in order to correlate the evolution of the VG model with the microstructure. The interquartile range of the curvature distributions appears to be the best one in that case, as described in the dedicated comment above.

**Line 272 and Figure 7:** the correlation is rather weak, and the nonlinearity is not very strong (similar correlation values for a linear fit); please explain colors in Figure 7.
Thank you for pointing that out. The linear fit is simpler and similar in terms of R2 (0.15 compared to 0.26 for the nonlinear fit) and MAE (2.5 compared to 2.3). However, our choice of regression is also motivated by the bounds of the function: the infinite bound of $n_{\mathrm{vg}}$ corresponds to the step WRC function that would be obtained with uniform pore sizes, and an asymptote value of $n_{\mathrm{vg}}$ for large values of $\sigma_{\mathrm{MC}}$ seems better than the negative values that would be obtained with a linear fit.

**Line 283:** this is an interesting result.

[Figure]

**Figure 6.** $n_{vg}$ parameter as a function of the interquartile range $IQR_{MC}$ obtained for the mean curvature distribution computed on each 3D dry snow image. Regressions for imbibition and drainage are shown with dotted and dashed lines.

**Lines 286/287** : again-> I don't think that this is reasonable.

As already mentioned, we did not simulate residual air during imbibition, which indeed constitutes a simplification of the process. This point was clarified in the revised version of the manuscript.

**Lines 303/304:** The correlation with the standard deviation of mean curvature is rather weak and such regression may result in large prediction errors.

We improved our characterization of the pore size distribution by taking the interquartile range of the mean curvature distribution, instead of the standard deviation, as the distributions of mean curvature do not always correspond to a Gaussian shape. As seen in Figure 6, it enables to describe both $n_{vg}$ for imbibition and drainage with the same inverse function, with standard deviations corresponding to: $(3.8 \pm 0.3) + (3.6 \pm 1.7)/IQR_{MC}$ for imbibition and $(6.4 \pm 1.6) + (25.6 \pm 10.3)/IQR_{MC}$ for drainage.

**Line 314:** Can you comment on the effect of applying different methods for the quantification of grain size?

The grain diameter estimation of Yamaguchi et al. (2012) slightly differs from ours: our definition is based on 3D snow images, whereas Yamaguchi et al.'s is based on its 2D counterpart obtained from outlines of disconnected ice grains. While the conceptual definition is quite similar for both of the methods, in which an "equivalent diameter" is estimated, some typical differences between 2D and 3D are well-documented (e.g Brzoska et al. (1999), Cooperdock et al. (2019)):

- The 2D method tends to neglect all snow structures related to necks between large disconnected grains;

- Due to the combination of gravity and projective effects, grain diameters may appear different in 2D than they really are in 3D;

- Tiny air bubbles or holes inside ice structures are often overlooked in 2D outlines.

Depending on grain types and specific morphologies, these methodological differences may result in an overestimation or an underestimation in grain sizes, so that an exact conversion between the two approaches cannot be reasonably achieved. As a consequence, comparing our results to those of Yamaguchi et al. (2012) in Fig 7 should be done by keeping in mind that estimation errors on the $\rho/d$ values might exist, potentially as systematic errors (stretching of regression curves to the right or to the left) or as an increased dispersion of the dataset. Based on existing literature comparisons (e.g Brzoska et al. (1999), Cooperdock et al. (2019)), and on the figures obtained here (number and diversity of investigated snow samples and morphologies), we are quite confident that such errors are moderate and that overall comparisons between regressions obtained from the different methodologies are valid. Precise comparisons are, however,

more difficult and the difference between the regression of Yamaguchi et al. (2012) and our regression for $\alpha_{vg}$ in drainage could be e.g. interpreted as a consequence of this methodology difference. In this regard, our regression can be seen as an updated version of Yamaguchi et al. (2012)'s regression, proposing estimations that are based on systematic 3D diameter estimations, which are well adapted to tomographic measurements. The above explanations were added in the "3.1.3 Analysis of the VG parameters" subsection.

**Figure 8a and 8b:**  please use the same intervals on x axis (0.1).
Modified accordingly.

**Figure 8:**  why do the Yamaguchi curves have the same maximum saturation? I thought they used the 90% rule as maximum water content.
To ease the reading of the different models, and their different maximum water saturation, we plotted the WRCs as a function of the effective saturation instead of the water content.

**Line 317:**  I do not agree with this conclusion considering the shape of the wetting curve, the lack of entrapped air in the model, and not enough samples to proof the non-linearity for small rho/d-values (smaller than 250'000 kg/m$^4$).
The comment on the shape of our simulated WRCs and the lack of entrapped air is addressed in a specific point above. We agree that we lack more samples to confirm the exponential decrease of the $\alpha$ parameter with rho/d, as in the data of Yamaguchi et al. (2012). We still used an exponential regression to be consistent with the regression form presented by Yamaguchi et al. (2012).

**Lines 322/322:**  the new VG-model reproduces well the simulated values - but it is not clear that the simulations are correct.
This section was fully revised so that we now compare the VG model proposed in Yamaguchi et al. (2012), in Yamaguchi et al. (2012), in Daanen and Nieber (2009), without basing it on a comparison with the simulated WRCs on our snow samples, but based on a more independent approach. Indeed, instead of doing a comparison based on our snow samples, we defined two 'imaginary' snow samples. The properties of those samples have been chosen to be representative of melt forms (sample 1: d = 1.5 mm, $\rho$ = 450 kg m$^{-3}$, IQR$_{MC}$ = 5 mm$^{-1}$) and precipitation particles (sample 2: d = 0.1 mm, $\rho$ = 130 kg m$^{-3}$, IQR$_{MC}$ = 15 mm$^{-1}$). The section was renamed 'Application of the different VG models on two representative snow samples'.

**Figure 10:**  I'm surprised that there are some samples with conductivity values of 10cm/day at effective water saturation of 0. How can this be? Do you have water flow along the residual water?
Thank you for pointing that out. Those points correspond to numerical artifacts that we had on a few images and have been removed from the figure. At this saturation, the liquid water phase is not connected from top to bottom, thus $\mathbb{K}_w^u = 0$ m s$^{-1}$.

**Equation 8 (and figures 10):**  you use the 'standard expression' with tortuosity factor "tau" of 1/2. This is an average value and can change with the porous medium (as discussed in Mualem 1976)
We added the definition of the tortuosity factor in the method section: "The exponent $\tau_{vg}$ describes the effects of the connectivity and tortuosity of the flow paths and is set to $1/2$, which is its default value (e.g., Mualem, 1976; Vereecken et al., 2010)." We also addressed this approximation in the section "Main limitations" that reads: "we chose a simple parameterization of the VG formulation, which can be refined using more parameters such as $m_{vg}$ and $\tau_{vg}$ (see eq. 2 and 11)."

**Line 343:**  the good agreement between the simulations and the 'standard formulation' of the Mualem-van Genuchten framework is no strong proof that the simulations are correct. You may try to fit the factor "tau" in Mualem van Genuchten for the different samples and check if the value changes with snow property.
Concerning the tortuosity parameter that can be included in the Mualem van Genuchten expression of hydraulic conductivity, at this stage, we use the common value 1/2. Its evolution with the snow microstructure will be studied in further work. This point is mentioned in the revised version of the manuscript.

[Figure]

**Figure 7.** Relative water vapor diffusivity $D^{\mathrm{u}}/D$ as a function of the liquid water saturation $S_w$ for drainage simulations of (a) the whole set of snow samples and (b) the 5 selected samples. The dry density of the snow samples is given by the colorbar. The proposed regression of unsaturated diffusivity is shown by a black solid line, and the models of Millington and Quirk (1961) and Moldrup et al. (2000) are shown with gray dotted and dashed lines respectively.

**Figure 11:** What are the predictions when you compute it without residual water? Maybe it is better to plot it as function of water content, not effective saturation.

Estimates of the thermal conductivity of snow without residual water, so for fully dry snow, can be found in Calonne et al. (2011), as we used the same set of dry snow images. Values for dry snow range from 0.06 W m$^{-1}$ K$^{-1}$ for a snow sample of density 103 kg m$^{-3}$ and 0.77 W m$^{-1}$ K$^{-1}$ for a snow sample of density 544 kg m$^{-3}$.

**Line 386:** how do you know that it is over-estimated?

The thermal conductivity estimated in the Crocus model (Eq.14) is higher than the simulated thermal conductivity on 3D images of snow, as shown in Figure 12, and the higher the water content, the larger the difference. The Crocus model applies the same thermal conductivity-density relationship, regardless of the proportion of ice and liquid water, using the regression from Yen (1981) developed for dry snow. However, water conducts heat four times less than ice, which explains the overestimation.

**Line 407/408:** That is an interesting finding (similar approaches exist for gas diffusion in soils ).

**Line 414** : you should compare this with other models like Moldrup et al. (2000).

Following this remark, we added two models for $D^{\mathrm{u}}$ in unsaturated soils: the model of Millington and Quirk (1961) and of Moldrup et al. (2000). Both models are in good agreement with the numerical estimate and show similar relationships to our regression, bounding our regression with slightly higher and lower values, respectively. This comparison is now included in the revised version of the manuscript with Fig. 7 and a comment in the text : "This relationship follow the general form of estimates of $D^{\mathrm{u}}/D_{\mathrm{dry}} = (1 - \theta_w/\phi)^a$ classically used for soils (Kristensen et al., 2010) with $a = 10/3$ in Millington and Quirk (1961) and $a = 5/2$ in Moldrup et al. (2000). The regression is shown in Fig. 7 by the black line, along with the soil models of Millington and Quirk (1961) and Moldrup et al. (2000) that are displayed with gray dotted and dashed lines. The latter two models fairly represent the behavior of $D^{\mathrm{u}}$ and bound the regression for snow with higher and lower values. Both are used for predicting $D^{\mathrm{u}}$ in structureless natural soils."

**Line 429:** you should state this in the captions of figures 10-14.

The revised manuscript was modified accordingly.

**Line 441:** the pore size was not presented in this study.

We replaced 'pore size' by 'mean curvature distribution'.

**References**

Adachi, S., Yamaguchi, S., Ozeki, T., and Kose, K.: Hysteresis in the water retention curve of snow measured using an MRI system, in: Proceedings to the 2012 International Snow Science Workshop, Anchorage, Alaska, vol. 485, https://arc.lib.montana.edu/snow-science/objects/issw-2012-918-922.pdf, 2012.

Adachi, S., Yamaguchi, S., Ozeki, T., and Kose, K.: Application of a magnetic resonance imaging method for nondestructive, three-dimensional, high-resolution measurement of the water content of wet snow samples, Frontiers in Earth Science, 8, https://doi.org/10.3389/feart.2020.00179, 2020.

Ahrenholz, B., Tölke, J., Lehmann, P., Peters, A., Kaestner, A., Krafczyk, M., and Durner, W.: Prediction of capillary hysteresis in a porous material using lattice-Boltzmann methods and comparison to experimental data and a morphological pore network model, Advances in Water Resources, 31, 1151–1173, https://doi.org/10.1016/j.advwatres.2008.03.009, 2008.

Berg, S., Rücker, M., Ott, H., Georgiadis, A., van der Linde, H., Enzmann, F., Kersten, M., Armstrong, R., de With, S., Becker, J., and Wiegmann, A.: Connected pathway relative permeability from pore-scale imaging of imbibition, Advances in Water Resources, 90, 24–35, https://doi.org/https://doi.org/10.1016/j.advwatres.2016.01.010, 2016.

Brzoska, J. B., Lesaffre, B., Coléou, C., Xu, K., and Pieritz, R. A.: Computation of 3D curvatures on a wet snow sample, Eur. Phys. J. AP, 7, 45–57, https://doi.org/10.1051/epjap:1999198, 1999.

Calonne, N., Flin, F., Morin, S., Lesaffre, B., Rolland du Roscoat, S., and Geindreau, C.: Numerical and experimental investigations of the effective thermal conductivity of snow, Geophys. Res. Lett., 38, L23 501, https://doi.org/10.1029/2011GL049234, 2011.

Calonne, N., Flin, F., Geindreau, C., Lesaffre, B., and Rolland du Roscoat, S.: Study of a temperature gradient metamorphism of snow from 3-D images: time evolution of microstructures, physical properties and their associated anisotropy, The Cryosphere, 8, 2255–2274, https://doi.org/10.5194/tc-8-2255-2014, 2014.

Cho, J. Y., Lee, H. M., Kim, J. H., Lee, W., and Lee, J. S.: Numerical simulation of gas-liquid transport in porous media using 3D color-gradient lattice Boltzmann method: trapped air and oxygen diffusion coefficient analysis, Engineering Applications of Computational Fluid Mechanics, 16, 177–195, https://doi.org/10.1080/19942060.2021.2008012, 2022.

Clayton, W. S.: Effects of pore scale dead-end air fingers on relative permeabilities for air sparging in soils, Water Resources Research, 35, 2909–2919, https://doi.org/10.1029/1999WR900202, 1999.

Coléou, C., Lesaffre, B., Brzoska, J.-B., Ludwig, W., and Boller, E.: Three-dimensional snow images by X-ray microtomography, Annals of Glaciology, 32, 75–81, https://doi.org/10.3189/172756401781819418, 2001.

Cooperdock, E. H. G., Ketcham, R. A., and Stockli, D. F.: Resolving the effects of 2-D versus 3-D grain measurements on apatite (U–Th) / He age data and reproducibility, Geochronology, 1, 17–41, https://doi.org/10.5194/gchron-1-17-2019, 2019.

Daanen, R. P. and Nieber, J. L.: Model for Coupled Liquid Water Flow and Heat Transport with Phase Change in a Snowpack, Journal of Cold Regions Engineering, 23, 43–68, https://doi.org/10.1061/(ASCE)0887-381X(2009)23:2(43), 2009.

Farooq, U., Gorczewska-Langner, W., and Szymkiewicz, A.: Water retention curves of sandy soils obtained from direct measurements, particle size distribution, and infiltration experiments, Vadose Zone Journal, 23, e20 364, https://doi.org/10.1002/vzj2.20364, 2024.

Galindo-Torres, S., Scheuermann, A., and Li, L.: Boundary effects on the Soil Water Characteristic Curves obtained from lattice Boltzmann simulations, Computers and Geotechnics, 71, 136–146, https://doi.org/https://doi.org/10.1016/j.compgeo.2015.09.008, 2016.

Katsushima, T., Yamaguchi, S., Kumakura, T., and Sato, A.: Experimental analysis of preferential flow in dry snowpack, Cold Regions Science and Technology, 85, 206–216, https://doi.org/10.1016/j.coldregions.2012.09.012, 2013.

Katsushima, T., Adachi, S., Yamaguchi, S., Ozeki, T., and Kumakura, T.: Nondestructive three-dimensional observations of flow finger and lateral flow development in dry snow using magnetic resonance imaging, Cold Regions Science and Technology, 170, 102 956, https://doi.org/10.1016/j.coldregions.2019.102956, 2020.

Knight, C. A.: The contact angle of water on ice, Journal of Colloid and Interface Science, 25, 280–284, https://doi.org/10.1016/0021-9797(67)90031-8, 1967.

Kristensen, A. H., Thorbjørn, A., Jensen, M. P., Pedersen, M., and Moldrup, P.: Gas-phase diffusivity and tortuosity of structured soils, Journal of Contaminant Hydrology, 115, 26–33, https://doi.org/10.1016/j.jconhyd.2010.03.003, 2010.

Lesaffre, B., Pougatch, E., and Martin, E.: Objective determination of snow-grain characteristics from images, Annals of Glaciology, 26, 112–118, https://doi.org/10.3189/1998AoG26-1-112-118, 1998.

Likos, W. J., Lu, N., and Godt, J. W.: Hysteresis and Uncertainty in Soil Water-Retention Curve Parameters, Journal of Geotechnical and Geoenvironmental Engineering, 140, 04013 050, https://doi.org/10.1061/(ASCE)GT.1943-5606.0001071, 2014.

Lombardo, M., Fees, A., Kaestner, A., van Herwijnen, A., Schweizer, J., and Lehmann, P.: Quantification of capillary rise dynamics in snow using neutron radiography, EGUsphere, 2025, 1–36, https://doi.org/10.5194/egusphere-2025-304, 2025.

Millington, R. and Quirk, J.: Permeability of porous solids, Transactions of the Faraday Society, 57, 1200–1207, https://doi.org/10.1039/TF9615701200, 1961.

460 Moldrup, P., Olesen, T., Schjønning, P., Yamaguchi, T., and Rolston, D. E.: Predicting the Gas Diffusion Coefficient in Undisturbed Soil from Soil Water Characteristics, Soil Science Society of America Journal, 64, 94–100, https://doi.org/10.2136/sssaj2000.64194x, 2000.

Mualem, Y.: A new model for predicting the hydraulic conductivity of unsaturated porous media, Water Resources Research, 12, 513–522, https://doi.org/10.1029/WR012i003p00513, 1976.

Vereecken, H., Weynants, M., Javaux, M., Pachepsky, Y., Schaap, M. G., and Genuchten, M. v.: Using Pedotransfer Functions to
465 Estimate the van Genuchten–Mualem Soil Hydraulic Properties: A ReviewAll rights reserved. No part of this periodical may be reproduced or transmitted in any form or by any means, electronic or mechanical, including photocopying, recording, or any information storage and retrieval system, without permission in writing from the publisher., Vadose Zone Journal, 9, 795–820, https://doi.org/10.2136/vzj2010.0045, 2010.

Vogel, H.-J., Tölke, J., Schulz, V. P., Krafczyk, M., and Roth, K.: Comparison of a Lattice-Boltzmann Model, a Full-Morphology
470 Model, and a Pore Network Model for Determining Capillary Pressure-Saturation Relationships, Vadose Zone Journal, 4, 380–388, https://doi.org/https://doi.org/10.2136/vzj2004.0114, 2005.

Yamaguchi, S., Watanabe, K., Katsushima, T., Sato, A., and Kumakura, T.: Dependence of the water retention curve of snow on snow characteristics, Annals of Glaciology, 53, 6–12, https://doi.org/10.3189/2012AoG61A001, 2012.

Yen, Y.-C.: Review of thermal properties of snow, ice, and sea ice, vol. 81, US Army, Corps of Engineers, Cold Regions Research and
475 Engineering Laboratory, https://apps.dtic.mil/sti/pdfs/ADA103734.pdf, 1981.

Zhang, Q., Liang, M., Zhang, Y., Wang, D., Yang, J., Chen, Y., Tang, L., Pei, X., and Zhou, B.: Numerical Study of Side Boundary Effects in Pore-Scale Digital Rock Flow Simulations, Fluids, 10, https://doi.org/10.3390/fluids10120305, 2025.